# Full-Spectrum Graph Neural Networks: Expressive and Scalable

Xiaohan Wang [* 1]   Deyu Bo [* 1]   Longlong Li [1]   Kelin Xia [1]

## Abstract

It is well established that spectral graph neural networks (GNNs) can universally approximate node signals; however, their expressive power remains bounded by the 1-dimensional Weisfeiler–Lehman test, which is mirrored in their lack of universality for higher-order signals. To go beyond this bound, we propose the Full-Spectrum GNNs (FSPECGNNs), a second-order generalization of classical spectral GNNs. FSPECGNN advances spectral filtering from two perspectives: (1) it lifts signals from the node domain to the node-pair domain; and (2) it extends the univariate spectral filter over eigenvalues to a bivariate filter over eigenvalue pairs. We show that classical spectral GNNs arise as a diagonal special case of FSPECGNNs, and prove that FSPECGNNs can be at most as expressive as Local 2-GNN while universally approximating node-pair signals, the latter being particularly beneficial for heterophilic graph learning. Moreover, FSPECGNNs admit scalable implementations that avoid explicit node-pair-level computations; combined with a low-rank approximation that reduces full-spectrum convolution to a combination of polynomial spectral filters, it enables learning on large graphs. Empirically, FSPECGNNs validate the predicted expressivity on substructure-counting benchmarks and delivers strong performance on heterophilic benchmarks. Our code is available at https://github.com/xwangxshi/FSpecGNN.

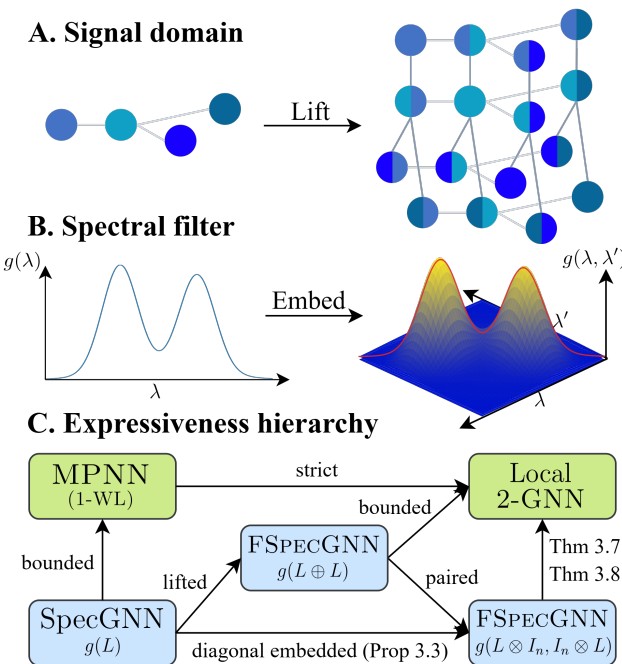

**A. Signal domain**

Lift

**B. Spectral filter**

$g(\lambda)$

Embed

$g(\lambda, \lambda')$

$\lambda'$

$\lambda$

$\lambda$

**C. Expressiveness hierarchy**

MPNN (1-WL) — strict → Local 2-GNN

bounded

bounded

FSPECGNN $g(L \oplus L)$

lifted

paired

Thm 3.7
Thm 3.8

SpecGNN $g(L)$ — diagonal embedded (Prop 3.3) → FSPECGNN $g(L \otimes I_n, I_n \otimes L)$

*Figure 1.* FSPECGNN generalizes spectral filtering by lifting signals to the node-pair domain and extending univariate eigenvalue filters to bivariate filters over eigenvalue pairs. Proposition 3.3 embeds spectral GNN (SpecGNN) to the diagonal restriction of FSPECGNN, and Theorem 3.7 and Theorem 3.8 establishes that FSPECGNN can be at most as expressive as Local 2-GNN.

## 1. Introduction

Graph Neural Networks (GNNs) are a standard paradigm for learning from data with underlying graph structure, combin-

ing feature transformations with graph propagations (Defferrard et al., 2016; Hamilton et al., 2017; Velickovic et al., 2018). A central line of research studies the expressivity of GNNs, which provides a principled lens for comparing the representational power of different architectures in learning from graph-structured data. A mainstream framework for such analyses characterizes expressivity by a model's ability to distinguish non-isomorphic graphs, ranging from the 1-dimensional Weisfeiler–Lehman (1-WL) test (Xu et al., 2019) to more recent homomorphism-based expressivity (Zhang et al., 2024), yielding a well-established expressiveness hierarchy for GNNs. Within this hierarchy, a standard way to surpass 1-WL is to lift message passing from the node domain $V$ to higher-order domains, such as node pairs $V \times V$ and, more generally, $k$-tuples (Morris et al., 2019; 2020). In contrast, spectral GNNs, which parameterize propagation as Laplacian filtering, have a comparatively

*Equal contribution  [1]Division of Mathematical Sciences, School of Physical and Mathematical Sciences, Nanyang Technological University, Singapore 637371, Singapore. Correspondence to: Kelin Xia <xiakelin@ntu.edu.sg>.

*Proceedings of the 43rd International Conference on Machine Learning*, Seoul, South Korea. PMLR 306, 2026. Copyright 2026 by the author(s).

incomplete expressivity picture: while they are universal for node-signal approximation (Wang & Zhang, 2022; Roth & Liebig, 2025), their ability to distinguish non-isomorphic graphs remains bounded by 1-WL (Wang & Zhang, 2022). This raises a natural question: *what is the spectral counterpart of lifting that enables spectral GNNs to systematically go beyond 1-WL?*

**Our contributions.** We propose *Full-Spectrum GNNs* (FSPECGNNs) as a second-order generalization of spectral GNNs; the same construction also extends directly to higher orders. Concretely, instead of filtering node signals $x \in \mathbb{R}^{|V|}$, full-spectrum filtering operates on node-pair signals $\varepsilon \in \mathbb{R}^{|V \times V|}$ and generalizes classical spectral filtering

$$U\big(g_\lambda \odot (U^\top x)\big) = \sum_i g(\lambda_i)\, u_i u_i^\top x \qquad (1)$$

$$\Downarrow$$

$$U\big(G_\lambda \odot (U^\top \varepsilon U)\big)U^\top = \sum_{i,j} g(\lambda_i, \lambda_j)\, u_i u_i^\top \varepsilon\, u_j u_j^\top \quad (2)$$

where $L = U\Lambda U^\top$ is the eigendecomposition of the graph Laplacian with eigenvectors $U = [u_1, \ldots, u_n]$ and eigenvalues $\Lambda = \mathrm{diag}(\lambda_1, \ldots, \lambda_n)$. FSPECGNN naturally extends the classic univariate spectral filter $g_\lambda = (g(\lambda_i))_{1 \leq i \leq |V|}$ to a bivariate matrix $G_\lambda = (g(\lambda_i, \lambda_j))_{1 \leq i,j \leq |V|}$. Proposition 3.3 further shows that Eq. (1) is recovered within Eq. (2) via a diagonal embedding.

FSPECGNN is **expressive**: it can surpass the 1-WL upper bound. In particular, under appropriate conditions, Theorem 3.8 shows that FSPECGNN matches the distinguishing power of Local 2-GNN (Zhang et al., 2024). Complementarily, Theorem 3.4 establishes a universal approximation result for any node-pair signals. FSPECGNN is also **scalable**: Section 3.4 discusses implementations of FSPECGNN to avoid explicit computations in the node-pair domain. We further propose a low-rank approximation in Eq. (3), which approximates a full-spectrum convolution using only a small number of polynomial spectral filters (He et al., 2021; 2022; Wang & Zhang, 2022). This leads to implementations Eq. (5) whose computational complexity is comparable to that of existing spectral GNNs.

We further demonstrate the utility of FSPECGNN on heterophilic graph learning tasks, where adjacent nodes often carry different labels and thus naive neighborhood aggregation can be detrimental (Luan et al., 2023). To understand heterophily from a spectral perspective, we study the optimal convolution that maximally separates representations across different labels. Theorem 4.1 shows that the resulting convolution suppresses (and, as the dimension grows to infinity, assigns zero weight to) entries corresponding to inter-class connections. We then prove a fundamental limitation: Theorem 4.2 establishes that such an optimal convolution cannot be realized within the class of classical spectral convolutions. In contrast, FSPECGNN can realize this optimal operator via a slight modification of our implementation. Together, these results identify heterophily as an inherently second-order phenomenon and highlight the importance of scalable second-order architectures such as FSPECGNN.

Finally, we evaluate FSPECGNN on substructure-counting benchmarks (Zhang et al., 2024), where the empirical results align with our expressivity predictions. Moreover, FSPECGNN is substantially more efficient in practice, achieving roughly $5\times$ lower runtime than the corresponding spatial second-order baselines and the lowest peak GPU memory among all compared methods. We also assess FSPECGNN on standard heterophilic node-classification benchmarks and demonstrate consistent gains over a range of representative baselines, while maintaining runtime and GPU consumption comparable to those of baseline models.

## 2. Preliminaries

**Notations.** Let $G = (V, E)$ be an undirected graph with $|V| = n$, $|E| = m$, $m \ll n^2$. The adjacency matrix is $A \in \mathbb{R}^{n \times n}$, the degree matrix is $D \in \mathbb{R}^{n \times n}$, and node features are represented by a matrix $X \in \mathbb{R}^{n \times d}$. We further fix a graph Laplacian $L \in \mathbb{R}^{n \times n}$ by $D - A$ or $I - D^{-1/2}AD^{-1/2}$. When all choices are admissible, we refer to any of them simply as $L$.

**Spectral GNNs.** The Laplacian admits the eigendecomposition $L = U\Lambda U^\top$, where $U = [u_1, \ldots, u_n]$ is an orthonormal eigenvector matrix and $\Lambda = \mathrm{diag}(\lambda_1, \ldots, \lambda_n)$ contains the eigenvalues. For a graph signal $x \in \mathbb{R}^n$, the graph Fourier transform (GFT) and its inverse are $\hat{x} = U^\top x$, $x = U\hat{x}$.

**Definition 2.1.** Given a continuous function $g : \mathbb{R} \to \mathbb{R}$, the corresponding *spectral filter* is the vector

$$g_\lambda := (g(\lambda_1), g(\lambda_2), \cdots, g(\lambda_n)).$$

Then, the *spectral convolution* is defined as

$$\begin{aligned} g_\lambda *_G x &:= U\big(g_\lambda \odot (U^\top x)\big) = Ug(\Lambda)U^\top x \\ &= \sum_i g(\lambda_i)\, u_i u_i^\top x, \end{aligned}$$

where $g(\Lambda) := \mathrm{diag}(g_\lambda) \in \mathbb{R}^{n \times n}$. When graph convolution is of the form $g_\lambda *_G x$, the resulting architecture is commonly referred to as a spectral GNN.

A common choice of the function $g$ is a polynomial $p(t) = \sum_{k=0}^K \theta_k t^k$ with learnable coefficients $\{\theta_k\}_{k=0}^K$. Define the corresponding polynomial of Laplacian by $p(L) := \sum_{k=0}^K \theta_k L^k$. With $L = U\Lambda U^\top$, we obtain

$$p(L) = U\Big(\sum_{k=0}^K \theta_k \Lambda^k\Big)U^\top = Up(\Lambda)U^\top$$

equivalently, we have $p(\boldsymbol{\lambda}) *_G x = p(L)x$, which avoid computing the eigendecomposition explicitly. For simplicity, given an arbitrary function $g$, we denote the spectral convolutional matrix by $g(L) := Ug(\Lambda)U^\top$, then similarly $g_\lambda *_G x = g(L)x$.

**Tensor product of matrices.** To describe the second-order spectral framework, we introduce

**Definition 2.2.** Let $A \in \mathbb{R}^{m \times n}$ and $B \in \mathbb{R}^{p \times q}$. Their *tensor product*, a.k.a Kronecker product, is the block matrix

$$A \otimes B := \begin{bmatrix} a_{11}B & \cdots & a_{1n}B \\ \vdots & \ddots & \vdots \\ a_{m1}B & \cdots & a_{mn}B \end{bmatrix} \in \mathbb{R}^{(mp) \times (nq)}.$$

For a matrix $X \in \mathbb{R}^{n \times n}$, the *vectorization* map is defined by column-stacking:

$$\mathrm{vec}(X) := \left[ X_{:,1}^\top, \ X_{:,2}^\top, \ \ldots, \ X_{:,n}^\top \right]^\top \in \mathbb{R}^{n^2},$$

where $X_{:,i} \in \mathbb{R}^n$ denotes the $i$-th column of $X$.

**Lemma 2.3** (Van Loan (2000))**.** *The Kronecker product and the vectorization map satisfy the following*

1. $\mathrm{vec} : \mathbb{R}^{n \times n} \to \mathbb{R}^{n^2}$ *is a linear space isomorphism.*

2. $A, B, X \in \mathbb{R}^{n \times n}$, $(B^\top \otimes A) \mathrm{vec}(X) = \mathrm{vec}(AXB)$.

3. *For* $u, v \in \mathbb{R}^n$, $v \otimes u = \mathrm{vec}(uv^\top)$.

## 3. Full-Spectrum GNNs

### 3.1. Spectral Filtering on Node-Pair Domain

Let $L = U\Lambda U^\top$ be the eigendecomposition of the Laplacian of a graph $G$. The Kronecker product $U \otimes U$ has columns $\{u_i \otimes u_j\}_{i,j=1}^n$, which form an orthonormal basis of $\mathbb{R}^{n^2}$. For a node-pair signal $\varepsilon \in \mathbb{R}^{n \times n}$, we will, by a slight abuse of notation, identify $\varepsilon$ with $\mathrm{vec}(\varepsilon) \in \mathbb{R}^{n^2}$ via the isomorphism $\mathrm{vec}$ *throughout the paper*. Then GFT and its inverse are defined as

$$\hat{\varepsilon} = (U \otimes U)^\top \varepsilon, \ \varepsilon = (U \otimes U)\hat{\varepsilon}.$$

Parallel to Definition 2.1, we have

**Definition 3.1.** Given a continuous function $g : \mathbb{R}^2 \to \mathbb{R}$, the corresponding *full-spectrum filter* is the matrix

$$G_\lambda := \left( g(\lambda_i, \lambda_j) \right)_{1 \le i,j \le n}$$

Then, the *full-spectrum convolution* is defined as

$$\begin{aligned} G_\lambda *_G \varepsilon &:= (U \otimes U)\left( G_\lambda \odot (U \otimes U)^\top \varepsilon \right) \\ &= (U \otimes U) G(\Lambda) (U \otimes U)^\top \varepsilon \\ &= \sum_{i,j} g(\lambda_i, \lambda_j) u_i u_i^\top \varepsilon u_j u_j^\top, \end{aligned}$$

where $G(\Lambda) := \mathrm{diag}\left( \mathrm{vec}(G_\lambda) \right) \in \mathbb{R}^{n^2 \times n^2}$.

Let $g$ be a bivariate polynomial $q(s, t) = \sum_{i,j} \alpha_{ij} s^i t^j$, then define the polynomial of Laplacians $q(L \otimes I_n, \ I_n \otimes L) := \sum_{i,j} \alpha_{ij}(L \otimes I_n)^i(I_n \otimes L)^j = \sum_{i,j} \alpha_{ij} L^i \otimes L^j$, where $I_n \in \mathbb{R}^{n \times n}$ is the identity matrix. We have

**Proposition 3.2.** *Define* $Q_\lambda = \left( q(\lambda_i, \lambda_j) \right)_{1 \le i,j \le n}$, *then*

$$q(L \otimes I_n, \ I_n \otimes L) = (U \otimes U) Q(\Lambda) (U \otimes U)^\top,$$

*where* $Q(\Lambda) = \mathrm{diag}\left( \mathrm{vec}(Q_\lambda) \right)$. *Equivalently,*

$$Q_\lambda *_G \varepsilon = q(L \otimes I_n, \ I_n \otimes L)\varepsilon.$$

The proof is provided in Appendix D (Proposition D.2). Motivated by Proposition 3.2, given an continuous function $g$, we denote the full spectrum convolutional matrix by

$$g(L \otimes I_n, I_n \otimes L) := (U \otimes U)G(\Lambda)(U \otimes U)^\top,$$

then for arbitrary such $g$,

$$G_\lambda *_G \varepsilon = g(L \otimes I_n, I_n \otimes L)\varepsilon.$$

Full-spectrum convolution admits a simple view as propagation on the Cartesian product graph $G \square G$ over node set $V \times V$, where $(u, v)$ is adjacent to $(u', v')$ iff $u = u'$ and $(v, v') \in E$, or $v = v'$ and $(u, u') \in E$. Concretely, $L \otimes I_n$ (resp. $I_n \otimes L$) propagates along the second (resp. first) coordinate while keeping the other fixed. As a special case, if we enforce simultaneous propagation along both coordinates by restricting the bivariate function to $g(s, t) = h(s + t)$ for some univariate $h$, the induced operator reduces to $g(L \oplus L)$, where $L \oplus L := L \otimes I_n + I_n \otimes L$, is the *Kronecker sum* of Laplacians. Moreover, $L \oplus L$ is exactly the Laplacian of the Cartesian product graph $G \square G$ (Brouwer & Haemers, 2011), so $g(L \oplus L)$ implements simultaneous propagation on both coordinates. In this sense, $g(L \oplus L)$ is a degraded second-order convolution: the signal is lifted to node pairs, but the filter is constrained to be univariate.

### 3.2. Classical Spectral Filtering as a Subfamily

Classical spectral filters can be characterized as a subfamily of full-spectrum filters via diagonal embedding.

**Proposition 3.3.** *Given node signal* $x \in \mathbb{R}^n$, *define the diagonal embedding* $\mathcal{E}(x) := \sum_{i=1}^n \hat{x}_i u_i u_i^\top$ *with* $\hat{x}_i := u_i^\top x$, *and the projection* $\mathcal{P}(H) := \sum_{i,j=1}^n (u_i^\top H u_j) u_i$ *for* $H \in \mathbb{R}^{n \times n}$. *Then*

$$\mathcal{P}(g(L \otimes I_n, \ I_n \otimes L) \mathcal{E}(x)) = U \tilde{g}(\Lambda) U^\top x,$$

*where* $\tilde{g}(\Lambda) := \mathrm{diag}(g(\lambda_i, \lambda_i))_{1 \le i \le n}$, *recovering the classical spectral filtering.*

The proof is provided in Appendix D (Proposition D.3). From this perspective, classical spectral filters can be viewed as *diagonal-spectrum* filters, obtained by restricting the full-spectrum filter to the diagonal of the paired spectrum.

## 3.3. Full-Spectrum GNN and Its Expressivity

. Based on the full-spectrum filtering, we propose the full-spectrum GNNs, i.e., FSPECGNNs. Given node features $X \in \mathbb{R}^{n \times d_1}$ and node-pair features $E \in \mathbb{R}^{n \times n \times d_2}$, we first lift them to representations in node-pair domain via $H_{uv} = \phi(X_u, X_v, E_{uv}) \in \mathbb{R}^d$, where $\phi$ can be either a hand-crafted encoder or a neural network. Stacking all pairwise representations yields a tensor $H \in \mathbb{R}^{n \times n \times d}$, which we reshape into a matrix $H \in \mathbb{R}^{n^2 \times d}$. A full-spectrum convolution layer is then defined as

$$H' = \sigma\big(g(L \otimes I_n, \, I_n \otimes L)\, H\, W\big),$$

where $g$ is a learnable bivariate function. Depending on the choice of readout, the filtered representations $H'$ can be used to produce edge/node/graph-level predictions.

Now We consider two complementary notions of expressivity for spectral GNNs. The first is specific to spectral methods and concerns their ability to universally approximate graph signals. The second is a model's ability to distinguish non-isomorphic graphs.

Parallel to Wang & Zhang (2022), who showed the linear spectral GNNs can universally approximate one-dimensional signals on $V$, we prove that the linear FSPECGNNs can universally approximate one-dimensional signals on $V \times V$, with proof provided in Appendix E (Theorem E.1).

**Theorem 3.4.** *Linear FSPECGNNs, obtained by removing the nonlinear activation, can produce any one-dimensional node-pair signal prediction if the normalized Laplacian $\tilde{L}$ has no multiple eigenvalues and the input representation $H$ contain all paired frequency components, namely, $(U \otimes U)^\top H$ is non-zero for every index $(u, v) \in V \times V$.*

We next study the ability of FSPECGNNs to distinguish non-isomorphic graphs. Let $G = (V, E)$ be a finite graph with an initial node labeling $\ell^{(0)} : V \rightarrow \Sigma$, where $\Sigma$ is a countable label set. We adopt the standard definitions of 1-WL and Local 2-GNN from Zhang et al. (2024).

**Definition 3.5.** The *1-dimensional Weisfeiler–Lehman test* refines node labels by

$$\ell^{(t)}(v) = \text{HASH}\Big(\ell^{(t-1)}(v), \{\!\!\{\ell^{(t-1)}(u) : u \in N(v)\}\!\!\}\Big),$$

where $N(v)$ denotes the neighborhood of $v$ and HASH is an perfect hash function. The process iterates until stabilization. Two graphs are *distinguished by 1-WL* if the multisets of node labels differ at some iteration.

**Definition 3.6.** The *local 2-GNN* operates on ordered node pairs $(u, v) \in V \times V$. It initializes $\ell^{(0)}(u, v) = \big(\ell^{(0)}(u), \ell^{(0)}(v), \mathbf{atp}(u, v)\big)$, where the *atomic type* is $\mathbf{atp}(u, v) = (\mathbf{1}_{u=v}, \mathbf{1}_{(u,v) \in E})$. For $t \geq 0$, define multisets $\mathcal{M}_{u \leftarrow v}^{(t)} := \{\!\!\{\ell^{(t)}(w, v) : w \in N(u)\}\!\!\}$ and $\mathcal{M}_{v \leftarrow u}^{(t)} :=$

$\{\!\!\{\ell^{(t)}(u, w) : w \in N(v)\}\!\!\}$, and update

$$\ell^{(t+1)}(u, v) = \text{HASH}\big(\ell^{(t)}(u, v), \mathcal{M}_{u \leftarrow v}^{(t)}, \mathcal{M}_{v \leftarrow u}^{(t)}\big).$$

The process iterates until stabilization.

First, consider FSPECGNNs that realizes $g(L \otimes I_n, \, I_n \otimes L)$ via bivariate polynomial $p$. By stacking the pairwise label $\ell^{(0)}(u, v)$ into a matrix $\mathcal{L} \in \mathbb{R}^{n^2 \times d}$ for some $d$, We show that the resulting models have expressivity upper-bounded by Local 2-GNNs.

**Theorem 3.7.** *Let $p$ be any polynomial of total degree at most $K$, and define the filtered linear representation $\mathcal{L}' = p(L \otimes I, I \otimes L)\mathcal{L}W$. Let $\chi_{(u,v)}^{(k)}$ denote the color of index $(u, v)$ after $k$ iterations of the Local 2-GNN refinement. Then for any $(u_1, v_1), (u_2, v_2) \in V \times V$ and any matrix $W$,*

$$\chi_{(u_1, v_1)}^{(K)} = \chi_{(u_2, v_2)}^{(K)} \implies \mathcal{L}'(u_1, v_1) = \mathcal{L}'(u_2, v_2).$$

In contrast, by allowing bivariate polynomials, there exists a polynomial $q(s, t)$ such that an FSPECGNN instantiated with $q$ is as expressive as Local 2-GNNs.

**Theorem 3.8.** *Let $\chi_{(u,v)}^{(\infty)}$ denote the stable Local 2-GNN color on index $(u, v)$. If $L$ has a simple spectrum and $U^\top \mathcal{L}U$ is index-wise nonzero, then there exists a bivariate polynomial $q$ and a matrix $W$, such that the filtered representation $\mathcal{L}' = q(L \otimes I_n, \, I_n \otimes L)\mathcal{L}W$ satisfies*

$$\mathcal{L}'(u_1, v_1) = \mathcal{L}'(u_2, v_2) \implies \chi_{(u_1, v_1)}^{(\infty)} = \chi_{(u_2, v_2)}^{(\infty)}.$$

Proofs are provided in Appendix E (Theorem E.3, E.6). Note that Theorem 3.8 is an existence result: in practice, the learned filter may or may not realize the specific polynomial $q$ guaranteed by the theorem. Consequently, the empirical expressivity of FSPECGNN can vary across tasks and may be slightly weaker or stronger than that of Local 2-GNNs, depending on the parameterization and optimization.

## 3.4. Scalability of FSPECGNN

A main challenge for second-order methods is scalability. Concretely, filtering in the node-pair domain is intractable, as it requires multiplying an $n^2 \times n^2$ matrix by an $n^2$-dimensional vector. In this section, we discuss implementations of FSPECGNNs, particularly scalable variants that avoid costly operations such as explicit eigendecomposition and node-pair-level computations. These implementations mainly involve parameterizations of bivariate function $g$.

**Route I: direct learning of $g(\lambda_i, \lambda_j)$.** When the graph is small, we can explicitly compute the eigendecomposition and learn the bivariate filter $g_\theta : \mathbb{R}^2 \rightarrow \mathbb{R}$ directly, e.g., using an MLP. In computing a full-spectrum convolution $g(L \otimes I_n, \, I_n \otimes L)H$, we follow Definition 3.1, constructing

$G_\lambda$ by $(G_\lambda)_{ij} = g_\theta(\lambda_i, \lambda_j)$, and apply $G_\lambda \odot ((U \otimes U^\top)\varepsilon)$ followed by inverse GFT transform, where $\varepsilon$ is one channel of $H$ and we apply the same convolution to all channels. This route is impractical for large graphs, as eigendecomposition requires $O(n^3)$ and GFT requires $O(n^2)$ time.

**Route II: polynomial parameterization of $g(\lambda_i, \lambda_j)$.** We parameterize the filter by a polynomial $q(L \otimes I_n, I_n \otimes L) = \sum_{p=0}^K \sum_{q=0}^K \alpha_{pq}(L^p \otimes L^q)$. Learning $g$ is then reduced to learning the coefficient matrix $(\alpha_{pq})$. A key computational benefit is that we never explicitly construct $L^p \otimes L^q \in \mathbb{R}^{n^2 \times n^2}$, instead, we exploit by Lemma 2.3 to have $(L^p \otimes L^q)\text{vec}(\varepsilon) = \text{vec}(L^q \varepsilon L^p)$, reducing computations to $n \times n$ matrix multiplications. Although we avoid computations in the node-pair domain, we still have $(K+1)^2$ polynomial terms; even with a recursive implementation, applying the filter still costs $O(K^2)$ left/right multiplications, which becomes expensive for high-degree polynomial filters.

**Further scalability via tensor decomposition.** To mitigate the at least quadratic dependence on $K$, we exploit the tensor-product structure of bivariate polynomial filters. Proofs are provided in Appendix D (Proposition D.4).

**Proposition 3.9.** *Let $P(s,t) = \sum_{i+j \le K} a_{ij} s^i t^j$ be a bivariate polynomial, and denote by $A = (a_{ij})$ its coefficient matrix. Then the induced operator $P(L \otimes I_n, I_n \otimes L)$ admits a decomposition of the form*

$$P(L \otimes I_n, I_n \otimes L) = \sum_{r=1}^R f_r(L) \otimes h_r(L)$$

*for univariate polynomials $\{f_r, h_r\}$ satisfying $\deg(f_r) \le K$ and $\deg(h_r) \le K$ for all $r$, if and only if $R \ge \text{rank}(A)$.*

In particular, a low-rank approximation of the coefficient matrix $A$ yields a truncated tensor decomposition with only a small number of terms:

$$\mathcal{T}_L^S := \sum_{r=1}^S f_r(L) \otimes h_r(L), \qquad S \ll \text{rank}(A). \quad (3)$$

Thus we only need to perform $O(SK)$ matrix multiplications with degree-$K$ univariate polynomials $f_r$ and $h_r$.

To bridge the theoretical formulation and practical implementations, Figure 2 summarizes the pipeline from full-spectrum filtering to its scalable realizations described in this section, together with the concrete neural network architectures introduced in Section 4.2 and Section 5.1.

# 4. Application to Heterophilic Graph Learning

## 4.1. Necessity of Off-diagonal Components

Classical spectral GNN restricts to diagonal spectral convolutions $g(L) = \sum_i g(\lambda_i) u_i u_i^\top$. Equivalently, for any

such convolution matrix $C = g(L)$, $U^\top C U$ is diagonal. In contrast, spatial message passing GNNs typically learn convolution operators whose spectral expansions over the basis $\{u_i u_j^\top\}$ contain non-zero off-diagonal components, i.e., $U^\top C U$ generally non-diagonal. We answer the following question in this section: are such off-diagonal components merely incidental artifacts of spatial message passing, or do they play a principled and necessary role in learning?

We study this question in heterophilic graph learning, where adjacent nodes often carry different labels and classical models such as GCNs can degrade substantially. To understand heterophily from a spectral perspective, we firstly characterize an optimal convolution operator that maximally separates representations across different labels, equivalently, maximally contracts representations within the same label, under suitable idealized assumptions.

**Model Assumption.** Let $G = (V, E)$ be a graph with $|V| = n$, whose nodes are partitioned into $k$ classes $\Pi = \{V_1, \ldots, V_k\}$ with $|V_a| = n_a$ and $\sum_{a=1}^k n_a = n$. For each class $a \in \{1, \ldots, k\}$, we randomly draw a class mean vector $m_a \in \mathbb{R}^d$ independently from the unit sphere $\mathbb{S}^{d-1}$.

Conditioned on these class means, node features are generated independently across nodes according to their class labels. Specifically, for each node $i \in V_{a_i}$, the feature vector $x_i \sim \mathcal{D}_{a_i}$, where each class-conditional distribution $\mathcal{D}_a$ on $\mathbb{R}^d$ has mean $m_a$ and covariance matrix $\Sigma_a \succeq 0$. We denote by $X \in \mathbb{R}^{n \times d}$ the resulting feature matrix, whose $i$-th row is $x_i^\top$. For convenience, we write $\tau_a := \text{tr}(\Sigma_a)$ for the total feature variance of class $a$, and assume $\tau_a > 0$ for all $a$.

Under this model, we now specify a measurement to quantify *class separability* after graph propagation. Rather than adopting a Bayesian formulation (Luan et al., 2023; Wang et al., 2024), we consider a deterministic classwise mean squared error

$$\mathcal{L}(C) := \sum_{a=1}^k \frac{1}{n_a} \sum_{p \in V_a} \mathbb{E} \left\| Y_p - m_a \right\|_2^2, \quad (4)$$

where $Y := CX \in \mathbb{R}^{n \times d}$ denotes the propagated feature matrix, $Y_p$ is its $p$-th row, and $m_a$ is the class-wise mean.

Eq. (4) is well suited to our setting: $\mathcal{L}(C)$ is differentiable with respect to $C$, and the optimal solution $C^* := \arg\min_C \mathcal{L}(C)$ admits a closed-form characterization.

**Theorem 4.1.** *Let $C^*$ be a minimizer of $\mathcal{L}(C)$. Assume $\tau_a n_a \ge \tau_0 > 0$ for each class $a \in \{1, \ldots, k\}$, where $\tau_0$ is a fixed positive constant lower bound. Then, as the feature dimension $d \to \infty$, $C^*$ becomes asymptotically block-diagonal with respect to the class partition $\Pi$, with*

$$(C^*)_{V_a \times V_b} \xrightarrow{\mathbb{P}} \begin{cases} s_a^* J_{V_a}, & a = b, \\ 0, & a \neq b, \end{cases} \qquad s_a^* \xrightarrow{\mathbb{P}} \frac{1}{n_a + \tau_a}.$$

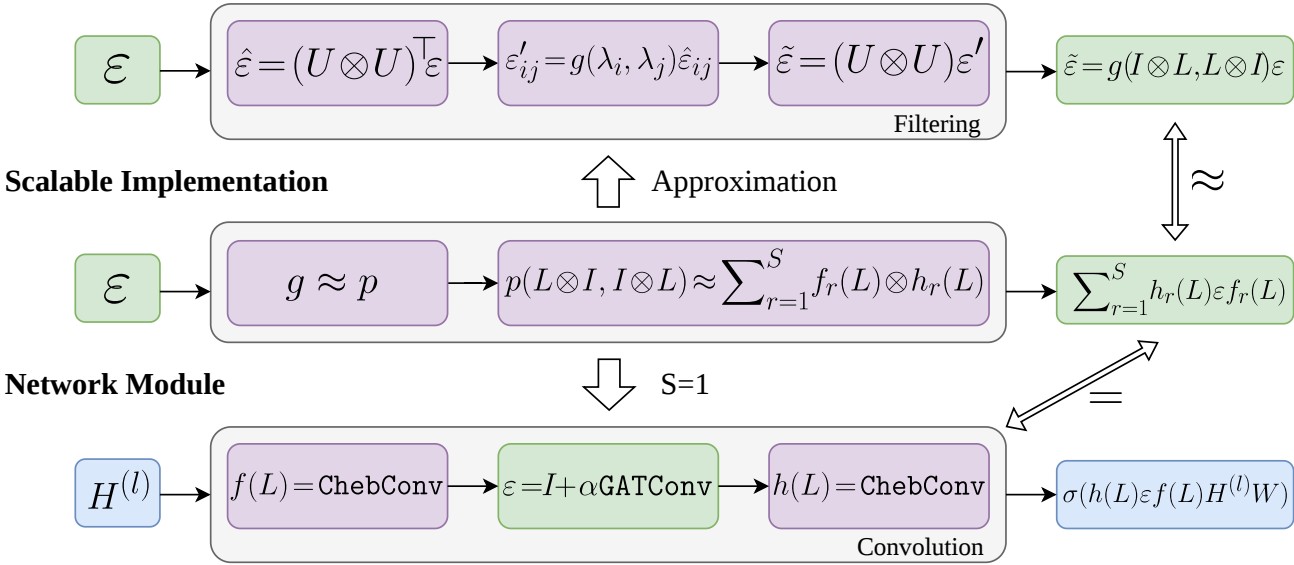

*Figure 2.* A high-level summary bridging the spectral formulation and its scalable implementation. Green shaded boxes indicate node-pair signals before or after spectral filtering; purple shaded boxes indicate filters; and blue shaded boxes indicate feature representations before or after convolution. Here, $g \approx p$ means that the bivariate spectral response $g$ is first parameterized by a bivariate polynomial $p$.

The proof is provided in Appendix F (Theorem F.9). It shows that the inter-class blocks of optimal convolution $C^\star$ vanish asymptotically, while intra-class blocks converge to class-dependent constants. We now prove that such convolutions **cannot** be realized by classical spectral filters.

**Theorem 4.2.** *Given a connected graph $G$, there exists a $G$-dependent integer $K \geq 1$ such that for any $K < k \leq n$, there exists a $k$-partition $V = V_1 \cup \cdots \cup V_k$ with label assignment $y_i = a \iff v_i \in V_a$ such that the following holds: if a diagonal-spectral operator $C = g(L)$ satisfies*

$$C_{ij} = 0 \quad \text{whenever } y_i \neq y_j,$$

*then necessarily $C = \alpha I_n$ for some $\alpha \in \mathbb{R}$.*

See Theorem F.12 for the proof. It shows that any spectral filter of the form $C = g(L)$ that enforces $C_{uv} = 0$ whenever $\ell(u) \neq \ell(v)$ must degenerate to $C = \alpha I$, and thus cannot approximate the optimal convolution in Theorem 4.1. Therefore, achieving the optimal class-separability convolution in general requires off-diagonal components, highlighting their importance for heterophilic graph learning.

The argument is supported by analysis of optimal convolutions across datasets with varying degrees of heterophily. Let $W_{ij} := (U^\top C U)^2_{ij}$. denote the spectral energy associated with the eigengraph component $u_i u_j^\top$. For a bandwidth parameter $\delta \in [0, 2]$, we define the *near-diagonal energy ratio* $E_C(\delta) := \sum_{|\lambda_i - \lambda_j| \leq \delta} W_{ij} / \sum_{i,j} W_{ij}$. Empirically, $E_{C^*}(\delta)$ of optimal convolution $C^*$ exhibits a clear trend: as the level of heterophily increases, spectral energy shifts

away from the diagonal, see Figure 3, indicating a growing reliance on off-diagonal components.

### 4.2. How FSPECGNN Models Off-diagonal

We provide a scalable implementation of FSPECGNN that models off-diagonal components without explicitly constructing node pair-domain operators.

We exploit the low-rank approximation $\mathcal{T}_L^S$ in Eq. (3), and take a small rank $S$, e.g., $S = 1$. Let $\mathcal{T}_L^1 = f(L) \otimes h(L)$, where $f$ and $h$ are univariate polynomial spectral filters. Given an initial convolution $\varepsilon \in \mathbb{R}^{n \times n}$, we have $f(L) \otimes h(L))\varepsilon = h(L)\,\varepsilon\,f(L)$ by Lemma 2.3. Thus a full-spectrum convolution layer can be implemented as

$$H' = \sigma\big(h(L)\,\varepsilon\,f(L)\,H\,W\big), \tag{5}$$

where $H \in \mathbb{R}^{n \times d}$ is a matrix of node representations. In practice, we can set both $f$ and $h$ to be classical filters such as `ChebConv` (Defferrard et al., 2016), `ChebIIConv` (He et al., 2022) or `BernConv` (He et al., 2021). The initial convolution can be either fixed or learnable, and we choose $\varepsilon = I + \alpha \text{GAT}$, where GAT is a single-layer GAT (Velickovic et al., 2018) and $\alpha$ is a learnable scalar.

Let $L = U\Lambda U^\top$. Expanding the linear convolution in Eq. (5) in the eigengraph basis gives

$$h(L)\,\varepsilon\,f(L) = \sum_{i,j} h(\lambda_i)\,f(\lambda_j)\,\big(u_i^\top \varepsilon u_j\big)\,u_i u_j^\top.$$

Therefore, an off-diagonal $u_i u_j^\top$ ($i \neq j$) appears whenever the coefficient $u_i^\top \varepsilon u_j$ is nonzero, see Corollary D.6 for more

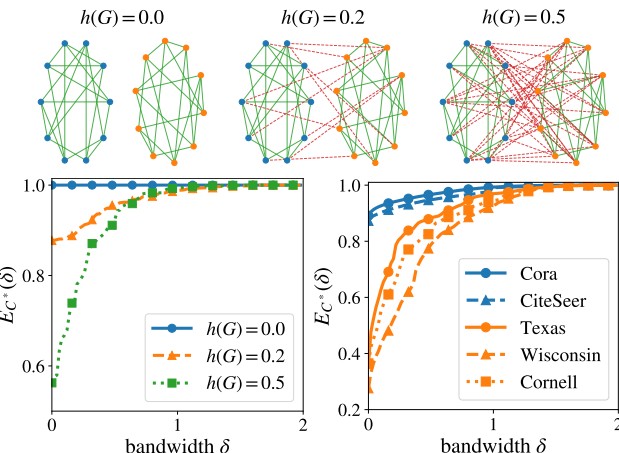

*Figure 3.* Off-diagonal components grow with heterophily. **Top:** Synthetic graphs with increasing heterophily $h(G) = \frac{1}{|E|} \sum_{(u,v) \in E} \mathbf{1}[\ell(u) \neq \ell(v)]$. **Bottom-left:** For each graph, we compute the optimal convolution and plot the near-diagonal energy ratio $E_{C^*}(\delta)$ as a function of the bandwidth $\delta$. Curves that are close to 1 already at $\delta = 0$ indicate that most spectral energy concentrates on the diagonal, whereas curves that start substantially below 1 and only approach 1 for larger $\delta$ indicate stronger off-diagonal energy. **Bottom-right:** The same trend holds on real-world datasets: Cora and Citeseer are homophilic and exhibit higher near-diagonal energy, while heterophilic datasets exhibit stronger off-diagonal energy.

explanations. GAT-based initialization fits this case, since the resulting $\varepsilon$ is feature-dependent and generally does not commute with the Laplacian $L$. In this view, GAT induces feature-dependent off-diagonal mixing, while $\mathcal{T}_L^S$ spectrally reweights both diagonal and off-diagonal components, tuning their balance via the frequency-pair response.

**Complexity.** Let $K_f$ and $K_h$ denote the polynomial degrees of $f$ and $h$. The linear convolution in Eq. (5) can be computed as $H_1 = f(L)H$, $H_2 = \varepsilon H_1$, and $H' = h(L)H_2$. Using sparse multiplications, the two polynomial spectral filters incur a cost of $O((K_f + K_h)md)$, while the single-layer GAT propagation costs $O(md)$ up to head-dependent constants. Consequently, the overall complexity is $O(Kmd)$ with $K \approx K_f + K_h$, matching the order of classical polynomial spectral layers such as `BernConv`. For a general rank $S$ in Eq. (3), the cost increases linearly in $S$.

## 5. Experiments

We benchmark FSpecGNNs on three representative tasks: node classification on heterophilic datasets, homomorphism counting, and cycle counting. For the counting tasks, our primary goal is not to push state-of-the-art performance; rather, we use them as controlled evaluations to validate the theoretical predictions derived in our analysis.

**Datasets.** For heterophilic node classification, we conduct experiments on eight widely used benchmark datasets, in-

cluding four small heterophilic graphs (Texas, Wisconsin, Chameleon, and Squirrel) and four large-scale heterophilic graphs (Roman Empire, Minesweeper, Tolokers, and Questions). Following Platonov et al. (2023), we use the directed clean versions of Chameleon and Squirrel, removing duplicate nodes. For all datasets, we adopt the sparse split strategy of Liu & Wang (2025), randomly partitioning nodes into training/validation/test sets with ratios of $2.5\%/2.5\%/95\%$.

For homomorphism counting, we adopt the benchmark of Zhao et al. (2022) and evaluate graph-level homomorphism counts as well as node/edge-anchored homomorphism count regression, using the same template graphs as Zhang et al. (2024). For (chordal) cycle counting, following Huang et al. (2023), we evaluate graph-level and node/edge-anchored (chordal) cycle subgraph count regression, again selecting the same subgraphs as Zhang et al. (2024).

**Baselines.** For heterophilic node classification, our baselines include five representative spectral GNNs: ChebNet, GPRGNN (Chien et al., 2021), BernNet, JacobiConv (Wang & Zhang, 2022), and ChebNetII, with results reported in Liu & Wang (2025). For homomorphism and cycle counting, we compare against MPNN, Subgraph GNN, Local 2-GNN, and Local 2-FGNN, adopting the results reported by Zhang et al. (2024), as well as Spectral Invariant GNN with results from Gai et al. (2025).

### 5.1. Heterophilic Node Classification

We adopt FSpecGNN in form of Eq. (5) with detailed implementation described in Section 4.2, yielding three variants denoted by FSpecGNN(Cheb/ChebII/Bern), respectively. The results are reported in the last three rows of Table 1. ROC-AUC is reported on Minesweeper, Tolokers, and Questions, while test accuracy is reported on the remaining datasets. Across all datasets, each variant consistently improves upon its corresponding base filter (e.g., FSpecGNN(Cheb) vs. ChebNet, and similarly for ChebII and Bern). Moreover, for every dataset, the best-performing FSpecGNN variant surpasses all preceding spectral baselines in Table 1.

### 5.2. GNN Expressivity

We adopt Route I in Section 3.4 and parameterize the spectral response $g$ using a two-layer MLP. Since a single spectral layer can aggregate global information, we do not follow the five propagation layers used in Zhang et al. (2024); instead, we use fewer layers with a larger hidden dimension, resulting in a comparable model size (approximately 300K parameters) to the baselines. Apart from the above modifications, we keep all remaining hyperparameters and setups identical to Zhang et al. (2024).

The reported performance is measured by normalized mean

*Table 1.* Node classification tasks on heterophilic datasets. High accuracy indicates good performance.

| Model | Texas | Wisconsin | Chameleon | Squirrel | Roman | Minesweeper | Tolokers | Questions |
|---|---|---|---|---|---|---|---|---|
| ChebNet | $36.76_{\pm6.76}$ | $33.17_{\pm7.83}$ | $25.65_{\pm3.56}$ | $22.58_{\pm2.56}$ | $45.32_{\pm0.65}$ | $86.34_{\pm0.19}$ | $69.88_{\pm0.26}$ | $48.55_{\pm1.04}$ |
| ChebNetII | $48.44_{\pm10.87}$ | $41.33_{\pm9.25}$ | $33.48_{\pm4.59}$ | $30.80_{\pm2.65}$ | $55.06_{\pm0.32}$ | $74.49_{\pm3.76}$ | $69.37_{\pm0.60}$ | $63.99_{\pm0.32}$ |
| GPRGNN | $48.55_{\pm7.00}$ | $40.79_{\pm3.62}$ | $30.44_{\pm4.29}$ | $24.33_{\pm2.68}$ | $55.48_{\pm1.30}$ | $86.68_{\pm0.32}$ | $67.05_{\pm0.97}$ | $53.76_{\pm0.41}$ |
| JacobiConv | $50.17_{\pm7.87}$ | $46.08_{\pm9.08}$ | $26.57_{\pm3.60}$ | $23.15_{\pm4.47}$ | $52.92_{\pm0.70}$ | $\underline{87.40}_{\pm0.13}$ | $70.24_{\pm0.18}$ | $55.68_{\pm0.54}$ |
| BernNet | $52.14_{\pm8.09}$ | $49.33_{\pm8.21}$ | $29.45_{\pm3.22}$ | $25.94_{\pm3.35}$ | $55.30_{\pm0.41}$ | $76.64_{\pm0.29}$ | $69.31_{\pm0.69}$ | $65.41_{\pm0.33}$ |
| **FSPECGNN(Cheb)** | $55.32_{\pm4.57}$ | $49.87_{\pm8.29}$ | $33.09_{\pm0.92}$ | $\mathbf{39.57}_{\pm0.67}$ | $54.35_{\pm0.73}$ | $\mathbf{88.30}_{\pm0.82}$ | $\mathbf{76.89}_{\pm0.91}$ | $75.87_{\pm0.37}$ |
| **FSPECGNN(ChebII)** | $\mathbf{57.05}_{\pm2.20}$ | $\underline{50.00}_{\pm5.42}$ | $\mathbf{39.60}_{\pm2.77}$ | $37.70_{\pm0.63}$ | $\mathbf{56.26}_{\pm1.08}$ | $84.25_{\pm0.65}$ | $\underline{76.37}_{\pm0.67}$ | $\underline{77.00}_{\pm0.37}$ |
| **FSPECGNN(Bern)** | $\underline{56.13}_{\pm0.46}$ | $\mathbf{54.58}_{\pm9.47}$ | $\underline{37.91}_{\pm3.94}$ | $37.59_{\pm1.32}$ | $56.16_{\pm1.19}$ | $84.17_{\pm1.01}$ | $74.50_{\pm0.69}$ | $\mathbf{77.11}_{\pm0.26}$ |

*Table 2.* Experimental results on (chordal) cycle counting, reported as normalized test MAE. Lower is better.

| Task / Model | Graph-level | | | | | | Node-level | | | | | | Edge-level | | | | | |
|---|---|---|---|---|---|---|---|---|---|---|---|---|---|---|---|---|---|---|
| MPNN | .358 | .208 | .188 | .146 | .261 | .205 | .600 | .413 | .300 | .207 | .318 | .237 | - | - | - | - | - | - |
| Spec. Inv. GNN | .072 | .072 | .089 | .089 | .060 | .099 | - | - | - | - | - | - | - | - | - | - | - | - |
| Subgraph GNN | .010 | .020 | .024 | .046 | .007 | .027 | .003 | .005 | .092 | .082 | .050 | .073 | .001 | .003 | .090 | .096 | .038 | .065 |
| Local 2-GNN | .008 | .011 | .017 | .034 | .007 | .016 | .002 | .005 | .010 | .023 | .004 | .015 | .001 | .005 | .010 | .019 | .005 | .014 |
| Local 2-FGNN | .003 | .004 | .010 | .020 | .003 | .010 | .004 | .006 | .012 | .021 | .004 | .014 | .003 | .006 | .012 | .022 | .005 | .012 |
| FSPECGNN | .005 | .018 | .024 | .033 | .006 | .032 | .003 | .009 | .020 | .029 | .009 | .021 | .003 | .008 | .026 | .045 | .009 | .033 |

*Table 3.* Experimental results on homomorphism counting. Red/blue nodes indicate marked vertices. Lower is better.

| Task / Model | Graph-level | | | Node-level | | Edge-level | | |
|---|---|---|---|---|---|---|---|---|
| MPNN | .300 | .233 | .254 | .505 | .478 | – | – | – |
| Spec. Inv. GNN | .045 | .048 | – | – | – | – | – | – |
| Subgraph GNN | .011 | .015 | .012 | .004 | .058 | .003 | .058 | .048 |
| Local 2-GNN | .008 | .008 | .010 | .003 | .004 | .005 | .006 | .008 |
| Local 2-FGNN | .003 | .005 | .004 | .005 | .005 | .007 | .007 | .008 |
| FSPECGNN | .006 | .009 | .008 | .006 | .008 | .008 | .008 | .012 |

*Table 4.* Ablation on architectural components. $\Delta$ indicates change from the full model.

| Variants | Texas | Squirrel | Roman |
|---|---|---|---|
| Full model | $55.3_{\pm4.6}$ | $39.6_{\pm0.7}$ | $54.4_{\pm0.7}$ |
| w/o Off-diagonal $\Delta$ | $51.0_{\pm7.1}$ $-4.3$ | $34.6_{\pm0.4}$ $-5.0$ | $53.8_{\pm1.8}$ $-0.6$ |
| w/o In-filter $f$ $\Delta$ | $49.5_{\pm11.0}$ $-5.8$ | $29.9_{\pm3.1}$ $-9.7$ | $52.0_{\pm1.5}$ $-2.4$ |
| w/o $f$ and off-diagonal $\Delta$ | $39.7_{\pm7.9}$ $-15.6$ | $24.5_{\pm1.1}$ $-15.1$ | $47.1_{\pm0.8}$ $-7.3$ |

absolute error (MAE). Results are summarized in Tables 2 and 3. In Table 2, cell colors indicate MAE ranges: blue for $< 0.025$, green for $[0.025, 0.05)$, orange for $[0.05, 0.1)$, and red for $\geq 0.1$. Overall, the empirical results align with our theoretical predictions. In particular, Theorems 3.7 and 3.8 suggest that FSPECGNNs should have expressivity comparable to Local 2-GNNs when equipped with sufficiently expressive spectral filters. This argument is supported by experiments: on all benchmarks, FSPECGNN achieves performance comparable to Local 2-GNN, and even outperforms it on some datasets. Moreover, compared to Spectral Invariant GNN, which also leverages graph spectral information, FSPECGNN achieves substantially lower MAE on these benchmarks.

### 5.3. Ablation Study

Table 4 studies the contribution of architectural components in FSPECGNN(Cheb) on heterophilic node classification. The full model uses two spectral filters: the in-filter $f$, applied to the input features before eigenspace interaction,

and the out-filter $h$, applied after interaction to refine the propagated representation. Removing off-diagonal interactions consistently degrades performance across datasets, underscoring the necessity of modeling cross-eigenspace coupling. Ablating both the in-filter $f$ and the off-diagonal interaction reduces the model to CHEBNET, yielding performance consistent with the baseline.

### 5.4. Time and Space Overhead

We first evaluate the computational efficiency of models on the heterophilic node classification benchmarks in Table 5 and Table 6. For each model, we report the average runtime per run, averaged over 10 runs, together with the peak GPU memory usage for each task. We observe that all variants of FSPECGNN are substantially faster and more memory-efficient than Local 2-GNN, while maintaining runtime and GPU memory usage comparable to classical spectral GNNs.

We then evaluate the computational efficiency of models

*Table 5.* Average runtime per run in seconds on heterophilic datasets. "–" indicates out-of-memory.

| Model | Texas | Wisconsin | Chameleon | Squirrel | Roman | Minesweeper | Tolokers | Questions |
|---|---|---|---|---|---|---|---|---|
| Local 2-GNN | 23.20 | 51.24 | – | – | – | – | – | – |
| ChebNet | 0.94 | 0.97 | 1.51 | 1.93 | 1.66 | 1.90 | 2.32 | 2.45 |
| **FSPECGNN(Cheb)** | 2.04 | 2.10 | 2.55 | 2.51 | 3.88 | 3.83 | 3.96 | 3.40 |
| ChebNetII | 2.07 | 1.90 | 2.55 | 2.73 | 2.96 | 3.07 | 3.63 | 3.49 |
| **FSPECGNN(ChebII)** | 2.41 | 2.78 | 3.17 | 3.27 | 4.31 | 4.34 | 4.80 | 5.16 |
| BernNet | 2.21 | 2.15 | 2.21 | 2.24 | 2.53 | 3.25 | 3.85 | 4.13 |
| **FSPECGNN(Bern)** | 2.17 | 2.48 | 2.69 | 3.35 | 3.05 | 3.52 | 4.66 | 6.01 |

*Table 6.* GPU memory usage (MiB) on heterophilic datasets. "–" indicates out-of-memory.

| Model | Texas | Wisconsin | Chameleon | Squirrel | Roman | Minesweeper | Tolokers | Questions |
|---|---|---|---|---|---|---|---|---|
| Local 2-GNN | 1830 | 3128 | – | – | – | – | – | – |
| ChebNet | 534 | 534 | 558 | 594 | 864 | 654 | 742 | 1150 |
| **FSPECGNN(Cheb)** | 646 | 648 | 692 | 760 | 1130 | 754 | 976 | 1242 |
| ChebNetII | 646 | 648 | 660 | 716 | 954 | 750 | 914 | 1230 |
| **FSPECGNN(ChebII)** | 650 | 658 | 698 | 752 | 1062 | 756 | 998 | 1244 |
| BernNet | 648 | 648 | 658 | 698 | 1058 | 756 | 954 | 1254 |
| **FSPECGNN(Bern)** | 652 | 658 | 694 | 758 | 1102 | 774 | 1040 | 1276 |

*Table 7.* Average iteration time and peak GPU memory per model across all homomorphism- and cycle-counting tasks.

| Model | Avg. Time (s/it) | Max GPU Mem (MB) |
|---|---|---|
| MPNN | 6.162 | 9688 |
| Subgraph NN | 6.145 | 11228 |
| Local 2-GNN | 7.569 | 11532 |
| Local 2-FGNN | 4.856 | 19886 |
| FSPECGNN | 1.111 | 9532 |

on the homomorphism- and cycle-counting benchmarks in Table 7. For each model, we report the average time per iteration, aggregated over graph/node/edge levels across both homomorphism- and cycle-counting tasks, and the peak GPU memory usage, taken as the maximum over all tasks. Notably, MPNN is realized on the lifted node-pair domain representation and reduces to MPNN only through a degenerate message passing, so it retains second-order computational overhead.

Further details, including overhead analysis for additional models and per-task evaluations, are provided in Appendix G.

## 6. Related Works

Several recent GNN architectures are closely related, yet they differ from our work in perspectives of the signal domain, the spectral filter, and the analysis focus. Spec-

former (Bo et al., 2023) learns a set-to-set map on the graph spectrum, but the resulting operator remains *diagonal* in the eigenbasis; our framework instead parameterizes $g(\lambda_i, \lambda_j)$ to enable *off-diagonal* spectral interactions. CITRUS (Einizade et al., 2024) leverages Cartesian product graphs and continuous diffusion for scalable learning on multi-domain data; our focus instead stays within a single graph and its internal spectral interactions. STSGNN (Chen et al., 2025) designs second-order spectral filters for spatio-temporal modeling by using two Laplacians from the spatial and temporal domains; in contrast, we study second-order spectral filters based on a single graph and analyze their expressivity and scalability. Finally, Spectral Invariant GNNs (Gai et al., 2025) inject spectral invariants into GNNs and study homomorphism expressivity, using spectral information in a positional-encoding-like manner rather than parameterizing a spectral filter. More related works are discussed in Appendix B.

## 7. Conclusion

We propose FSPECGNNs as a second-order generalization of spectral GNNs. It can surpass the expressivity limits of 1-WL, while admitting scalable implementations that extend to large graphs. Overall, as second-order spectral GNNs, FSPECGNNs are both expressive and scalable, making it well suited to tasks with inherently second-order structure.

## Acknowledgements

This work was supported in part by the Singapore Ministry of Education Academic Research fund Tier 1 grant RG16/23, Tier 2 grants MOE-T2EP20125-0004 and MOE-T2EP50223-0036.

The authors gratefully acknowledge helpful discussions with Xinqi Gong, Fei Han, Jianzhao Gao, and Yifei Yang.

## Impact Statement

This paper presents work whose goal is to advance the field of machine learning, specifically in spectral graph neural networks. Our contributions are methodological and are intended for general-purpose use. We do not anticipate substantial ethical concerns or negative societal consequences beyond those commonly associated with machine learning research.

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

# A. Arrangement of the Appendix

The appendix is organized as follows.

- Section B: More Related Work. A more detailed discussion of related work.

- Section C: Discussions and Potential Limitations. Possible extensions of this work and its current limitations.

- Section D: Algebraic Properties and Parameterizations of Full-Spectrum Filters. Includes proofs for Section 3, except for Section 3.3.

- Section E: Expressivity of Full-Spectrum GNNs. Includes proofs for Section 3.3.

- Section F: An Analysis of Spectral Filters for ClassWise Separability. Includes proofs for Section 4.

- Section G: Experimental Details.

# B. More Related Work

## B.1. Spectral GNN Architectures

Spectral GNNs implement graph convolution as a Laplacian filter $g(L)$, but exact eigendecomposition is typically avoided via scalable approximations. Defferrard et al. (2016) introduced truncated Chebyshev polynomial parameterizations to realize localized spectral filtering, and GCN (Kipf & Welling, 2017) can be viewed as a first-order simplification that yields an efficient, widely used propagation rule. Subsequent work increasingly emphasizes stable and flexible spectral-response parameterizations beyond vanilla Chebyshev expansions: ChebNetII (He et al., 2022) attributes ChebNet's weaker performance to poorly behaved learned coefficients, while BernNet (He et al., 2021) learns arbitrary spectral filters via Bernstein approximation, enabling controlled realization of diverse frequency responses. In parallel, diffusion-based formulations connect propagation to PageRank: PPNP/APPNP (Gasteiger et al., 2019) derive personalized-PageRank propagation decoupled from prediction, and GPR-GNN (Chien et al., 2021) generalizes this view by learning generalized PageRank weights to adapt across homophily/heterophily regimes. Finally, recent research finds that the choice of polynomial basis materially affects optimization and practical performance, motivating a sequence of methods that select or learn improved polynomial bases for spectral filtering (Wang & Zhang, 2022; Guo & Wei, 2023). Moreover, spectral methods have been explored for heterophily by moving beyond purely low-pass smoothing: either by injecting learnable high-pass responses or by combining multiple spectral filters to adapt across homophily/heterophily regimes (Bo et al., 2021; Yan et al., 2023). Recent studies also highlight the role of polynomial bases under heterophily, motivating heterophily-aware and universal basis construction beyond predefined Laplacian bases (Huang et al., 2024).

## B.2. Expressivity of Spectral GNNs

Expressivity has been a central theme in the GNN literature, and is most commonly studied through the WL-based hierarchy. Xu et al. (2019) shows that standard neighborhood aggregation is at most as powerful as 1-WL for graph discrimination, which catalyzed extensive WL-based analyses and the development of higher-order GNNs matching stronger WL variants (Morris et al., 2019; Maron et al., 2019; Morris et al., 2020). Going beyond this coarse WL lens, (Zhang et al., 2024) propose homomorphism expressivity as a finer quantitative hierarchy for comparing substructure-counting capabilities. In contrast, theoretical understanding of spectral GNN expressivity is relatively limited: Wang & Zhang (2022) establish universality results for spectral filtering on node signals, while also relating spectral models to 1-WL limitations. More recently, (Roth & Liebig, 2025) extend the universality analysis to multi-channel (MIMO) settings. Gai et al. (2025) study spectral-invariant architectures that inject spectral information as basis-invariant positional encodings; their expressivity has been characterized via WL- and homomorphism-based analysis.

# C. Discussions and Potential Limitations

## C.1. Discussions on FSPECGNN

**Extending Full-Spectrum Filtering to Higher Orders.** Although we instantiate the framework at second order, i.e. the node-pair domain, the same construction extends canonically to $k$-th order signals on $V^k$. Let $L$ be a chosen graph Laplacian

on $G$ and let $I_n$ be the $n \times n$ identity. On the $k$-fold product domain, define the commuting operators

$$L^{(1)} := L \otimes I_n^{\otimes(k-1)}, \quad L^{(2)} := I_n \otimes L \otimes I_n^{\otimes(k-2)}, \ldots, L^{(k)} := I_n^{\otimes(k-1)} \otimes L.$$

A $k$-variate spectral filter can then be parameterized as $g\big(L^{(1)}, \ldots, L^{(k)}\big)$. For example, the third-order case acts on $V^3$ via

$$g(L \otimes I_n \otimes I_n, \ I_n \otimes L \otimes I_n, \ I_n \otimes I_n \otimes L),$$

which propagates along each coordinate while keeping the others fixed. This higher-order viewpoint provides a systematic route to designing spectral GNNs that operate on $k$-tuple domains, and enables a systematic connection between higher-order spectral constructions and the expressiveness hierarchy of GNNs, e.g. (Zhang et al., 2024).

**Beyond Graph Laplacians.** A further direction is to move beyond graph Laplacians to Laplacians defined on higher-dimensional combinatorial structures, such as simplicial or cell complexes. These domains admit Laplacian operators, e.g. Hodge Laplacians, acting on $p$-cochains, including edge/face-signals, whose spectra encode topology/geometry-related modes that are invisible to purely node-level models. From this perspective, our use of the Cartesian product provides one concrete instance of how additional structure induces a corresponding notion of propagation: full-spectrum filtering can be interpreted as propagation along different coordinates of a product graph. For richer higher-order structures, the induced propagation is generally governed not by the graph Laplacian alone, but by the Laplacians intrinsic to the underlying domain, suggesting that analogous spectral formalisms could be developed to capture these more complex propagation patterns. For example, cell complex neural networks generalize message passing via boundary/coboundary operators, inducing propagation patterns determined by higher-order incidence relations (Hajij et al., 2020), and BScNets further exploit block Hodge Laplacians to couple interactions across different dimensions, enabling information mixing between cells of different orders (Chen et al., 2022). Developing spectral analyses and learning methods for these Laplacians remains under-explored and is a natural extension of our perspective.

### C.2. Potential Limitations and Future Work

**Universal Approximation on Multi-channel Node-pair Signals** Our discussion on universal approximation focuses on 1-dimensional node-pair signals. In practice, approximation capacity can be strengthened by adopting a MIMO design similar to Roth & Liebig (2025), i.e., learning multiple input/output channels with channel mixing, which is standard in spectral GNNs and can be incorporated into our framework directly.

**Underexplored implementation choices for FSPECGNN.** While our framework is general, our empirical instantiation explores only a limited portion of the design space. First, for heterophily node classification we adopt a particular initial convolution in the form of the pair-domain signal, $\varepsilon = I + \alpha \, \mathrm{GAT}$, which is a reasonable but by no means canonical choice; alternative initial convolutions may yield better performance. Second, for scalability we mainly use the low-rank parameterization with $S = 1$, i.e., a single filter pair $(f, h)$. The general form allows $S > 1$ with multiple filter pairs, raising additional questions on how to compose and coordinate these components effectively. A systematic exploration of $\varepsilon$ and multi-component designs is left to future work.

## D. Algebraic Properties and Parameterizations of Full-Spectrum Filters

### D.1. Basic Properties

**Definition D.1.** Let $G = (V, E)$ be an undirected graph. The *Cartesian product graph* $G \square G$ is the graph with vertex set $V \times V$, where two vertices $(u, v)$ and $(u', v')$ are adjacent if and only if either $u = u'$ and $(v, v') \in E$, or $v = v'$ and $(u, u') \in E$.

Under the above convention, the action of $L \otimes I_n$ and $I_n \otimes L$ can be described explicitly as follows: for $(u, v) \in V \times V$,

$$\big((L \otimes I_n)\varepsilon\big)(u, v) = (\varepsilon L)(u, v) = \sum_{v' \in V} \varepsilon(u, v') \, L_{v', v},$$

$$\big((I_n \otimes L)\varepsilon\big)(u, v) = (L\varepsilon)(u, v) = \sum_{u' \in V} L_{u, u'} \, \varepsilon(u', v).$$

That is, right (resp. left) multiplication by $L$ propagates the signal by changing the second (resp. first) coordinate of $G \square G$ while keeping the other fixed, see the first row of Figure 1 for an illustration.

We restate and prove Proposition 3.2 as follows.

**Proposition D.2.** *Let $L \in \mathbb{R}^{n \times n}$ be symmetric with eigendecomposition $L = U\Lambda U^\top$, where $U = [u_1, \ldots, u_n]$ is orthogonal and $\Lambda = \mathrm{diag}(\lambda_1, \ldots, \lambda_n)$. Let $q : \mathbb{R} \times \mathbb{R} \to \mathbb{R}$ be a bivariate polynomial*

$$q(s, t) = \sum_{a,b \geq 0} \alpha_{ab} s^a t^b.$$

*Define*

$$q(L \otimes I_n, \ I_n \otimes L) := \sum_{a,b \geq 0} \alpha_{ab} (L \otimes I_n)^a (I_n \otimes L)^b.$$

*Then*

$$q(L \otimes I_n, \ I_n \otimes L) = (U \otimes U) \, \mathrm{diag}\!\Big( q(\lambda_i, \lambda_j) \Big)_{1 \leq i,j \leq n} (U \otimes U)^\top,$$

*i.e., the operator is diagonal in the basis $\{u_i \otimes u_j\}_{i,j=1}^n$, with eigenvalues $q(\lambda_i, \lambda_j)$.*

*Proof.* Since $L = U\Lambda U^\top$, we have $L^a = U\Lambda^a U^\top$ for any integer $a \geq 0$. Using the mixed-product property of the Kronecker product,

$$(A \otimes B)(C \otimes D) = (AC) \otimes (BD),$$

we obtain

$$(L \otimes I_n)^a = L^a \otimes I_n, \qquad (I_n \otimes L)^b = I_n \otimes L^b,$$

and hence

$$(L \otimes I_n)^a (I_n \otimes L)^b = (L^a \otimes I_n)(I_n \otimes L^b) = L^a \otimes L^b.$$

Therefore,

$$q(L \otimes I_n, \ I_n \otimes L) = \sum_{a,b \geq 0} \alpha_{ab} \, L^a \otimes L^b.$$

Substituting $L^a = U\Lambda^a U^\top$ and $L^b = U\Lambda^b U^\top$ gives

$$q(L \otimes I_n, \ I_n \otimes L) = \sum_{a,b \geq 0} \alpha_{ab} \, (U\Lambda^a U^\top) \otimes (U\Lambda^b U^\top)$$

$$= (U \otimes U)\Big( \sum_{a,b \geq 0} \alpha_{ab} \, \Lambda^a \otimes \Lambda^b \Big)(U^\top \otimes U^\top).$$

It remains to identify the middle term. Since $\Lambda^a$ and $\Lambda^b$ are diagonal, $\Lambda^a \otimes \Lambda^b$ is diagonal with diagonal entries $\lambda_i^a \lambda_j^b$ indexed by pairs $(i, j)$ such that $1 \leq i, j \leq n$. Hence

$$\sum_{a,b \geq 0} \alpha_{ab} \, \Lambda^a \otimes \Lambda^b = \mathrm{diag}\!\Big( \sum_{a,b \geq 0} \alpha_{ab} \lambda_i^a \lambda_j^b \Big)_{1 \leq i,j \leq n} = \mathrm{diag}\!\Big( q(\lambda_i, \lambda_j) \Big)_{1 \leq i,j \leq n}.$$

Finally, $U^\top \otimes U^\top = (U \otimes U)^\top$ because $U$ is orthogonal, which yields

$$q(L \otimes I_n, \ I_n \otimes L) = (U \otimes U) \, \mathrm{diag}\!\Big( q(\lambda_i, \lambda_j) \Big)_{1 \leq i,j \leq n} (U \otimes U)^\top.$$

$\square$

We restate and prove Proposition 3.3 as follows.

**Proposition D.3.** *Define the diagonal embedding $\mathcal{E} : \mathbb{R}^n \to \mathbb{R}^{n \times n}$ and the projection $\mathcal{P} : \mathbb{R}^{n \times n} \to \mathbb{R}^n$ by*

$$\mathcal{E}(x) := \sum_{i=1}^n \hat{x}_i \, u_i u_i^\top, \qquad \hat{x}_i := u_i^\top x,$$

*and*

$$\mathcal{P}(H) \; := \; \sum_{i,j=1}^{n} (u_i^\top H u_j) \, u_i.$$

*Then*

$$\mathcal{P}(g(L \otimes I_n, \; I_n \otimes L) \, \mathcal{E}(x)) \; = \; U \operatorname{diag}\big(g(\lambda_1, \lambda_1), \ldots, g(\lambda_n, \lambda_n)\big) U^\top x.$$

*Equivalently, if we define a univariate spectral response $\tilde{g}(\lambda) := g(\lambda, \lambda)$, then*

$$\mathcal{P}(g(L \otimes I_n, \; I_n \otimes L) \, \mathcal{E}(x)) \; = \; U \, \tilde{g}(\Lambda) \, U^\top x,$$

*which recovers the classical spectral filtering with response $\tilde{g}$.*

*Proof.* We first record two elementary facts: (i) By definition, $\mathcal{E}(x) \in \operatorname{span}\{u_i u_i^\top\}_{i=1}^n$; (ii) $L(u_i u_i^\top) = (L u_i) u_i^\top = \lambda_i \, u_i u_i^\top$, and $(u_i u_i^\top) L = u_i (u_i^\top L) = \lambda_i \, u_i u_i^\top$ as $L u_i = \lambda_i u_i$ and $L$ is symmetric. Therefore we have

$$g(L \otimes I_n, \; I_n \otimes L) \, (u_i u_i^\top) \; = \; g(\lambda_i, \lambda_i) \, u_i u_i^\top.$$

Linearity then gives

$$g(L \otimes I_n, \; I_n \otimes L) \, \mathcal{E}(x) = \sum_{i=1}^{n} g(\lambda_i, \lambda_i) \, \hat{x}_i \, u_i u_i^\top.$$

Finally, we apply $\mathcal{P}$ and get

$$\mathcal{P}\left( \sum_{i=1}^{n} g(\lambda_i, \lambda_i) \, \hat{x}_i \, u_i u_i^\top \right) = \sum_{j,k=1}^{n} \left( u_j^\top \left( \sum_{i=1}^{n} g(\lambda_i, \lambda_i) \, \hat{x}_i \, u_i u_i^\top \right) u_k \right) u_j.$$

Using orthonormality, the coefficient of $u_j$ is exactly $g(\lambda_j, \lambda_j) \hat{x}_j$. Thus

$$\mathcal{P}(g(L \otimes I_n, \; I_n \otimes L) \, \mathcal{E}(x)) = U \operatorname{diag}(g(\lambda_1, \lambda_1), \ldots, g(\lambda_n, \lambda_n)) U^\top x,$$

which proves the claim. $\qquad\square$

### D.2. Tensor Decomposition of Full-Spectrum Filters.

Note that $(L \otimes I_n)^i (I_n \otimes L)^j = (I_n \otimes L)^j (L \otimes I_n)^i = L^i \otimes L^j$. We restate and prove Proposition 3.9 as follows.

**Proposition D.4.** *Let $P(I_n \otimes L, \; L \otimes I_n)$ be a bivariate polynomial of total degree $K$ in the two matrix variables $I_n \otimes L$ and $L \otimes I_n$. We write*

$$P(I_n \otimes L, \; L \otimes I_n) = \sum_{\substack{0 \le i,j \le K \\ i+j \le K}} c_{ij} \, (I_n \otimes L)^i (L \otimes I_n)^j = \sum_{\substack{0 \le i,j \le K \\ i+j \le K}} c_{ij} \, L^i \otimes L^j,$$

*where the coefficients are collected into a $(K+1) \times (K+1)$ matrix $A = (c_{ij})_{0 \le i,j \le K}$. Then there exist polynomial pairs $\{(f_r, h_r)\}_{r=1}^R$ with $\deg f_r \le K$ and $\deg h_r \le K$ for all $r$, such that*

$$P(I_n \otimes L, \; L \otimes I_n) \; = \; \sum_{r=1}^{R} f_r(L) \otimes h_r(L),$$

*if and only if $R \ge \operatorname{rank}(A)$.*

**Lemma D.5.** *Let $A \in \mathbb{R}^{n \times n}$ be a square matrix. Then*

1. *If $\operatorname{rank}(A) = R$, there exist vectors $\alpha_1, \ldots, \alpha_R \in \mathbb{R}^n$ and $\beta_1, \ldots, \beta_R \in \mathbb{R}^n$ such that*

$$A \; = \; \sum_{i=1}^{R} \alpha_i \beta_i^\top.$$

2. *Conversely, if A can be written as*

$$A = \sum_{i=1}^{R} \alpha_i \beta_i^\top,$$

*then* $\operatorname{rank}(A) \leq R$.

*In particular, the minimal number of rank-one matrices $\{\alpha_i \beta_i^\top\}$ required in such a decomposition equals* $\operatorname{rank}(A)$.

*Proof.* (1) Since $\operatorname{rank}(A) = R$, the column space of $A$ has dimension $R$. Choose a basis $c_1, \ldots, c_R \in \mathbb{R}^n$ for the column space and set $C = [\, c_1 \ \cdots \ c_R \,] \in \mathbb{R}^{n \times R}$. Then there exists $D \in \mathbb{R}^{R \times n}$ such that $A = CD$. Writing $D$ row-wise as $D = [\, b_1 \ \cdots \ b_R \,]^\top$ with $b_i \in \mathbb{R}^n$, we obtain

$$A = \sum_{i=1}^{R} c_i b_i^\top,$$

where we set $\alpha_i = c_i$, $\beta_i = b_i$.

(2) Suppose $A = \sum_{i=1}^{R} \alpha_i \beta_i^\top$. For any $x \in \mathbb{R}^n$,

$$Ax = \sum_{i=1}^{K} \alpha_i (\beta_i^\top x) \in \operatorname{span}\{\alpha_1, \ldots, \alpha_K\}.$$

Hence the column space of $A$ is contained in $\operatorname{span}\{\alpha_i\}$, which has dimension at most $K$. Therefore $\operatorname{rank}(A) \leq K$. Combining with part (1), the minimal number of rank-one matrices required is exactly $\operatorname{rank}(A)$. $\qquad\square$

*Proof of Proposition D.4.* We may write

$$f_r(L) = \sum_{i=0}^{K} \alpha_{r,i} L^i, \qquad h_r(L) = \sum_{j=0}^{K} \beta_{r,j} L^j,$$

where $\alpha_{r,i} = 0$ and $\beta_{r,j} = 0$ whenever $i > \deg f_r$ or $j > \deg h_r$, and $\deg f_r + \deg h_r \leq K$ for all $r$. Then

$$\sum_{r=1}^{R} f_r(L) \otimes h_r(L) = \sum_{r=1}^{R} \sum_{i,j=0}^{K} \alpha_{r,i} \beta_{r,j} \, L^i \otimes L^j = \sum_{i,j=0}^{K} \left( \sum_{r=1}^{R} \alpha_{r,i} \beta_{r,j} \right) L^i \otimes L^j.$$

For each $r$, define coefficient vectors

$$\alpha_r = [\, \alpha_{r,0}, \alpha_{r,1}, \ldots, \alpha_{r,K} \,]^\top \in \mathbb{R}^{K+1}, \qquad \beta_r = [\, \beta_{r,0}, \beta_{r,1}, \ldots, \beta_{r,K} \,]^\top \in \mathbb{R}^{K+1}.$$

Let $B \in \mathbb{R}^{(K+1) \times (K+1)}$ be the coefficient matrix $B = \sum_{r=1}^{R} \alpha_r \beta_r^\top$, i.e. $B_{ij} = \sum_{r=1}^{R} \alpha_{r,i} \beta_{r,j}$. Then

$$\sum_{r=1}^{R} f_r(L) \otimes h_r(L) = \sum_{i,j=0}^{K} B_{ij} \, L^i \otimes L^j.$$

On the other hand, we know $P(I_n \otimes L, L \otimes I_n) = \sum_{i,j=0}^{K} A_{ij} L^i \otimes L^j$, then we have the equivalence

$$P(I_n \otimes L, L \otimes I_n) = \sum_{r=1}^{R} f_r(L) \otimes h_r(L) \quad \Longleftrightarrow \quad A = B.$$

Suppose there exists such $f_r$ and $h_r$, with $A = B = \sum_{r=1}^{R} \alpha_r \beta_r^\top$, we know $\operatorname{rank}(A) \leq R$ by Lemma D.5. Conversely, without loss of generality, we assume $\operatorname{rank}(A) = R$. Through the decomposition in Lemma D.5, we have $A = \sum_{r=1}^{K} \alpha_r \beta_r^\top$. Then the polynomials can be constructed by $f_r(L) = \sum_{i=0}^{K} \alpha_{r,i} L^i$ and $h_r(L) = \sum_{i=0}^{K} \beta_{r,i} L^i$.

$\qquad\square$

**Corollary D.6.** *Let $L = U\Lambda U^\top \in \mathbb{R}^{n \times n}$ denote the eigendecomposition of a symmetric Laplacian, and let $M$ be a square matrix of the same dimension. Then the filtered matrix $P(I \otimes L, L \otimes I)\operatorname{vec}(M)$ admits the spectral expansion*

$$P(I \otimes L, L \otimes I)(M) = \sum_{i,j} g(\lambda_i, \lambda_j)\langle u_i, Mu_j\rangle u_i u_j^\top$$

*where $g(\lambda_i, \lambda_j) = \sum_{r=1}^{R} f_r(\lambda_i)\, h_r(\lambda_j)$ for polynomial pairs $\{(f_r, h_r)\}_{r=1}^{R}$.*

*Proof.* According to Proposition D.4, there exist polynomial pairs $\{(f_r, h_r)\}_{r=1}^{R}$ such that

$$P(I \otimes L,\, L \otimes I) = \sum_{r=1}^{R} f_r(L) \otimes h_r(L).$$

Since $f_r(L) = \sum_i f_r(\lambda_i)\, u_i u_i^\top$ and $h_r(L) = \sum_j h_r(\lambda_j)\, u_j u_j^\top$, we have

$$\Big(f_r(L) \otimes h_r(L)\Big)(M) = f_r(L)\, M\, h_r(L) = \Big(\sum_i f_r(\lambda_i)u_i u_i^\top\Big)M\Big(\sum_j h_r(\lambda_j)u_j u_j^\top\Big)$$

$$= \sum_{i,j} f_r(\lambda_i)h_r(\lambda_j)\, u_i \underbrace{(u_i^\top M u_j)}_{\langle u_i, Mu_j\rangle}\, u_j^\top.$$

Summing over $r$ completes the proof. $\qquad\square$

# E. Expressivity of Full-Spectrum GNNs

## E.1. Universal Approximation on Node-Pair Signals

We restate and prove Theorem 3.4 as follows.

**Theorem E.1.** *Let $\tilde{L} = U\Lambda U^\top$ be the normalized Laplacian with a simple spectrum , i.e., $\lambda_1, \ldots, \lambda_n$ are pairwise distinct. Consider a* linear FSPECGNN *acting on a $d$-channel node-pair representation $H \in \mathbb{R}^{n \times n \times d} \cong \mathbb{R}^{n^2 \times d}$:*

$$\mathcal{T}_{g,W}(H) := (U \otimes U)\Big(G_\lambda \odot \big((U \otimes U)^\top H\big)\Big)W, \qquad (G_\lambda)_{ij} = g(\lambda_i, \lambda_j),$$

*where $g : \mathbb{R}^2 \to \mathbb{R}$ is a bivariate spectral function, and $W \in \mathbb{R}^{d \times 1}$ produces a one-dimensional output. Assume that the input $H$ satisfies that for every $1 \le i, j \le n$ there exists at least one channel $c \in \{1, \cdots, d\}$ with*

$$\big(U^\top H^{(c)} U\big)_{ij} \neq 0.$$

*Then for any target one-dimensional node-pair signal $Y \in \mathbb{R}^{n \times n} \cong \mathbb{R}^{n^2}$, there exist a vector $W \in \mathbb{R}^{d \times 1}$ and a bivariate function $g$ such that $\mathcal{T}_{g,W}(H) = Y$.*

*Proof.* Write the $c$-th channel of $H$ as $H^{(c)} \in \mathbb{R}^{n \times n}$, and define

$$C^{(c)} := U^\top H^{(c)} U \in \mathbb{R}^{n \times n}, \qquad c \in \{1, \cdots, d\}.$$

For a feature transformation matrix $W = (w_1, \ldots, w_d)^\top \in \mathbb{R}^{d \times 1}$, define the transformed one-dimensional representation

$$\bar{H} := \sum_{c=1}^{d} w_c\, H^{(c)} \in \mathbb{R}^{n \times n}, \qquad \bar{C} := U^\top \bar{H} U = \sum_{c=1}^{d} w_c\, C^{(c)}.$$

Therefore we have:

$$\mathcal{T}_{g,W}(H) = U\Big(G_\lambda \odot (U^\top \bar{H} U)\Big)U^\top = U\Big(G_\lambda \odot \bar{C}\Big)U^\top.$$

Hence, to realize an arbitrary target $Y$, it suffices to choose $W$ so that $\bar{C}_{ij} \neq 0$ for all $i, j$, and then choose $g$ so that $g(\lambda_i, \lambda_j)\bar{C}_{ij} = \widehat{Y}_{ij}$, where $\widehat{Y} := U^\top Y U$.

**Choose $W$.** Fix any $(i,j)$. By the support assumption, there exists at least one channel $c$ with $C_{ij}^{(c)} \neq 0$. Therefore the scalar linear form

$$\bar{C}_{ij} = \sum_{c=1}^{d} w_c \, C_{ij}^{(c)}$$

is not identically zero, and its zero set $\{W : \bar{C}_{ij} = 0\}$ is a proper hyperplane in $\mathbb{R}^d$. Since there are finitely many pairs $(i,j)$, the union

$$\mathcal{Z} := \bigcup_{i=1}^{n} \bigcup_{j=1}^{n} \{W : \bar{C}_{ij} = 0\}$$

is a finite union of proper hyperplanes, hence a measure-zero subset of $\mathbb{R}^d$. Consequently, we can pick $W \notin \mathcal{Z}$, so that

$$\bar{C}_{ij} \neq 0, \qquad \forall \, i, j \in \{1, \cdots, n\}.$$

**Choose $g$.** With such a choice of $W$, define $\widehat{Y} := U^\top Y U$ and

$$M_{ij} := \frac{\widehat{Y}_{ij}}{\bar{C}_{ij}}, \qquad \forall \, i, j \in \{1, \cdots, n\}.$$

If we construct $g$ satisfying $g(\lambda_i, \lambda_j) = M_{ij}$ for all $i, j$, then we will entrywisely have

$$\left( U^\top \mathcal{T}_{g,W}(H) U \right)_{ij} = g(\lambda_i, \lambda_j) \, \bar{C}_{ij} = M_{ij} \, \bar{C}_{ij} = \widehat{Y}_{ij},$$

so $U^\top \mathcal{T}_{g,W}(H) U = \widehat{Y}$ and thus $\mathcal{T}_{g,W}(H) = Y$.

Because the spectrum is simple, define the univariate Lagrange polynomials

$$\ell_i(s) := \prod_{k \neq i} \frac{s - \lambda_k}{\lambda_i - \lambda_k}, \qquad \text{so that} \qquad \ell_i(\lambda_{i'}) = \mathbf{1}[i = i'].$$

Now define the bivariate polynomial

$$q(s,t) := \sum_{i=1}^{n} \sum_{j=1}^{n} M_{ij} \, \ell_i(s) \, \ell_j(t).$$

Then for any pair $(i,j)$,

$$q(\lambda_i, \lambda_j) = \sum_{i'=1}^{n} \sum_{j'=1}^{n} M_{i'j'} \, \ell_{i'}(\lambda_i) \, \ell_{j'}(\lambda_j) = M_{ij}.$$

Taking $g = q$ completes the construction and yields $\mathcal{T}_{q,W}(H) = Y$. $\qquad \square$

### E.2. Non-Isomorphic Graph Distinguishability of FSPECGNNs

At the node level, we have the following result reformulated from Wang & Zhang (2022), which states that any spectral GNN expressed as a polynomial of the normalized Laplacian is upper-bounded in expressive power by 1-WL.

**Proposition E.2.** *Let $G = (V, E, \ell)$ be a labeled graph with node labels $\ell : V \to \mathbb{R}^d$, and let $\widetilde{L}$ denote its normalized graph Laplacian. Consider a $K$-th order linear polynomial spectral GNN*

$$Z = p(\widetilde{L}) X W = \sum_{t=0}^{K} \alpha_t \, \widetilde{L}^t (XW),$$

*where $X$ is the matrix form of $\ell$ and $W \in \mathbb{R}^{d \times d'}$. For each node $u \in V$, define the output at node $u$ by $Z(u) = [\, Z \,]_{u,:}$, i.e. the $u$-th row of $Z \in \mathbb{R}^{n \times d'}$. Let $\chi_G^{1-\mathrm{WL},(k)}(u)$ denote the color of node $u$ on graph $G$ after $k$ iterations of the 1-WL refinement initialized with labels $\ell$. Then, for any $i, j \in V$,*

$$\chi_G^{1-\mathrm{WL},(K+1)}(i) = \chi_G^{1-\mathrm{WL},(K+1)}(j) \implies Z(i) = Z(j).$$

*Proof.* Let $\widetilde{A} = I - \widetilde{L}$ denote the normalized adjacency matrix, then there exist real coefficients $\{\theta_t\}_{t=0}^K$ such that $Z$ can be equivalently written as

$$Z = \sum_{t=0}^K \alpha_t (I - \widetilde{A})^t (XW) = \sum_{t=0}^K \theta_t \, \widetilde{A}^t (XW),$$

Accordingly, $Z$ can be viewed as a $(K+1)$-step embedding

$$h_i^{(k+1)} = H^{(k)}\left( h_i^{(k)}, \; \{\!\{ h_j^{(k)} : j \in N_G(i) \}\!\} \right),$$

where $h_i^{(k)}$ is the embedding of node $i$ at step $k$ with $h_i^{(0)} = (XW)_{i,:}$, $N_G(i)$ denotes the neighborhood of node $i$ in graph $G$ and each update function $H^{(k)}$ is defined as follows.

- For the first step $k = 0$:

$$H^{(0)}\left( \ell(i), \; \{\!\{ \ell(j) : j \in N(i) \}\!\} \right) = \left( |N(i)|, \; \theta_K \ell(i), \; \ell(i) \right).$$

- For intermediate steps $k \in \{1, 2, \ldots, K-1\}$:

$$H^{(k)}\left( h_i^{(k)}, \; \{\!\{ h_j^{(k)} : j \in N(i) \}\!\} \right) =$$

$$\left( p_1(h_i^{(k)}), \; \sum_{j \in N(i)} \frac{p_2(h_j^{(k)})}{\sqrt{p_1(h_i^{(k)})\, p_1(h_j^{(k)})}} + \theta_{K-k}\, p_3(h_i^{(k)}), \; p_3(h_i^{(k)}) \right).$$

- For the final step $k = K$:

$$H^{(K)}\left( h_i^{(K)}, \; \{\!\{ h_j^{(K)} : j \in N(i) \}\!\} \right) = \sum_{j \in N(i)} \frac{p_2(h_j^{(K)})}{\sqrt{p_1(h_i^{(K)})\, p_1(h_j^{(K)})}} + \theta_0\, p_3(h_i^{(K)}).$$

Here $p_1$, $p_2$ and $p_3$ denote the projections onto the respective coordinates of the vector $h_i^{(k)}$, namely,

$$p_1(h_i^{(k+1)}) = p_1(h_i^{(k)}), \tag{6}$$

$$p_2(h_i^{(k+1)}) = \sum_{j \in N(i)} \frac{p_2(h_j^{(k)})}{\sqrt{p_1(h_i^{(k)})\, p_1(h_j^{(k)})}} + \theta_{K-k}\, p_3(h_i^{(k)}), \tag{7}$$

$$p_3(h_i^{(k+1)}) = p_3(h_i^{(k)}) \tag{8}$$

for $k \in \{0, 1, \cdots, K-1\}$ with $p_1(h_i^{(0)}) = |N(i)|$, $p_2(h_i^{(0)}) = \theta_K \ell(i)$ and $p_3(h_i^{(0)}) = \ell(i)$. According to (6) and (8) with their initializations, we can rewrite (7) as

$$p_2(h_i^{(k+1)}) = \sum_{j \in N_G(i)} \widetilde{A}_{ij}\, p_2(h_j^{(k)}) + \theta_{K-k}\, \ell(i) = \sum_j \widetilde{A}_{ij}\, p_2(h_j^{(k)}) + \theta_{K-k}\, \ell(i),$$

in which $\widetilde{A}_{ij}$ denotes the $(i,j)$-entry of the normalized adjacency matrix $\widetilde{A}$. For the final step, we similarly have

$$h_i^{(K+1)} = \sum_j \widetilde{A}_{ij}\, p_2(h_j^{(K)}) + \theta_0\, \ell(i).$$

Hence, we inductively find that the overall output is the composition of all embedding steps

$$Z = H^{(K)} \circ H^{(K-1)} \circ \cdots \circ H^{(0)}(XW).$$

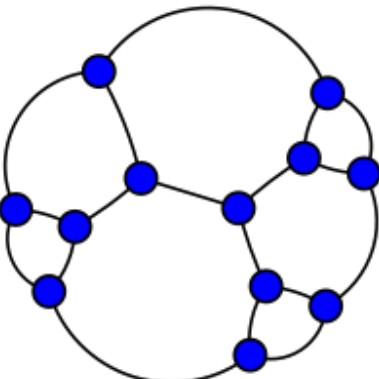

*Figure 4.* The Frucht graph (image adapted from Wikipedia contributors (2025)). Despite having a trivial automorphism group, the stable 1-WL refinement assigns a single color to all vertices.

Recall that the 1-WL color refinement is defined by

$$\chi^{1-\mathrm{WL},(k+1)}(i) = \mathsf{hash}\left(\chi^{1-\mathrm{WL},(k)}(i), \; \{\!\!\{\chi^{1-\mathrm{WL},(k)}(j) : j \in N(i)\}\!\!\}\right),$$

where hash is a perfect hash function and $\chi^{1-\mathrm{WL},(0)}(i) = \ell(i)$. Since each update function $H^{(k)}$ depends only on the current state of node $i$ and the multiset of its neighbors' states through permutation-invariant operations that are shared across all nodes, the updates $H^{(k)}$ are consistent with the refinement steps of 1-WL. Consequently, if two nodes have the same 1-WL color after the $k$-th iteration, then they receive identical updates under $H^{(k)}$, i.e.,

$$\chi_G^{1-\mathrm{WL},(k)}(i) = \chi_G^{1-\mathrm{WL},(k)}(j) \quad \Longrightarrow \quad h_i^{(k)} = h_j^{(k)}.$$

By induction over $k$, all intermediate representations remain equal, and therefore the final outputs coincide:

$$\chi_G^{1-\mathrm{WL},(K+1)}(i) = \chi_G^{1-\mathrm{WL},(K+1)}(j) \quad \Longrightarrow \quad y(i) = y(j).$$

$\square$

A subtle conceptual issue arises when bridging the apparent inconsistency noted in Wang & Zhang (2022). In Wang & Zhang (2022), Proposition 4.3 shows that, for a fixed graph and fixed node features, a linear GNN with a degree-$K$ spectral polynomial filter is bounded by $(K+1)$ iterations of the 1-WL test, as formulated above. On the other hand, Theorem 4.1 establishes that, under the "no multiple eigenvalues" and "no missing frequency components" conditions, linear spectral GNNs enjoy a universal approximation property and can, in principle, map all nodes to distinct outputs. This seems at odds with the well-known fact that 1-WL often fails to distinguish certain pairs of vertices.

To bridge this gap, Wang & Zhang (2022) argues that their spectral conditions imply that 1-WL can distinguish all "non-isomorphic nodes" (Corollary 4.4), and moreover that the graph admits no nontrivial labeled automorphisms (Theorem 4.6). Consequently, every pair of nodes becomes non-isomorphic in their sense, and 1-WL must distinguish all vertices, thereby removing the apparent contradiction with Proposition 4.3.

Although the formal statements of Theorem 4.5 and Theorem 4.6 are mathematically correct, the way they are invoked to resolve the apparent inconsistency is not justified, as it is important to distinguish between two similar yet inequivalent notions of node equivalence. One notion is *automorphism-based equivalence*: two vertices are considered "isomorphic" if they lie in the same orbit of the labeled automorphism group $\mathrm{Aut}(G, \ell)$. The other is 1-WL *equivalence*: two vertices receive the same stable 1-WL color or, equivalently, have isomorphic unrolling trees (Morris et al., 2020). In general, these notions do not coincide. Even with a trivial automorphism group, 1-WL may still collapse all nodes to a single color. A classical example is the Frucht graph (see Figure 4): it is 3-regular and asymmetric, i.e. its automorphism group consists only of the identity, yet 1-WL with uniform initial labels fails to distinguish any of its vertices.

The reconciliation between universality and the 1-WL upper bound does not actually stem from the absence of nontrivial automorphisms. Rather, it is the universality assumptions themselves that severely constrain the admissible graph-signal

pairs: a simple Laplacian spectrum, together with labels having full spectral support, enforces a highly non-degenerate labeling. When 1-WL is run on such labels, its refinement becomes much more discriminative; in fact, combined with the upper bound in Proposition 4.3, universality forces the stable 1-WL partition on $(G, \ell)$ to refine all the way down to singletons, since any two vertices mapped to different values by a universal linear spectral GNN must already be separated by 1-WL.

We restate and prove Theorem 3.7 as follows.

**Theorem E.3.** *Let $G = (V, E, \ell)$ be a labeled graph with node labels $\ell : V \to \mathbb{R}^d$, and let $\widetilde{L}$ denote its normalized graph Laplacian. Define*

$$A := \widetilde{L} \otimes I_n, \qquad B := I_n \otimes \widetilde{L},$$

*where $n = |V|$. Let*

$$p(x, y) = \sum_{\substack{a,b \geq 0 \\ a+b \leq K}} \alpha_{ab} x^a y^b$$

*be a bivariate polynomial of total degree at most $K$. Consider the edge-level signal*

$$\varepsilon : V \times V \to \mathbb{R}^{2d+2}, \qquad \varepsilon(u, v) = \big(\ell(u), \ell(v), \mathbb{I}[u = v], \mathbb{I}[\{u, v\} \in E]\big),$$

*and let $\mathcal{L} \in \mathbb{R}^{n^2 \times (2d+2)}$ be its matrix form, i.e., the $(u, v)$-th row of $\mathcal{L}$ is $\varepsilon(u, v)$. For a learnable matrix $W \in \mathbb{R}^{(2d+2) \times d'}$, define*

$$\mathcal{L}' := p(A, B)\mathcal{L}W \in \mathbb{R}^{n^2 \times d'}.$$

*For $(u, v) \in V \times V$, denote by*

$$\mathcal{L}'(u, v) := \mathcal{L}'_{(u,v),:}$$

*the $(u, v)$-th row of $\mathcal{L}'$.*

*Let $\chi_G^{\text{Local2-GNN},(k)}(u, v)$ denote the color of the pair $(u, v)$ after $k$ iterations of the Local 2-GNN refinement initialized with the pair label $\varepsilon$. Then for any $(u_1, v_1), (u_2, v_2) \in V \times V$,*

$$\chi_G^{\text{Local2-GNN},(K)}(u_1, v_1) = \chi_G^{\text{Local2-GNN},(K)}(u_2, v_2) \implies \mathcal{L}'(u_1, v_1) = \mathcal{L}'(u_2, v_2).$$

We utilize the following definition and lemma from Morris et al. (2020).

**Definition E.4.** Let $G = (V, E, \ell)$ be a labeled graph. For a pair $(u, v) \in V \times V$, define its depth-$k$ local unrolling tree, denoted by

$$\text{L-UNR}_G^{(k)}(u, v),$$

recursively as follows.

- For $k = 0$, $\text{L-UNR}_G^{(0)}(u, v)$ consists of a single root, whose vertex label is the initial pair label

$$\varepsilon(u, v) = \big(\ell(u), \ell(v), \mathbb{I}[u = v], \mathbb{I}[\{u, v\} \in E]\big).$$

- For $k \geq 1$, $\text{L-UNR}_G^{(k)}(u, v)$ is the rooted directed tree obtained by taking a root labeled by $\varepsilon(u, v)$ and attaching:

  1. for every $u' \in N_G(u)$, one child whose attached subtree is $\text{L-UNR}_G^{(k-1)}(u', v)$, with the edge from the root to this child labeled by 1;
  2. for every $v' \in N_G(v)$, one child whose attached subtree is $\text{L-UNR}_G^{(k-1)}(u, v')$, with the edge from the root to this child labeled by 2.

Two such trees are said to be isomorphic if there exists a bijection between their vertex sets that preserves: (i) the root, (ii) the parent–child relation, (iii) the vertex labels, and (iv) the directed edge labels in $\{1, 2\}$.

**Lemma E.5** ([Morris et al. (2020)](#)). *For any labeled graph $G$, any $(u_1, v_1), (u_2, v_2) \in V \times V$, and any $k \geq 0$,*

$$\chi_G^{\text{Local2-GNN},(k)}(u_1, v_1) = \chi_G^{\text{Local2-GNN},(k)}(u_2, v_2)$$

*if and only if*

$$\text{L-UNR}_G^{(k)}(u_1, v_1) \cong \text{L-UNR}_G^{(k)}(u_2, v_2).$$

*Proof of Theorem [E.3](#).* Assume that

$$\chi_G^{\text{Local2-GNN},(K)}(u_1, v_1) = \chi_G^{\text{Local2-GNN},(K)}(u_2, v_2).$$

By Lemma [E.5](#), this implies

$$\text{L-UNR}_G^{(K)}(u_1, v_1) \cong \text{L-UNR}_G^{(K)}(u_2, v_2).$$

Set

$$X := \mathcal{L}W \in \mathbb{R}^{n^2 \times d'}.$$

Since the initial pair label is exactly $\varepsilon$, and $X(u, v) = \varepsilon(u, v)W$, the value $X(u, v)$ depends only on the root label of $\text{L-UNR}_G^{(0)}(u, v)$. Hence, whenever two depth-0 local unrolling trees are isomorphic, the corresponding rows of $X$ are equal.

We now claim that for every $a, b \geq 0$ with $a + b \leq K$,

$$(A^a B^b X)(u_1, v_1) = (A^a B^b X)(u_2, v_2),$$

where $A := \widetilde{L} \otimes I_n$ and $B := I_n \otimes \widetilde{L}$. Once this is proved, linearity yields

$$\mathcal{L}'(u_1, v_1) = \sum_{a+b \leq K} \alpha_{ab}(A^a B^b X)(u_1, v_1) = \sum_{a+b \leq K} \alpha_{ab}(A^a B^b X)(u_2, v_2) = \mathcal{L}'(u_2, v_2),$$

which is exactly the desired conclusion.

It remains to prove the claim. We proceed by induction on the total degree $a + b$.

**Base case:** $a + b = 0$**.** Then $a = b = 0$, so

$$A^0 B^0 X = X.$$

Since the roots of $\text{L-UNR}_G^{(0)}(u_1, v_1)$ and $\text{L-UNR}_G^{(0)}(u_2, v_2)$ have the same label under the above isomorphism, we obtain

$$X(u_1, v_1) = X(u_2, v_2).$$

**Induction step:** Assume the claim holds for all pairs $(a', b')$ with $a' + b' \leq t - 1$, where $1 \leq t \leq K$, and let $a + b = t$. We consider two cases.

*Case 1: $a \geq 1$.* Write

$$A^a B^b X = A(A^{a-1} B^b X).$$

For any $(u, v) \in V \times V$, using the row-wise action of $A = \widetilde{L} \otimes I_n$, we have

$$(A^a B^b X)(u, v) = \widetilde{L}_{uu}(A^{a-1} B^b X)(u, v) + \sum_{u' \in N_G(u)} \widetilde{L}_{uu'}(A^{a-1} B^b X)(u', v).$$

Since

$$\text{L-UNR}_G^{(K)}(u_1, v_1) \cong \text{L-UNR}_G^{(K)}(u_2, v_2),$$

their roots have the same label $\varepsilon$, hence in particular $\ell(u_1) = \ell(u_2)$, $\ell(v_1) = \ell(v_2)$, and

$$\mathbb{I}[u_1 = v_1] = \mathbb{I}[u_2 = v_2], \qquad \mathbb{I}[\{u_1, v_1\} \in E] = \mathbb{I}[\{u_2, v_2\} \in E].$$

Moreover, because the rooted tree isomorphism preserves the multiset of children connected by an edge labeled 1, there is a bijection between the neighbors of $u_1$ and the neighbors of $u_2$ matching

$$\text{L-UNR}_G^{(K-1)}(u', v_1) \quad \text{and} \quad \text{L-UNR}_G^{(K-1)}(\phi(u'), v_2)$$

for a suitable bijection $\phi : N_G(u_1) \to N_G(u_2)$. In particular,

$$\deg_G(u_1) = \deg_G(u_2),$$

and therefore the corresponding normalized Laplacian coefficients coincide:

$$\widetilde{L}_{u_1 u_1} = \widetilde{L}_{u_2 u_2}, \qquad \widetilde{L}_{u_1 u'} = \widetilde{L}_{u_2 \phi(u')} \quad \text{for all } u' \in N_G(u_1),$$

since for the normalized Laplacian the off-diagonal entry depends only on the existence of the edge and the endpoint degrees. Now $a - 1 + b = t - 1 \le K - 1$, so by the induction hypothesis,

$$(A^{a-1} B^b X)(u_1, v_1) = (A^{a-1} B^b X)(u_2, v_2),$$

and for every $u' \in N_G(u_1)$,

$$(A^{a-1} B^b X)(u', v_1) = (A^{a-1} B^b X)(\phi(u'), v_2).$$

Substituting these equalities into the above expansion gives

$$(A^a B^b X)(u_1, v_1) = (A^a B^b X)(u_2, v_2).$$

*Case 2: $b \ge 1$.* This is completely analogous. Writing

$$A^a B^b X = B(A^a B^{b-1} X),$$

for any $(u, v) \in V \times V$ we have

$$(A^a B^b X)(u, v) = \widetilde{L}_{vv} (A^a B^{b-1} X)(u, v) + \sum_{v' \in N_G(v)} \widetilde{L}_{vv'} (A^a B^{b-1} X)(u, v').$$

Since the rooted tree isomorphism preserves the children connected by an edge labeled 2, there is a bijection

$$\psi : N_G(v_1) \to N_G(v_2)$$

such that

$$\text{L-UNR}_G^{(K-1)}(u_1, v') \cong \text{L-UNR}_G^{(K-1)}(u_2, \psi(v'))$$

for all $v' \in N_G(v_1)$. Hence

$$\deg_G(v_1) = \deg_G(v_2),$$

and the corresponding normalized Laplacian coefficients also coincide. Since $a + (b - 1) = t - 1$, the induction hypothesis implies

$$(A^a B^{b-1} X)(u_1, v_1) = (A^a B^{b-1} X)(u_2, v_2),$$

and for every $v' \in N_G(v_1)$,

$$(A^a B^{b-1} X)(u_1, v') = (A^a B^{b-1} X)(u_2, \psi(v')).$$

Therefore,

$$(A^a B^b X)(u_1, v_1) = (A^a B^b X)(u_2, v_2).$$

This completes the induction, and hence proves the theorem. $\qquad \square$

We restate and prove Theorem 3.8 as follows.

**Theorem E.6.** *Following the notation of Theorem E.3, consider linear polynomial* FSPECGNN *of the form*

$$\mathcal{L}' = q(\widetilde{L} \otimes I, \ I \otimes \widetilde{L})(\mathcal{L}W),$$

*where $W \in \mathbb{R}^{n^2 \times 1}$ is a transform matrix. Let $\{u_i\}_{i=1}^n$ denote the eigenvectors in the eigendecomposition $\widetilde{L} = \sum_{i=1}^n \lambda_i u_i u_i^\top$. Suppose that the graph and signal satisfy:*

(i) *$\widetilde{L}$ has a simple spectrum, i.e. $\lambda_1, \ldots, \lambda_n$ are pairwise distinct;*

(ii) *the spectral coefficients $c_{ij} := u_i^\top \mathrm{Matrix}(\mathcal{L}W) \, u_j$ are nonzero for every pair $(i,j) \in \{1, \ldots, n\}^2$.*

*Let $\chi_G^{\text{Local2-GNN},(\infty)}(i, j)$ denote the stable Local 2-GNN color of the pair $(u, v) \in V \times V$. Then there exists a polynomial $q$ in two variables such that the output satisfies*

$$\mathcal{L}'(u_1, v_1) = \mathcal{L}'(u_2, v_2) \quad \Longrightarrow \quad \chi_G^{\text{Local2-GNN},(\infty)}(u_1, v_1) = \chi_G^{\text{Local2-GNN},(\infty)}(u_2, v_2).$$

*Equivalently, the partition of $V \times V$ induced by $\mathcal{L}'$ is a refinement of the stable Local 2-GNN coloring. Moreover, if the stable Local 2-GNN partition on $V \times V$ is not discrete, i.e. some color class has size at least two, then this refinement is strict.*

*Proof.* Since $\widetilde{L}$ has eigendecomposition $\widetilde{L} = \sum_{i=1}^n \lambda_i u_i u_i^\top$ with pairwise distinct eigenvalues, the commuting operators

$$A := \widetilde{L} \otimes I \quad \text{and} \quad B := I \otimes \widetilde{L}$$

on $\mathbb{R}^{n^2}$ admit a joint eigenbasis $\{u_i \otimes u_j\}_{i,j=1}^n$. On this basis, $A$ and $B$ act as

$$A(u_i \otimes u_j) = \lambda_i(u_i \otimes u_j), \qquad B(u_i \otimes u_j) = \lambda_j(u_i \otimes u_j).$$

Viewing $\varepsilon$ as a vector in $\mathbb{R}^{n^2}$ via the usual vectorization, it has a unique expansion

$$\varepsilon = \sum_{i,j=1}^n c_{ij} (u_i \otimes u_j), \qquad c_{ij} = (u_i \otimes u_j)^\top \varepsilon = u_i^\top \varepsilon \, u_j.$$

By assumption (ii), all $c_{ij}$ are nonzero.

For any real-valued function $g : V \times V \to \mathbb{R}$, we may write its vectorization in the same basis as

$$g = \sum_{i,j=1}^n d_{ij} (u_i \otimes u_j)$$

for suitable coefficients $d_{ij} \in \mathbb{R}$. Consider a polynomial $q$ in two variables. Its action on $\varepsilon$ is diagonal in the joint eigenbasis:

$$q(A, B) \varepsilon = \sum_{i,j=1}^n q(\lambda_i, \lambda_j) \, c_{ij} (u_i \otimes u_j).$$

Thus, to realize the desired set of coefficients $\{d_{ij}\}$, it suffices to choose $q$ such that

$$q(\lambda_i, \lambda_j) \, c_{ij} = d_{ij} \quad \Longleftrightarrow \quad q(\lambda_i, \lambda_j) = \frac{d_{ij}}{c_{ij}} \quad \text{for all } i, j.$$

Since the grid $\{(\lambda_i, \lambda_j) : 1 \le i, j \le n\}$ is finite and all $c_{ij} \neq 0$, there always exists a bivariate polynomial $q$ interpolating the values $d_{ij}/c_{ij}$ at these $n^2$ points, e.g. by multivariate Lagrange interpolation. Therefore, under assumptions (i) and (ii), the family $q(\widetilde{L} \otimes I, \ I \otimes \widetilde{L})$ acting on $\varepsilon$ is universal on the finite domain $V \times V$: it can realize any target signal $g : V \times V \to \mathbb{R}$.

Now let $\mathcal{C} = \{C_1, \ldots, C_s\}$ denote the stable Local 2-GNN partition of $V \times V$, where each $C_t$ is a color class

$$C_t = \left\{ (i,j) \in V \times V : \chi_G^{\text{Local2-GNN},(\infty)}(i,j) = t \right\}.$$

Choose pairwise disjoint intervals $I_1, \ldots, I_s \subset \mathbb{R}$, and for each $t \in \{1, \ldots, s\}$ assign distinct real values $\{\gamma_{t,\alpha} \in I_t : (i_\alpha, j_\alpha) \in C_t\}$ to the elements of $C_t$. Define the target function $g : V \times V \to \mathbb{R}$ by

$$g(i,j) = \gamma_{t,\alpha} \quad \text{if } (i,j) = (i_\alpha, j_\alpha) \in C_t.$$

By construction, if two pairs belong to different Local 2-GNN color classes, say $(u_1, v_1) \in C_{t_1}$ and $(u_2, v_2) \in C_{t_2}$ with $t_1 \neq t_2$, then $g(u_1, v_1) \in I_{t_1}$ and $g(u_2, v_2) \in I_{t_2}$, so $g(u_1, v_1) \neq g(u_2, v_2)$. Hence

$$g(u_1, v_1) = g(u_2, v_2) \quad \implies \quad \chi_G^{\text{Local2-GNN},(\infty)}(u_1, v_1) = \chi_G^{\text{Local2-GNN},(\infty)}(u_2, v_2).$$

Moreover, if some color class $C_t$ has size at least two, we chose the $\gamma_{t,\alpha}$ within $I_t$ to be pairwise distinct, so $g$ is not constant on $C_t$ and the induced partition strictly refines $\mathcal{C}$.

By Theorem E.1, there exists a polynomial $q$ such that

$$\mathcal{L}' = q(\widetilde{L} \otimes I, \ I \otimes \widetilde{L})\,(\mathcal{L}W).$$

For this $q$ we therefore have

$$\mathcal{L}'(u_1, v_1) = \mathcal{L}'(u_2, v_2) \quad \implies \quad \chi_G^{\text{Local2-GNN},(\infty)}(u_1, v_1) = \chi_G^{\text{Local2-GNN},(\infty)}(u_2, v_2).$$

i.e. the partition refines the stable Local 2-GNN coloring. If some $C_t$ has size at least two, the refinement is strict by our choice of $g$. $\qquad\square$

## F. An Analysis of Spectral Filters for ClassWise Separability

### F.1. Optimal Convolution under Classwise Separability

#### F.1.1. SETTING AND NOTATION

Let $G = (V, E)$ be a graph with $|V| = n$. Nodes are partitioned into $k$ classes $\Pi = \{V_1, \ldots, V_k\}$ with $|V_a| = n_a$ and $\sum_a n_a = n$. For each class $a \in \{1, \ldots, k\}$, we associate a mean vector $m_a \in \mathbb{R}^d$, drawn independently and uniformly at random from the unit sphere $\mathbb{S}^{d-1}$. Conditioned on these means, node features are drawn independently according to their class labels:

$$x_i \sim \mathcal{D}_{a_i}, \qquad a_i \text{ such that } i \in V_{a_i},$$

where each class distribution $\mathcal{D}_a$ on $\mathbb{R}^d$ has mean $m_a$ and finite covariance $\Sigma_a \succeq 0$. Denote by $X \in \mathbb{R}^{n \times d}$ the resulting feature matrix with rows $x_i^\top$, and write $\tau_a := \text{tr}(\Sigma_a)$ for the total variance of class $a$, where we assume $\tau_a > 0$ for every $a$. This model can be viewed as a high-dimensional isotropic mixture in which class means are randomly oriented and features fluctuate around them with bounded covariance.

A (linear) convolution is a matrix $C \in \mathbb{R}^{n \times n}$ producing $Y := CX$. We evaluate $C$ by the classwise mean-regression MSE

$$\mathcal{L}(C) := \sum_{a=1}^k \frac{1}{n_a} \sum_{p \in V_a} \mathbb{E} \left\| Y_p - m_a \right\|_2^2. \tag{9}$$

**Definition F.1.** Let $\Pi = \{V_1, \ldots, V_k\}$ be the class partition of the node set $V$ with $|V_a| = n_a$. For each class $a$, let $\mathfrak{S}_{n_a}$ be the permutation group acting on the indices $V_a$, and define the product group

$$\mathcal{G} := \prod_{a=1}^k \mathfrak{S}_{n_a},$$

whose elements $\pi = (\pi_1, \ldots, \pi_k)$ act on $V$ by independently permuting the nodes within each class. The corresponding permutation matrix $P_\pi$ acts on vectors or matrices as a block permutation operator.

A matrix $C \in \mathbb{R}^{n \times n}$ is said to be *classwise permutation equivariant* if

$$C = P_\pi \, C \, P_\pi^\top, \qquad \forall \, \pi \in \mathcal{G}.$$

Equivalently, $C$ commutes with the representation of $\mathcal{G}$ on $\mathbb{R}^n$.

For any matrix $C$, its *Reynolds average* with respect to $\mathcal{G}$ is defined as

$$\overline{C} := \mathbb{E}_{\pi \sim \mathrm{Unif}(\mathcal{G})} \big[ P_\pi \, C \, P_\pi^\top \big],$$

which is classwise permutation equivariant and thus belongs to the commutant of $\mathcal{G}$.

### F.1.2. Algebraic Structure and a Bias–Variance Decomposition

Before analyzing the loss $\mathcal{L}(C)$ in its explicit form, we first show that the Reynolds average of any matrix $C$ can only decrease the loss. This allows us to restrict attention, without loss of generality, to classwise-equivariant matrices.

**Lemma F.2.** *Let $\mathcal{G} = \prod_{a=1}^{k} \mathfrak{S}_{n_a}$ act on indices by classwise permutations, and define the Reynolds average of $C \in \mathbb{R}^{n \times n}$ by $\overline{C}$. Then the classwise mean-regression loss satisfies*

$$\mathcal{L}(\overline{C}) \leq \mathcal{L}(C).$$

*Proof.* For fixed feature matrix $X \in \mathbb{R}^{n \times d}$, with $\| \cdot \|_F$ denoting the Frobenius norm, write

$$\ell(C; X) := \sum_{a=1}^{k} \frac{1}{n_a} \big\| (CX)_a - \mathbf{1}_{n_a} m_a^\top \big\|_F^2.$$

The map $C \mapsto CX$ is linear, and $A \mapsto \sum_a \frac{1}{n_a} \| A_a - \mathbf{1} m_a^\top \|_F^2$ is a quadratic form in $A$, and thus convex. Hence $\ell(C; X)$ is convex in $C$, and taking expectation over $X$ preserves convexity, so $\mathcal{L}(C) = \mathbb{E}_X[\ell(C; X)]$ is convex in $C$. Since $\overline{C}$ is the group average of linear transforms of $C$, convexity gives

$$\ell(\overline{C}; X) = \ell\big( \mathbb{E}_\pi[P_\pi C P_\pi^\top]; X \big) \leq \mathbb{E}_\pi \, \ell(P_\pi C P_\pi^\top; X).$$

Taking expectation in $X$ and exchanging the order of expectations,

$$\mathcal{L}(\overline{C}) = \mathbb{E}_X \ell(\overline{C}; X) \leq \mathbb{E}_\pi \mathbb{E}_X \ell(P_\pi C P_\pi^\top; X).$$

For any $\pi \in \mathcal{G}$,

$$\ell(P_\pi C P_\pi^\top; X) = \ell(C; P_\pi^\top X),$$

because $(P_\pi C P_\pi^\top) X = P_\pi [C(P_\pi^\top X)]$ and Frobenius norms are invariant under left multiplication by the permutation matrices $P_{\pi_a}$ within each class. Under the classwise i.i.d. model in Section F.1.1, $P_\pi^\top X$ and $X$ are identically distributed, so $\mathbb{E}_X \ell(C; P_\pi^\top X) = \mathbb{E}_X \ell(C; X) = \mathcal{L}(C)$. Substituting this into the previous inequality yields

$$\mathcal{L}(\overline{C}) \leq \mathcal{L}(C).$$

$\square$

**Corollary F.3.** *An optimal solution of $\min_C \mathcal{L}(C)$ can be chosen among classwise permutation equivariant matrices.*

We next characterize the algebraic structure of all classwise permutation equivariant matrices, which will serve as the reduced parameter space in which the optimization can be carried out according to Corollary F.3.

**Lemma F.4.** *Let $\mathcal{G} = \prod_{a=1}^{k} \mathfrak{S}_{n_a}$ act on $\mathbb{R}^n$ by permuting the coordinates within each class $V_a$. A matrix $C \in \mathbb{R}^{n \times n}$ is classwise permutation equivariant if and only if there exist real scalars $\{\alpha_a, \beta_a\}_{a=1}^{k}$ and $\{\gamma_{ab}\}_{a \neq b}$ such that*

$$C = \sum_{a=1}^{k} \big( \alpha_a I_{V_a} + \beta_a J_{V_a} \big) + \sum_{a \neq b} \gamma_{ab} \, J_{V_a, V_b}, \tag{10}$$

*where:*

- $I_{V_a}$ denotes the identity matrix on the coordinates indexed by $V_a$;

- $J_{V_a}$ is the $n_a \times n_a$ all-ones matrix;

- $J_{V_a,V_b}$ is the $n_a \times n_b$ all-ones block between classes $a$ and $b$.

*Equivalently, the commutant of the representation of $\mathcal{G}$ on $\mathbb{R}^n$ is spanned by the matrices $\{I_{V_a}, J_{V_a}, J_{V_a,V_b} : a \neq b\}$.*

*Proof.* Let $C$ commute with every $P_\pi \in \mathcal{G}$. Since $\mathcal{G}$ acts independently on each class block, the condition $P_\pi C = C P_\pi$ decomposes into intra-class and inter-class parts.

**Intra-class blocks:** Fix a class $a$ and consider $C_{V_a,V_a}$, the submatrix of $C$ with both rows and columns indexed by the nodes in $V_a$. For any $\pi_a \in \mathfrak{S}_{n_a}$, $P_{\pi_a} C_{V_a,V_a} = C_{V_a,V_a} P_{\pi_a}$. It is a classical result that the commutant of the full symmetric group acting by permutation matrices on $\mathbb{R}^{n_a}$ is two-dimensional, spanned by $I_{V_a}$ and $J_{V_a}$. Hence there exist scalars $\alpha_a, \beta_a$ such that $C_{V_a,V_a} = \alpha_a I_{V_a} + \beta_a J_{V_a}$.

**Inter-class blocks:** For $a \neq b$, consider $C_{V_a,V_b}$. The commutation condition with independent permutations $P_{\pi_a}$ on $V_a$ and $P_{\pi_b}$ on $V_b$ implies

$$P_{\pi_a} C_{V_a,V_b} P_{\pi_b}^\top = C_{V_a,V_b}, \qquad \forall\, (\pi_a, \pi_b) \in \mathfrak{S}_{n_a} \times \mathfrak{S}_{n_b}.$$

The only matrices invariant under left and right actions of the full symmetric groups are constant matrices, i.e. scalar multiples of the all-ones block $J_{V_a,V_b}$. Thus there exists $\gamma_{ab} \in \mathbb{R}$ such that $C_{V_a,V_b} = \gamma_{ab} J_{V_a,V_b}$. Combining all blocks Eq.(10).

Conversely, any $C$ of the form Eq.(10) clearly commutes with every $P_\pi \in \mathcal{G}$, so the characterization is complete. □

**Corollary F.5.** *For any matrix $C \in \mathbb{R}^{n \times n}$, its Reynolds average $\overline{C} = \mathbb{E}_{\pi \sim \mathrm{Unif}(\mathcal{G})}[P_\pi C P_\pi^\top]$ lies in the commutant characterized in Lemma F.4 and thus admits the representation Eq.(10).*

We then turn to the statistical objective $\mathcal{L}(C)$ and express it in a form amenable to optimization.

**Lemma F.6.** *Let $Y = CX$ and write $D_\tau := \bigoplus_{a=1}^k \tau_a I_{V_a}$. For any $C$ and any $p \in V_a$,*

$$\mathbb{E}\, Y_p = \sum_{b=1}^k C_{p,V_b}\,(\mathbf{1}_{n_b} m_b^\top), \qquad \mathrm{tr}\, \mathrm{Var}(Y_p) = \sum_{b=1}^k \tau_b\, \|C_{p,V_b}\|_2^2.$$

*Consequently,*

$$\mathcal{L}(C) = \underbrace{\sum_{a=1}^k \frac{1}{n_a} \sum_{p \in V_a} \|\mathbb{E} Y_p - m_a\|_2^2}_{\text{squared bias}} + \underbrace{\sum_{a=1}^k \frac{1}{n_a} \sum_{p \in V_a} \sum_{b=1}^k \tau_b\, \|C_{p,V_b}\|_2^2}_{\text{variance}}. \tag{11}$$

*Proof.* For each class $b$, the random vectors $\{x_j : j \in V_b\}$ are i.i.d. with $\mathbb{E}[x_j] = m_b$ and $\mathrm{Cov}(x_j) = \Sigma_b$. Hence

$$\mathbb{E}[X_{V_b}] = \mathbf{1}_{n_b} m_b^\top.$$

Let $C_{p,V_b} \in \mathbb{R}^{1 \times n_b}$ denote the row of $C$ restricted to columns $V_b$. By classwise independence,

$$\mathrm{Var}(C_{p,V_b} X_{V_b}) = \sum_{j \in V_b} C_{pj}^2 \Sigma_b = \|C_{p,V_b}\|_2^2\, \Sigma_b,$$

so that

$$\mathbb{E}\|C_{p,V_b}(X_{V_b} - \mathbb{E} X_{V_b})\|_2^2 = \mathrm{tr}\big[\mathrm{Var}(C_{p,V_b} X_{V_b})\big] = \|C_{p,V_b}\|_2^2\, \mathrm{tr}(\Sigma_b) = \tau_b \|C_{p,V_b}\|_2^2.$$

Finally, using the bias–variance identity

$$\mathbb{E}\|Z - m\|_2^2 = \|\mathbb{E} Z - m\|_2^2 + \mathrm{tr}\, \mathrm{Var}(Z),$$

and substituting $Z = Y_p = C_{p,:} X$, we obtain (11). □

**Corollary F.7.** *Suppose $C$ is classwise permutation equivariant, i.e. it has the form Eq.(10). Then, for any $p \in V_a$,*

$$\mathbb{E} Y_p = (\alpha_a + \beta_a n_a) m_a + \sum_{b \neq a} \gamma_{ab} n_b m_b,$$

*and, writing $\tau_b := \mathrm{tr}(\Sigma_b)$,*

$$\mathrm{tr} \, \mathrm{Var}(Y_p) = \tau_a \Big( (\alpha_a + \beta_a)^2 + (n_a - 1)\beta_a^2 \Big) + \sum_{b \neq a} \tau_b n_b \gamma_{ab}^2.$$

*Consequently, the per-class contribution to the loss equals the quadratic*

$$\phi_a(\alpha_a, \beta_a, \gamma_{a\bullet}) = \Big\| u_a + M_{-a} \gamma_{a\bullet} \Big\|_2^2 + \tau_a \Big( (\alpha_a + \beta_a)^2 + (n_a - 1)\beta_a^2 \Big) + \gamma_{a\bullet}^\top D_{-a} \gamma_{a\bullet}, \qquad (12)$$

*where $\gamma_{a\bullet} := (\gamma_{ab})_{b \neq a} \in \mathbb{R}^{k-1}$, $u_a := (\alpha_a + n_a \beta_a - 1) m_a$, $M_{-a} = [n_b m_b]_{b \neq a} \in \mathbb{R}^{d \times (k-1)}$, $D_{-a} = \mathrm{diag}(\tau_b n_b)_{b \neq a} \succeq 0$.*

**Theorem F.8.** *Under the model assumed in Section F.1.1, the minimizer of $\min_C \mathcal{L}(C)$ has the following block structure.*

*(1) Intra-class blocks. For each class $a$,*

$$(C^\star)_{aa} = \alpha_a^\star I_{V_a} + \beta_a^\star J_{V_a}.$$

- *If $n_a > 1$, then $\alpha_a^\star = 0$ and*

$$\beta_a^\star = \frac{A_a}{n_a A_a + \tau_a}, \quad A_a = \|m_a\|_2^2 - \sum_{\substack{b \neq a \\ c \neq a}} n_b n_c \, \langle m_a, m_b \rangle \big[ (G_{-a} + D_{-a})^{-1} \big]_{bc} \langle m_c, m_a \rangle,$$

  *where $G_{-a}$ and $D_{-a}$ are defined in Corollary F.7.*

- *If $n_a = 1$, the objective depends only on $\alpha_a + \beta_a$, and one may set $\alpha_a^\star = 0$ without loss of generality.*

*(2) Inter-class blocks. For every $b \neq a$,*

$$(C^\star)_{ab} = \gamma_{ab}^\star J_{V_a, V_b},$$

*where the coefficients $\gamma_{ab}^\star$ satisfy the linear system*

$$\sum_{c \neq a} (G_{-a} + D_{-a})_{bc} \, \gamma_{ac}^\star = (1 - n_a \beta_a^\star) \, n_b \, \langle m_a, m_b \rangle, \quad b \neq a.$$

*Equivalently, solving for $\gamma_{ab}^\star$ gives*

$$\gamma_{ab}^\star = \frac{\tau_a}{n_a A_a + \tau_a} \sum_{c \neq a} \big[ (G_{-a} + D_{-a})^{-1} \big]_{bc} n_c \, \langle m_a, m_c \rangle.$$

*The minimizer $C^\star$ is unique.*

*Proof.* By Corollary F.3, we may restrict to the commutant form Eq.(10). Fix a class $a$. From Corollary F.7, the per-class contribution of $V_a$ to $\mathcal{L}(C)$ is Eq.(12).

**Convexity and strict convexity:** For each class $a$, recall from Eq.(12) that

$$\phi_a(\alpha_a, \beta_a, \gamma_{a\bullet}) = \Big\| u_a + M_{-a} \gamma_{a\bullet} \Big\|_2^2 + \tau_a \Big( (\alpha_a + \beta_a)^2 + (n_a - 1)\beta_a^2 \Big) + \gamma_{a\bullet}^\top D_{-a} \gamma_{a\bullet},$$

where $u_a = (\alpha_a + n_a \beta_a - 1) m_a$, $M_{-a} = [n_b m_b]_{b \neq a}$, and $D_{-a} = \mathrm{diag}(\tau_b n_b)_{b \neq a} \succ 0$ as we assume any $\tau_b > 0$ in Section F.1.1. The Hessian of $\phi_a$ with respect to $(\alpha_a, \beta_a, \gamma_{a\bullet})$ is the block matrix

$$H_a = 2 \begin{bmatrix} A & B \\ B^\top & G_{-a} + D_{-a} \end{bmatrix}, \qquad A = \begin{bmatrix} \|m_a\|_2^2 + \tau_a & n_a \|m_a\|_2^2 + \tau_a \\ n_a \|m_a\|_2^2 + \tau_a & n_a^2 \|m_a\|_2^2 + n_a \tau_a \end{bmatrix}, \qquad B = \begin{bmatrix} h_a^\top \\ n_a h_a^\top \end{bmatrix},$$

with $G_{-a} = M_{-a}^\top M_{-a} \succeq 0$ and $h_a = M_{-a}^\top m_a$.

Each term in Eq.(12) is a nonnegative quadratic form: the first term is a squared Euclidean norm, the second is $\tau_a$ times a positive semidefinite quadratic in $(\alpha_a, \beta_a)$, and the last is $\gamma_{a\bullet}^\top D_{-a}\gamma_{a\bullet}$ with $D_{-a} \succ 0$. Therefore $\phi_a$ is convex and $H_a \succeq 0$ for any class $a$.

To determine when $H_a$ is positive definite, apply the Schur complement criterion to its block form:

$$H_a \succ 0 \quad \Longleftrightarrow \quad A \succ 0 \text{ and } S := G_{-a} + D_{-a} - B^\top A^{-1} B \succ 0.$$

A direct calculation gives

$$\det A = \tau_a \, (n_a - 1) \, (n_a \|m_a\|_2^2 + \tau_a), \qquad A_{11} = \|m_a\|_2^2 + \tau_a > 0.$$

Hence $A \succ 0$ if and only if $n_a > 1$ and $\tau_a > 0$. Under these assumptions, we compute

$$B^\top A^{-1} B = c \, h_a h_a^\top, \qquad c = \frac{n_a}{n_a \|m_a\|_2^2 + \tau_a} \in \left(0, \frac{1}{\|m_a\|_2^2}\right),$$

so that

$$S = (G_{-a} - c \, h_a h_a^\top) + D_{-a}.$$

For any $v \in \mathbb{R}^{k-1}$, the Cauchy–Schwarz inequality yields

$$v^\top (G_{-a} - c \, h_a h_a^\top) v = \|M_{-a} v\|_2^2 - c \, \langle m_a, M_{-a} v\rangle^2 \ \geq \ (1 - c\|m_a\|_2^2) \, \|M_{-a} v\|_2^2.$$

Since $\tau_a > 0$ implies $1 - c\|m_a\|_2^2 > 0$ and $D_{-a} \succ 0$ when all $\tau_b > 0$, we obtain

$$v^\top S v \ \geq \ (1 - c\|m_a\|_2^2) \, \|M_{-a} v\|_2^2 + \sum_{b \neq a} \tau_b n_b \, v_b^2 \ > \ 0 \quad \text{for all } v \neq 0.$$

Thus $S \succ 0$ and hence $H_a \succ 0$ whenever $n_a > 1$ and $\tau_a > 0$ for all classes. Consequently, each $\phi_a$ is strictly convex, so is the overall loss $\mathcal{L}(C) = \sum_a \frac{1}{n_a}\phi_a$.

**Eliminating $\gamma_{a\bullet}$:** For fixed $(\alpha_a, \beta_a)$,

$$\nabla_\gamma \phi_a = 2M_{-a}^\top (u_a + M_{-a}\gamma_{a\bullet}) + 2D_{-a}\gamma_{a\bullet}.$$

Setting the gradient to zero gives the normal equations

$$(G_{-a} + D_{-a}) \, \gamma_{a\bullet} + M_{-a}^\top u_a = 0.$$

Note that $G_{-a} + D_{-a} \succ 0$, its unique solution is

$$\gamma_{a\bullet}^\star(\alpha_a, \beta_a) = -(G_{-a} + D_{-a})^{-1} M_{-a}^\top u_a = -(\alpha_a + n_a\beta_a - 1) \, (G_{-a} + D_{-a})^{-1} h_a. \tag{13}$$

**Solving for $\alpha_a^\star$, $\beta_a^\star$ and $\gamma_{a\bullet}^\star$:** Define the reduced function

$$\tilde{\phi}_a(\alpha_a, \beta_a) := \min_\gamma \phi_a(\alpha_a, \beta_a, \gamma) = \phi_a(\alpha_a, \beta_a, \gamma_{a\bullet}^\star(\alpha_a, \beta_a)).$$

Because $\phi_a$ is convex-quadratic and the minimizer in $\gamma$ is unique, the envelope theorem implies that the partial derivatives of $\tilde{\phi}_a$ are simply the partial derivatives of $\phi_a$ evaluated at $\gamma_{a\bullet}^\star$; that is,

$$\frac{\partial \tilde{\phi}_a}{\partial \alpha_a} = \frac{\partial \phi_a}{\partial \alpha_a}\bigg|_{\gamma = \gamma_{a\bullet}^\star}, \qquad \frac{\partial \tilde{\phi}_a}{\partial \beta_a} = \frac{\partial \phi_a}{\partial \beta_a}\bigg|_{\gamma = \gamma_{a\bullet}^\star}.$$

Hence, when differentiating we may treat $\gamma_{a\bullet}^\star$ as a constant.

Let $z_a := u_a + M_{-a}\gamma_{a\bullet}^\star$, we obtain the first-order conditions

$$\partial_{\alpha_a} \tilde{\phi}_a = 2\langle z_a, m_a\rangle + 2\tau_a(\alpha_a + \beta_a) = 0, \qquad \partial_{\beta_a} \tilde{\phi}_a = 2n_a\langle z_a, m_a\rangle + 2\tau_a(\alpha_a + n_a\beta_a) = 0. \tag{14}$$

Multiply the first equality of Eq.(14) by $n_a$ and subtract it from the second:

$$0 = 2(n_a - 1)\tau_a\,\alpha_a.$$

Thus $\alpha_a^\star = 0$ for $n_a > 1$. When $n_a = 1$, the two equations coincide, so the loss depends only on $\alpha_a + \beta_a$; we set $\alpha_a^\star = 0$ without loss of generality.

With $\alpha_a = 0$, $u_a = (n_a\beta_a - 1)m_a$. Using Eq.(13),

$$M_{-a}\gamma_{a\bullet}^\star = -(n_a\beta_a - 1)\,M_{-a}(G_{-a} + D_{-a})^{-1}h_a,$$

and hence

$$\langle z_a, m_a \rangle = (n_a\beta_a - 1)\Big(\|m_a\|_2^2 - h_a^\top (G_{-a} + D_{-a})^{-1}h_a\Big) = (n_a\beta_a - 1)A_a,$$

where

$$A_a := \|m_a\|_2^2 - h_a^\top (G_{-a} + D_{-a})^{-1}h_a.$$

Substituting into the first equation of Eq.(14) (with $\alpha_a = 0$) gives

$$(n_a\beta_a - 1)A_a + \tau_a\beta_a = 0, \qquad \Rightarrow \qquad \beta_a^\star = \frac{A_a}{n_a A_a + \tau_a}.$$

Substituting $\beta_a^\star$ into Eq.(13) yields

$$\gamma_{a\bullet}^\star = -(n_a\beta_a^\star - 1)(G_{-a} + D_{-a})^{-1}h_a = \frac{\tau_a}{n_a A_a + \tau_a}(G_{-a} + D_{-a})^{-1}h_a,$$

which is equivalent to the stated linear system and its explicit solution.

**Nonnegativity of $A_a$ and uniqueness:** The quantity $A_a$ is nonnegative. Indeed, for any scalar $t$,

$$\min_\gamma \big\| t\,m_a + M_{-a}\gamma \big\|_2^2 + \gamma^\top D_{-a}\gamma = t^2\Big(\|m_a\|_2^2 - h_a^\top(G_{-a} + D_{-a})^{-1}h_a\Big) = t^2 A_a \geq 0.$$

Hence $A_a \geq 0$. Finally, given $n_a > 1$ for every $a$, strict convexity of $\phi_a$ ensures uniqueness of the minimizer.

When $n_a = 1$, the function $\phi_a$ depends on $(\alpha_a, \beta_a)$ only through their sum $s_a = \alpha_a + \beta_a$. Indeed, substituting $n_a = 1$ into Eq.(12) gives

$$\phi_a(\alpha_a, \beta_a, \gamma_{a\bullet}) = \big\|(s_a - 1)m_a + M_{-a}\gamma_{a\bullet}\big\|_2^2 + \tau_a s_a^2 + \gamma_{a\bullet}^\top D_{-a}\gamma_{a\bullet}, \qquad s_a = \alpha_a + \beta_a.$$

Hence $\phi_a$ is strictly convex in $s_a$ and $\gamma_{a\bullet}$, but flat along the direction $(1, -1, 0, \ldots, 0)$ in the parameter space of $(\alpha_a, \beta_a, \gamma_{a\bullet})$. Consequently, the minimizer $(\alpha_a^\star, \beta_a^\star, \gamma_{a\bullet}^\star)$ is not unique: all pairs $(\alpha_a, \beta_a)$ satisfying $\alpha_a + \beta_a = s_a^\star$ yield the same minimal value, where $s_a^\star$ and $\gamma_{a\bullet}^\star$ are uniquely determined. Nevertheless, the corresponding intra-class block

$$(C^\star)_{aa} = \alpha_a^\star I_{V_a} + \beta_a^\star J_{V_a} = (\alpha_a^\star + \beta_a^\star)J_{V_a} = s_a^\star J_{V_a}$$

is unique, so the matrix $C^\star$ itself remains uniquely defined. $\qquad\qquad\square$

When features are sampled from a high-dimensional distribution, the optimal convolution is nearly block-diagonal, and its intra-class coefficients asymptotically approach a deterministic limit. We then restate and prove Theorem 4.1 as follows.

**Theorem F.9.** *Under the model assumed in Section F.1.1, let $C^\star$ be the minimizer characterized in Theorem F.8. Assume $\tau_b n_b \geq \tau_0 > 0$ for all $b$, where $\tau_0$ is a fixed positive constant providing a uniform lower bound on the classwise variance scales. Then, as the dimensions of features $d \to \infty$,*

$$\beta_a^\star \xrightarrow{\;\mathbb{P}\;} \frac{1}{n_a + \tau_a}, \qquad \gamma_{ab}^\star \xrightarrow{\;\mathbb{P}\;} 0 \quad (b \neq a).$$

*Hence the optimal convolution $C^\star$ becomes asymptotically block-diagonal, with intra-class rank-one blocks $s_a^\star J_{V_a}$ where $s_a^\star \to (n_a + \tau_a)^{-1}$ in probability.*

*Proof.* We keep $k$ and the class sizes $\{n_b\}_{b=1}^k$ fixed as $d \to \infty$. Under the model in Section F.1.1, the class means satisfy $\|m_a\|_2 \equiv 1$ for all $a$, and each $\tau_a$ is fixed, hence $1/(n_a + \tau_a)$ denotes a deterministic limit.

**Concentration of random directions:** If $u, v \sim \mathrm{Unif}(\mathbb{S}^{d-1})$ are independent, then $\Pr\big(|\langle u, v\rangle| \geq t/\sqrt{d}\big) \leq 2e^{-ct^2}$ for some universal $c > 0$. Let $t = \sqrt{C \log d}$ with $C > 2/c$. By a union bound over ordered pairs $(a, b)$ with $b \neq a$, with probability at least $1 - 2k(k-1)d^{-cC}$,

$$|\langle m_a, m_b\rangle| \leq \sqrt{\frac{C \log d}{d}} \quad \text{for all } a \neq b.$$

Writing $h_a := M_{-a}^\top m_a$ with entries $(h_a)_b = n_b \langle m_b, m_a\rangle$ $(b \neq a)$,

$$\|h_a\|_2^2 = \sum_{b \neq a} n_b^2 \langle m_b, m_a\rangle^2 \leq \frac{C \log d}{d} \sum_{b \neq a} n_b^2,$$

and since $k, \{n_b\}$ are fixed, uniformly for all $a$,

$$\|h_a\|_2 = \mathcal{O}_\mathbb{P}\bigg(\sqrt{\frac{\log d}{d}}\bigg),$$

where $\mathcal{O}_\mathbb{P}$ indicates that the bound holds in probability, i.e., $|h_a|_2/\sqrt{\frac{\log d}{d}}$ is bounded in probability.

**Controlling** $(G_{-a} + D_{-a})^{-1}$. By assumption, $D_{-a} = \mathrm{diag}(\tau_b n_b)_{b \neq a} \succeq \tau_0 I$, hence

$$G_{-a} + D_{-a} \succeq D_{-a} \succeq \tau_0 I \implies \|(G_{-a} + D_{-a})^{-1}\|_{\mathrm{op}} \leq \tau_0^{-1}.$$

Here $\|\cdot\|_{\mathrm{op}}$ denotes the operator norm, equal to the largest singular value of a symmetric matrix and to the maximal eigenvalue in absolute value. Therefore,

$$h_a^\top (G_{-a} + D_{-a})^{-1} h_a \leq \|(G_{-a} + D_{-a})^{-1}\|_{\mathrm{op}} \cdot \|h_a\|_2^2 = \mathcal{O}_\mathbb{P}\bigg(\frac{\log d}{d}\bigg),$$

uniformly in $a$.

**Asymptotics of** $A_a, \gamma_{a\bullet}^\star$, **and the diagonal block coefficients:** By Theorem F.8, we have

$$A_a := \|m_a\|_2^2 - h_a^\top (G_{-a} + D_{-a})^{-1} h_a = 1 - \mathcal{O}_\mathbb{P}\bigg(\frac{\log d}{d}\bigg)$$

and

$$n_a A_a + \tau_a = (n_a + \tau_a)\big[1 + \mathcal{O}_\mathbb{P}((\log d)/d)\big].$$

Moreover, the inter-class coefficients satisfy

$$\gamma_{a\bullet}^\star = \frac{\tau_a}{n_a A_a + \tau_a}(G_{-a} + D_{-a})^{-1} h_a,$$

so using the bounds above with $k, \{n_b\}$ fixed,

$$\|\gamma_{a\bullet}^\star\|_2 \leq \frac{\tau_a}{n_a + \tau_a} \cdot \frac{1 + \mathcal{O}_\mathbb{P}((\log d)/d)}{\tau_0} \cdot \|h_a\|_2 = \mathcal{O}_\mathbb{P}\bigg(\sqrt{\frac{\log d}{d}}\bigg),$$

uniformly in $a$. Hence each component $\gamma_{ab}^\star \to 0$ in probability for $b \neq a$.

For the diagonal blocks we distinguish the two cases but treat them under a unified limit.

- *Case $n_a > 1$.* Theorem F.8 yields $\alpha_a^\star = 0$ and

$$\beta_a^\star = \frac{A_a}{n_a A_a + \tau_a} = \frac{1 - \mathcal{O}_\mathbb{P}((\log d)/d)}{(n_a + \tau_a)\big[1 + \mathcal{O}_\mathbb{P}((\log d)/d)\big]} = \frac{1}{n_a + \tau_a} + \mathcal{O}_\mathbb{P}\bigg(\frac{\log d}{d}\bigg),$$

so $\beta_a^\star \xrightarrow{\mathbb{P}} 1/(n_a + \tau_a)$ and $(C^\star)_{aa} = \beta_a^\star J_{V_a}$.

- *Case $n_a = 1$.* In this case $\phi_a$ depends on $(\alpha_a, \beta_a)$ only through $s_a = \alpha_a + \beta_a$ and

$$\gamma_{a\bullet}^\star = -(s_a - 1)(G_{-a} + D_{-a})^{-1}h_a, \qquad \tilde{\phi}_a(s_a) = \min_\gamma \phi_a(s_a, \gamma) = (s_a - 1)^2 A_a + \tau_a s_a^2.$$

The first-order condition gives $(s_a - 1)A_a + \tau_a s_a = 0$, hence

$$s_a^\star = \frac{A_a}{A_a + \tau_a} = \frac{1 - \mathcal{O}_{\mathbb{P}}((\log d)/d)}{(1 + \tau_a)\left[1 + \mathcal{O}_{\mathbb{P}}((\log d)/d)\right]} = \frac{1}{1 + \tau_a} + \mathcal{O}_{\mathbb{P}}\left(\frac{\log d}{d}\right),$$

so $s_a^\star \xrightarrow{\mathbb{P}} 1/(1 + \tau_a)$ and $(C^\star)_{aa} = s_a^\star J_{V_a}$.

Combining the two cases, define the diagonal block coefficient

$$t_a^\star := \begin{cases} \beta_a^\star, & n_a > 1, \\ s_a^\star, & n_a = 1, \end{cases}$$

then in both cases $t_a^\star \xrightarrow{\mathbb{P}} 1/(n_a + \tau_a)$ and $(C^\star)_{aa} = t_a^\star J_{V_a}$.

**Asymptotic block-diagonality.** Each off-diagonal block satisfies $(C^\star)_{ab} = \gamma_{ab}^\star J_{V_a, V_b}$, thus

$$\|(C^\star)_{ab}\|_{\mathrm{op}} = \sqrt{n_a n_b}\, |\gamma_{ab}^\star| \leq \sqrt{n_a n_b}\, \|\gamma_{a\bullet}^\star\|_2 = \mathcal{O}_{\mathbb{P}}\left(\sqrt{\frac{\log d}{d}}\right) \to 0.$$

Therefore $C^\star$ converges in probability to a block-diagonal matrix with intra-class rank-one blocks $t_a^\star J_{V_a}$, where $t_a^\star \to (n_a + \tau_a)^{-1}$ in probability.

$\square$

## F.2. Spectral GNNs Cannot Express the Optimal Convolution

**Definition F.10.** Let $L \in \mathbb{R}^{n \times n}$ be a real symmetric matrix, e.g. a graph Laplacian. Denote by

$$\mathrm{spec}(L) = \{\lambda_1 < \lambda_2 < \cdots < \lambda_m\}$$

the set of distinct eigenvalues of $L$, and let each $\lambda$ have multiplicity $m_\lambda = \dim \ker(L - \lambda I)$. For each $\lambda \in \mathrm{spec}(L)$, let $U_\lambda \in \mathbb{R}^{n \times m_\lambda}$ collect an orthonormal basis of $\ker(L - \lambda I)$. Define the orthogonal projector onto this eigenspace by

$$E_\lambda := U_\lambda U_\lambda^\top.$$

*Remark* F.11. Then $\{E_\lambda\}_{\lambda \in \mathrm{spec}(L)}$ satisfies

$$E_\lambda^\top = E_\lambda, \quad E_\lambda^2 = E_\lambda, \quad E_\lambda E_\mu = 0 \ (\lambda \neq \mu), \quad \sum_{\lambda \in \mathrm{spec}(L)} E_\lambda = I_n,$$

and the spectral resolution

$$L = \sum_{\lambda \in \mathrm{spec}(L)} \lambda E_\lambda.$$

Moreover, $E_\lambda$ is independent of the choice of orthonormal basis $U_\lambda$.

To simplify the discussion, we assume the feature dimension $d \to +\infty$, under which the optimal convolution minimizing $\mathcal{L}(C)$ becomes block-diagonal:

$$(C^\star)_{aa} = \frac{1}{n_a + \tau_a}, \qquad (C^\star)_{ab} = 0 \ (b \neq a).$$

From the viewpoint of expressiveness, our discussion concerns not a single operator but an entire *family of convolution operators* parameterized by learnable coefficients. A spectral GNN corresponds to the family

$$\mathcal{S} = \mathrm{span}\{E_\lambda : \lambda \in \mathrm{spec}(L)\} = \{\, C(g) = \textstyle\sum_\lambda g(\lambda)\, E_\lambda : \ g \in \mathbb{R}^{\mathrm{spec}(L)} \,\},$$

that is, all linear filters expressible as functions of the Laplacian $L$. The expressive power of this model family depends on whether, through learning, it can realize or approximate the desired target convolution $C^\star$.

We show that the optimal convolution $C^\star$ lies strictly outside this spectral subspace and cannot even be approximated within it to arbitrary accuracy. In other words, no choice of spectral coefficients $g(\lambda)$ can reproduce or arbitrarily approach $C^\star$. In analogy with control theory, the family of spectral convolutions is *structurally unreachable* from $C^\star$, indicating that this function class is inherently incapable of reaching the optimal convolution during learning, and thereby revealing a fundamental limitation in its representational capacity. We restate and prove Theorem 4.2 as follows.

**Theorem F.12.** *Let $G = (V, E)$ be a connected graph with Laplacian eigendecomposition $L = \sum_{\lambda \in \mathrm{spec}(L)} \lambda\, E_\lambda$ as in the definition above, and $|V| = n$. Define the class of spectral convolution matrices*

$$\mathcal{C} := \left\{ C(c) = \sum_{\lambda \in \mathrm{spec}(L)} c_\lambda\, E_\lambda \ : \ c = (c_\lambda)_{\lambda \in \mathrm{spec}(L)} \in \mathbb{R}^m \right\},$$

*where $m = |\mathrm{spec}(L)|$. Each $C(c)$ applies a scalar response $c_\lambda$ to the entire eigenspace of eigenvalue $\lambda$.*

*Then there exists an integer $K \geq 1$ such that, for any integer $K < k \leq n$, there exists a $k$-partition $V = V_1 \cup \cdots \cup V_k$, with labels $y_i = t$ if and only if $v_i \in V_t$, having the following property: if $C(c) \in \mathcal{C}$ satisfies*

$$C(c)_{ij} = 0 \quad \text{whenever } y_i \neq y_j,$$

*then necessarily $C(c) = \alpha I_n$ for some $\alpha \in \mathbb{R}$.*

To prove the theorem, we begin with some preparations.

**Definition F.13** (Indexing conventions). Let $M \in \mathbb{R}^{a \times b}$ be a matrix.

- For a row index $i \in [a]$, we denote by $M_{-i} \in \mathbb{R}^{(a-1) \times b}$ the matrix obtained by removing the $i$-th row of $M$.

- For row and column indices $i \in [a]$ and $j \in [b]$, we denote by $M_{-i,-j} \in \mathbb{R}^{(a-1) \times (b-1)}$ the matrix obtained by removing the $i$-th row and the $j$-th column of $M$.

- For a column subset $S \subseteq [b]$, we write $M_{[:,S]} \in \mathbb{R}^{a \times |S|}$ for the submatrix consisting of the columns of $M$ indexed by $S$.

- When $S = \{\ell\}$ is a singleton, we abbreviate $M_{[:,\{\ell\}]}$ as $M_{:,\ell}$.

These notations will be used throughout for brevity.

Let $L = U\Lambda U^\top$ be as above. For a vertex $i \in V$, set

$$D_i := \mathrm{diag}\big(u_1(i), u_2(i), \ldots, u_n(i)\big) \in \mathbb{R}^{n \times n}.$$

Define

$$M_i := U_{-i}\, D_i \ \in \ \mathbb{R}^{(n-1) \times n}.$$

Equivalently, the $\ell$-th column of $M_i$ is $(M_i)_{:,\ell} = u_\ell(i)\,\big(u_\ell\big)_{-i}$, i.e. it vanishes if and only if $u_\ell(i) = 0$, and otherwise it is a nonzero scalar multiple of $u_\ell$ with its $i$-th entry removed.

We will collect a vertical stack of such matrices so that each newly added block *activates* at least one previously-zero column and together they cover all columns:

**Definition F.14.** Let $I = (i_1, \ldots, i_r)$ be an ordered list of vertices, and define the stacked matrix

$$M_I := \begin{bmatrix} M_{i_1} \\ \vdots \\ M_{i_r} \end{bmatrix} \in \mathbb{R}^{r(n-1) \times n}.$$

For each vertex $i \in V$, denote by

$$S_i := \{\ell \in [n] : (M_i)_{:,\ell} \neq 0\}$$

the set of column indices activated by $M_i$.

We say that $I$ is *column-activating* if it satisfies

- Every column of $M_I$ contains at least one nonzero entry, i.e. $\bigcup_{k=1}^{r} S_{i_k} = [n]$.

- Each step $k$ introduces a new activation, i.e. $\exists \ell \in S_{i_k} \setminus \bigcup_{t<k} S_{i_t}$. Equivalently, $M_{i_k}$ contributes a nonzero entry in a column that was entirely zero in the previous partial stack $[M_{i_1}; \ldots; M_{i_{k-1}}]$.

We call such an index list $I$ *minimal* if it is inclusion-minimal with respect to the activated columns, namely

$$\bigcup_{i \in J} S_i = \bigcup_{i \in I} S_i \implies J = I \quad \text{for all } J \subseteq I.$$

*Remark* F.15 (Greedy construction). The greedy rule produces a column-activating list $I$. Starting from the empty cover $S^{(0)} := \varnothing$, we iteratively choose nodes that maximally enlarge the current coverage:

$$i_k := \arg\max_{i \in V} \big| S_i \setminus S^{(k-1)} \big|, \qquad S^{(k)} := S^{(k-1)} \cup S_{i_k}.$$

The process stops once $S^{(k)} = [n]$, meaning each column of $M_I$ is nonzero in some block $M_{i_t}$. The resulting index list $I = (i_1, i_2, \ldots, i_K)$ is *column-activating*.

While we do not rely on the optimality of its length, each selected block $M_{i_k}$ strictly increases the coverage $S^{(k)}$, so the greedy rule yields a minimal column-activating list in the sense of Definition F.14.

**Proposition F.16.** *Let $U = [u_1, \ldots, u_n] \in \mathbb{R}^{n \times n}$ be an orthogonal matrix $U^\top U = I_n$, and let $U_k = [u_1, \ldots, u_k] \in \mathbb{R}^{n \times k}$ with $1 \leq k \leq n$. Fix an index $s \in \{1, \ldots, n\}$, and let $e_s \in \mathbb{R}^n$ denote the $s$-th standard basis vector. Define the row-selection matrix*

$$R := \begin{bmatrix} I_{s-1} & 0 & 0 \\ 0 & 0 & I_{n-s} \end{bmatrix} \in \mathbb{R}^{(n-1) \times n},$$

*which deletes the $s$-th row of any vector or matrix it multiplies on the left. Let $\widetilde{U}_k := R U_k = (U_k)_{-s}$ be the matrix obtained by removing the $s$-th row of $U_k$. Then*

$$\mathrm{rank}(\widetilde{U}_k) = \begin{cases} k, & \text{if and only if} \quad \sum_{i=1}^{k} u_i(s)^2 < 1, \\ k-1, & \text{if and only if} \quad \sum_{i=1}^{k} u_i(s)^2 = 1. \end{cases}$$

*Equivalently,*

$$\mathrm{rank}(\widetilde{U}_k) = k \iff e_s \notin \mathrm{span}(U_k) \iff \exists i > k \text{ such that } u_i(s) \neq 0,$$

*and*

$$\mathrm{rank}(\widetilde{U}_k) = k-1 \iff e_s \in \mathrm{span}(U_k) \iff u_{k+1}(s) = \cdots = u_n(s) = 0.$$

*Proof.* Since $R$ deletes the $s$-th row, its nullspace is $\mathrm{span}\{e_s\}$, and $R^\top R = I_n - e_s e_s^\top$. We compute the Gram matrix

$$G := \widetilde{U}_k^\top \widetilde{U}_k = U_k^\top R^\top R U_k = U_k^\top (I_n - e_s e_s^\top) U_k = I_k - v v^\top, \qquad v := U_k^\top e_s \in \mathbb{R}^k.$$

The matrix $G$ is a rank-one perturbation of $I_k$, whose eigenvalues are

$$\mathrm{spec}(G) = \{1 - |v|^2, \underbrace{1, \ldots, 1}_{k-1}\}.$$

Hence $G$ is invertible if and only if $|v|^2 \neq 1$; that is,

$$|v|^2 = \|U_k^\top e_s\|^2 = \sum_{i=1}^k u_i(s)^2 < 1.$$

Since $\sum_{i=1}^n u_i(s)^2 = 1$, we have $0 \leq \sum_{i=1}^k u_i(s)^2 \leq 1$. Therefore $G$ is invertible if and only if $\sum_{i=1}^k u_i(s)^2 < 1$, yielding $\text{rank}(\widetilde{U}_k) = k$ in this case, because a Gram matrix $A^\top A$ is positive definite if and only if the columns of $A$ are linearly independent. In the remaining case $\sum_{i=1}^k u_i(s)^2 = 1$, $G$ has a single zero eigenvalue, so $\text{rank}(G) = k - 1$, and hence $\text{rank}(\widetilde{U}_k) = k - 1$. $\qquad\square$

**Lemma F.17.** *Let $A \in \mathbb{R}^{m_1 \times n}$ and $B \in \mathbb{R}^{m_2 \times n}$. Denote by $S_A, S_B \subseteq [n]$ the index sets of their nonzero columns with $|S_A| = p$ and $|S_B| = q$. Assume*

$$\text{rank}(A) = p - 1, \qquad \text{rank}(B) = q - 1.$$

*Let $J := S_A \cap S_B$ with $s := |J|$, and $S := S_A \cup S_B$ with $u := |S|$. Let $\alpha, \beta \in \mathbb{R}^n$ be (uniquely determined up to a nonzero scalar on their supports) such that*

$$\text{supp}(\alpha) \subseteq S_A, \ A\alpha = 0, \ \alpha|_{S_A} \neq 0, \qquad \text{supp}(\beta) \subseteq S_B, \ B\beta = 0, \ \beta|_{S_B} \neq 0.$$

*Write $\alpha_J, \beta_J \in \mathbb{R}^s$ for their restrictions to $J$, and define*

$$R : \mathbb{R}^2 \to \mathbb{R}^s, \qquad (t, s) \mapsto t\,\alpha_J - s\,\beta_J.$$

*Let $M := \begin{bmatrix} A \\ B \end{bmatrix}$. Then*

$$\text{rank}(M) = u - 2 + \text{rank}(R).$$

*Proof.* Let $C := M[:, S]$ denote the submatrix formed by the $u$ nonzero columns of $M$; clearly $\text{rank}(M) = \text{rank}(C)$. For any $\gamma \in \mathbb{R}^u$,

$$C\gamma = 0 \iff A[:, S]\gamma = 0 \text{ and } B[:, S]\gamma = 0.$$

Since $\text{rank}(A[:, S_A]) = p - 1$ (resp. $\text{rank}(B[:, S_B]) = q - 1$), the column dependencies on $S_A$ (resp. on $S_B$) each form a one-dimensional subspace spanned by $\alpha|_{S_A}$ (resp. $\beta|_{S_B}$). Hence there exist $(t, s) \in \mathbb{R}^2$ such that

$$\gamma|_{S_A} = t\,\alpha|_{S_A}, \qquad \gamma|_{S_B} = s\,\beta|_{S_B}.$$

On the intersection $J = S_A \cap S_B$, these two prescriptions must coincide, i.e. $t\,\alpha_J = s\,\beta_J$, which is equivalent to $R(t, s) = 0$.

Define the linear map

$$\Phi : \ker R \longrightarrow \ker C, \qquad (t, s) \longmapsto \gamma(t, s),$$

where $\gamma(t, s)$ is the unique vector satisfying $\gamma_{S_A} = t\alpha_{S_A}$ and $\gamma_{S_B} = s\beta_{S_B}$. The condition $(t, s) \in \ker R$ ensures that $\gamma(t, s)$ is well defined on the overlap $J$. The map $\Phi$ is linear, injective (since $\alpha|_{S_A}$ and $\beta|_{S_B}$ are nonzero), and surjective (as every $\gamma \in \ker C$ arises from some $(t, s)$ satisfying $R(t, s) = 0$). Hence $\Phi$ is a linear isomorphism, and therefore

$$\dim \ker C = \dim \ker R = 2 - \text{rank}(R).$$

Finally, applying the rank–nullity theorem to $C$ (which has $u$ columns) yields

$$\text{rank}(C) = u - \dim \ker C = u - \big(2 - \text{rank}(R)\big) = u - 2 + \text{rank}(R).$$

Since $\text{rank}(M) = \text{rank}(C)$, the claimed formula follows. $\qquad\square$

**Corollary F.18.** *With the notation above:*

1. *If $s = 0$, then $\text{rank}(R) = 0$ and $\text{rank}(M) = u - 2$.*

2. *If $s = 1$, then* $\mathrm{rank}(R) \in \{0, 1\}$*; in particular,* $\alpha_J = \beta_J = 0 \iff \mathrm{rank}(R) = 0$*, otherwise* $\mathrm{rank}(R) = 1$*, hence* $\mathrm{rank}(M) = u - 1$.

3. *If $s \geq 2$, then:*
   - *if $\alpha_J$ and $\beta_J$ are linearly independent,* $\mathrm{rank}(R) = 2$ *and* $\mathrm{rank}(M) = u$*;*
   - *if they are collinear but not both zero,* $\mathrm{rank}(R) = 1$ *and* $\mathrm{rank}(M) = u - 1$*;*
   - *if $\alpha_J = \beta_J = 0$,* $\mathrm{rank}(R) = 0$ *and* $\mathrm{rank}(M) = u - 2$.

*Proof of Theorem F.12.* Let $U = [u_1, \ldots, u_n]$ be the orthogonal eigenbasis of $L$, and define $M_i = U_{-i} \mathrm{diag}(u_1(i), \ldots, u_n(i))$ as above. The $\ell$-th column of $M_i$ equals $u_\ell(i) (u_\ell)_{-i}$ and vanishes if and only if $u_\ell(i) = 0$. Let $I = (i_1, \ldots, i_K)$ be a minimal column-activating list according to Definition F.14, so that each $M_{i_k}$ introduces at least one previously inactivated column. We denote $K = |I|$ as the length of this activating list.

**Problem Reduction.** We first assume that the graph $G$ has a simple spectrum, so that all eigenvalues of $L$ are distinct. Accordingly, each frequency component can be assigned an independent coefficient, and the spectral response vector $c$ lies in $\mathbb{R}^n$.

For any $c \in \mathbb{R}^n$, recall the spectral convolution matrix

$$C(c) = U \, \mathrm{diag}(c) U^\top = \sum_{\ell=1}^n c_\ell \, u_\ell u_\ell^\top.$$

For each vertex $i$, denote by $u_\ell(i)$ the $i$-th entry of the eigenvector $u_\ell$. Then the $(i, j)$-entry of $C(c)$ can be written explicitly as $C(c)_{ij} = \sum_{\ell=1}^n c_\ell \, u_\ell(i) \, u_\ell(j)$. Fix a vertex $i$, collect all off-diagonal entries of the $i$-th row into a row vector $C(c)_{i,-i} \in \mathbb{R}^{1 \times (n-1)}$. Using the above expression, we obtain

$$C(c)_{i,-i} = \big[ C(c)_{ij} \big]_{j \neq i} = \sum_{\ell=1}^n c_\ell \, u_\ell(i) \, u_{\ell,-i}^\top.$$

Hence we can write

$$\begin{aligned}
C(c)_{i,-i} &= \big[ u_1(i) u_1(-i)^\top, \ u_2(i) u_2(-i)^\top, \ \ldots, \ u_n(i) u_n(-i)^\top \big] c \\
&= U_{-i} \, \mathrm{diag}\big(u_1(i), \ldots, u_n(i)\big) c \\
&= U_{-i} D_i \, c \\
&= M_i \, c.
\end{aligned}$$

Then for any subset $I = (i_1, \ldots, i_s)$,

$$M_I c = \begin{bmatrix} M_{i_1} \\ \vdots \\ M_{i_s} \end{bmatrix} c = \begin{bmatrix} C(c)_{i_1, -i_1} \\ \vdots \\ C(c)_{i_s, -i_s} \end{bmatrix}.$$

Hence $M_I c = 0$ if and only if $C(c)_{ij} = 0$ for all $i \in I$ and $j \neq i$, meaning that all off-diagonal entries of $C(c)$ in those rows vanish.

We now construct a $(K{+}1)$-partition of $V$ as follows: assign the vertices $i_1, \ldots, i_K$ to distinct singleton classes $V_1, \ldots, V_K$, and group all remaining vertices into the $(K{+}1)$-th class $V_{K+1}$. Let $y_i = t$ if and only if $v_i \in V_t$. For this labeling, define the linear operator

$$T : \mathbb{R}^n \longrightarrow \mathbb{R}^{N_T}, \qquad T(c) = (C(c)_{ij})_{y_i \neq y_j},$$

which collects all cross-class entries of $C(c)$ and the codomain dimension $N_T$ equals the number of such pairs. Then

$$\ker(T) = \{ c : C(c)_{ij} = 0 \text{ whenever } y_i \neq y_j \}$$

represents the set of filters that vanish on all cross-class entries. Since $M_I$ encodes a subset of these cross-class constraints, we always have

$$\ker(T) \subseteq \ker(M_I).$$

Moreover, the constant vector $\mathbf{1}$ satisfies $C(\mathbf{1}) = I_n$, hence $\mathbf{1} \in \ker(T)$ and $\mathbf{1} \in \ker(M_I)$. Therefore, to prove Theorem F.12, it suffices to show $\ker(T) = \mathrm{span}\{\mathbf{1}\}$, which in turn follows if we can establish that $\mathrm{rank}(M_I) = n - 1$. We will analyze this rank condition in the following discussion.

**Single block analysis:** For $i \in V$, let $S_i := \{\ell \in [n] : u_\ell(i) \neq 0\}$ be the set of nonzero columns of $M_i$. Restrict to these columns and write

$$M_i[:, S_i] = (U_{-i})_{[:, S_i]} \operatorname{diag}(u_\ell(i))_{\ell \in S_i}.$$

Since the diagonal factor is invertible, $\operatorname{rank}(M_i[:, S_i]) = \operatorname{rank}((U_{-i})_{[:, S_i]})$. According to Proposition F.16, we have

$$\operatorname{rank}(M_i) = |S_i| - 1$$

as $\sum_{\ell \in S_i} u_\ell(i)^2 = 1$ by its definition. Moreover, the *unique* (up to scale) column dependency of $M_i[:, S_i]$ is the constant vector on $S_i$: indeed,

$$\sum_{\ell \in S_i} 1 \cdot (u_\ell(i)(u_\ell)_{-i}) = \Big( \sum_{\ell \in S_i} u_\ell(i) u_\ell \Big)_{-i} = (e_i)_{-i} = 0,$$

so the kernel of $M_i[:, S_i]$ is spanned by $\mathbf{1}_{S_i}$. We will denote this support–restricted kernel vector by $\alpha^{(i)} = \mathbf{1}_{S_i}$.

**Two–block stacking:** Let $A := M_i$ and $B := M_j$ with supports $S_A = S_i$ and $S_B = S_j$, and let $J := S_A \cap S_B$. By single block analysis, each block has a one–dimensional support–restricted kernel generated by $\alpha = \mathbf{1}_{S_A}$ and $\beta = \mathbf{1}_{S_B}$, respectively. On the overlap $J$ we have

$$\alpha_J = \mathbf{1}_J, \qquad \beta_J = \mathbf{1}_J,$$

so $\alpha_J$ and $\beta_J$ are *collinear and nonzero* whenever $J \neq \varnothing$. Applying corollary F.18, with $u = |S_A \cup S_B|$ and $R : \mathbb{R}^2 \to \mathbb{R}^{|J|}$, $(t, s) \mapsto t\,\alpha_J - s\,\beta_J$, we obtain

$$\operatorname{rank}\begin{bmatrix} A \\ B \end{bmatrix} = u - 2 + \operatorname{rank}(R).$$

Since $\alpha_J$ and $\beta_J$ are collinear and not both zero when $J \neq \varnothing$, we get

$$\operatorname{rank}(R) = 1 \quad \text{whenever } J \neq \varnothing.$$

Therefore,

$$\operatorname{rank}\begin{bmatrix} M_i \\ M_j \end{bmatrix} = |S_i \cup S_j| - 1 \quad \text{whenever } S_i \cap S_j \neq \varnothing.$$

We claim $S_i \cap S_j \neq \varnothing$ is always true. For a connected graph Laplacian, either combinatorial or symmetric normalized, the eigenvector corresponding to $\lambda = 0$ is $u_1 \propto \mathbf{1}$ or $u_1 \propto D^{1/2}\mathbf{1}$ respectively, whose entries are all nonzero. Hence $1 \in S_i$ for every $i \in V$, so $S_i \cap S_j$ *always* contains the index 1. Consequently, in the Laplacian setting considered in Theorem F.12, we indeed have $\operatorname{rank}(R) = 1$ for every pair of blocks.

**Multi–block induction:** Let $I = (i_1, \ldots, i_K)$ be any minimal column–activating list and define $S^{(r)} := \bigcup_{t=1}^r S_{i_t}$. By induction with the two–block formula above,

$$\operatorname{rank}\begin{bmatrix} M_{i_1} \\ \vdots \\ M_{i_r} \end{bmatrix} = |S^{(r)}| - 1 \quad (r = 1, 2, \ldots, K),$$

because at each step the overlap is nonempty (it contains the constant eigenvector's index). By minimal activation, $S^{(K)} = [n]$, and hence

$$\operatorname{rank}(M_I) = n - 1.$$

**Non-simple spectra and larger partitions:** The above reasoning does not rely on the spectrum being simple. Let

$$W = \bigoplus_{\lambda \in \operatorname{spec}(L)} \operatorname{span}\{\mathbf{1}_{m_\lambda}\} = \Big\{ c = \sum_\lambda \alpha_\lambda \mathbf{1}_{m_\lambda} \Big\}.$$

Since $\ker(T) = \operatorname{span}\{\mathbf{1}\}$ for all $c \in \mathbb{R}^n$, we have $\ker(T) \cap W = \operatorname{span}\{\mathbf{1}\}$. Hence $c_\lambda \equiv \alpha$ for all $\lambda$, and thus $C(c) = \alpha I$.

Moreover, once the minimal activating family $I$ of length $K$ is found, any larger partition with $k > K$ that refines this structure (i.e. containing all vertices of $I$ in separate classes) will also satisfy the same property, since stacking additional $M_i$ cannot decrease the rank of $M_I$. $\qquad \square$

*Remark* F.19. In practice, for many real-world graphs the minimal activating length $K$ is often $K = 1$, since there typically exists at least one vertex $i$ whose corresponding Laplacian eigenvector coordinates $\{u_\ell(i)\}_{\ell=1}^n$ are all nonzero. In such cases, the bipartition obtained by isolating this single vertex already achieves $\mathrm{rank}(M_I) = n - 1$ and thus suffices to guarantee $C(c) = \alpha I_n$.

**Corollary F.20.** *Assume that at least one class contains more than one node. Then the optimal convolution $C^\star$ lies at a positive distance from the spectral convolution subspace $\mathcal{S} = \mathrm{span}\{E_\lambda : \lambda \in \mathrm{spec}(L)\}$.*

*Proof.* Under this assumption, $C^\star$ has the block-diagonal form

$$(C^\star)_{aa} = \frac{1}{n_a + \tau_a}, \qquad (C^\star)_{ab} = 0 \ (b \neq a),$$

and hence all inter-class blocks vanish. Since at least one block has size greater than one, $C^\star$ is not a scalar multiple of the identity. By Theorem F.12, any spectral convolution matrix $C(g) = \sum_\lambda g(\lambda)E_\lambda$ with vanishing inter-class entries must collapse to a scalar multiple of the identity. Therefore no choice of coefficients $g(\lambda)$ can yield $C(g) = C^\star$, i.e. the system of equations $C(g) = C^\star$ has no solution. As $\mathcal{S}$ is a finite-dimensional (hence closed) subspace of the matrix space $\mathbb{R}^{n \times n}$, a point outside it must have a strictly positive distance to it. Consequently, $\mathrm{dist}(C^\star, \mathcal{S}) > 0$. $\qquad\square$

*Remark* F.21. Corollary F.20 implies that the spectral convolution family cannot represent the optimal convolution $C^\star$, nor can it approximate it arbitrarily closely. In fact, this limitation is robust: even if one perturbs $C^\star$ slightly within the class of block-diagonal matrices with vanishing inter-class entries, no spectral convolution can realize or approximate these perturbed operators either. The reason is structural rather than numerical: according to Theorem F.12, as long as at least one class contains more than one node and the inter-class interference is suppressed ($C_{ab} = 0$ for $a \neq b$), any nontrivial intra-class variation leads to matrices that fall outside the spectral subspace. This shows that the expressive capacity of spectral convolutions is severely constrained by their diagonal structure in the spectral domain.

## G. Experimental Details

All experiments are run on a GPU NVIDIA RTX A6000 with 48G memory.

### G.1. Heterophilic Node Classification

**Datasets** The dataset statistics, including the numbers of nodes and edges, feature dimensionality, number of node classes, and edge homophily ratios, are summarized in Table 8.

*Table 8.* Statistics of heterophilic (Pei et al., 2020; Rozemberczki et al., 2021; Platonov et al., 2023).

| Statistics | Texas | Wisconsin | Chameleon | Squirrel | Roman_Empire | Minesweeper | Tolokers | Questions |
|---|---|---|---|---|---|---|---|---|
| # Nodes | 183 | 251 | 890 | 2,223 | 22,662 | 10,000 | 11,758 | 48,921 |
| # Edges | 295 | 466 | 27,168 | 131,436 | 32,927 | 39,402 | 519,000 | 153,540 |
| # Features | 1,703 | 1,703 | 2,325 | 2,089 | 300 | 7 | 10 | 301 |
| # Classes | 5 | 5 | 5 | 5 | 18 | 2 | 2 | 2 |
| # Edge Homophily | 0.11 | 0.21 | 0.24 | 0.22 | 0.05 | 0.68 | 0.59 | 0.84 |

**Hyperparameter Setting** For all datasets, we train for at most $1{,}000$ epochs with early stopping (patience $= 200$). On the four large heterophilic datasets (Roman Empire, Tolokers, Minesweeper, Questions), we fix the MLP hidden size to $512$ and the polynomial order to $5$. For the remaining datasets, we use a hidden size of $64$ and set the polynomial order to $2$.

We perform a grid search over the training hyperparameters of FSPECGNN, including the dropout rate, learning rate, weight decay, and the number of GAT heads. The corresponding search ranges are reported in Table 9.

**Time and Space Overhead** We report the average runtime per run and the peak GPU memory usage for each model and each task in Table 10 and Table 11.

*Table 9.* Hyperparameter search ranges. $S(\cdot)$ is the Sigmoid function.

| Hyper-parameters | Range |
|---|---|
| dropout in MLP | 0.1, 0.3, 0.5, 0.7, 0.9 |
| dropout after MLP | 0.1, 0.3, 0.5, 0.7, 0.9 |
| learning rate of $W$ | 0.005, 0.01, 0.02 |
| weight decay of $W$ | 0.0, 0.0001, 0.0005 |
| learning rate for Propagation | 0.005, 0.01, 0.02 |
| weight decay for Propagation | 0.0, 0.0001, 0.0005 |
| number of GAT heads | 1, 2, 4 |
| initial value of $\alpha$ | S(-6), S(-4), S(-2), S(0) |

*Table 10.* Average runtime per run in seconds on heterophilic datasets. "–" indicates out-of-memory.

| Model | Texas | Wisconsin | Chameleon | Squirrel | Roman | Minesweeper | Tolokers | Questions |
|---|---|---|---|---|---|---|---|---|
| GPRGNN | 1.06 | 0.90 | 1.05 | 0.95 | 0.99 | 1.19 | 1.30 | 1.55 |
| JacobiConv | 1.10 | 1.13 | 1.08 | 1.09 | 1.24 | 1.98 | 1.23 | 1.65 |
| Subgraph GNN | 15.70 | 33.16 | – | – | – | – | – | – |
| Local 2-GNN | 23.20 | 51.24 | – | – | – | – | – | – |
| ChebNet | 0.94 | 0.97 | 1.51 | 1.93 | 1.66 | 1.90 | 2.32 | 2.45 |
| **FSPECGNN(Cheb)** | 2.04 | 2.10 | 2.55 | 2.51 | 3.88 | 3.83 | 3.96 | 3.40 |
| ChebNetII | 2.07 | 1.90 | 2.55 | 2.73 | 2.96 | 3.07 | 3.63 | 3.49 |
| **FSPECGNN(ChebII)** | 2.41 | 2.78 | 3.17 | 3.27 | 4.31 | 4.34 | 4.80 | 5.16 |
| BernNet | 2.21 | 2.15 | 2.21 | 2.24 | 2.53 | 3.25 | 3.85 | 4.13 |
| **FSPECGNN(Bern)** | 2.17 | 2.48 | 2.69 | 3.35 | 3.05 | 3.52 | 4.66 | 6.01 |

*Table 11.* GPU memory usage (MiB) on heterophilic datasets. "–" indicates out-of-memory.

| Model | Texas | Wisconsin | Chameleon | Squirrel | Roman | Minesweeper | Tolokers | Questions |
|---|---|---|---|---|---|---|---|---|
| GPRGNN | 646 | 648 | 652 | 688 | 954 | 750 | 876 | 1160 |
| JacobiConv | 648 | 648 | 652 | 672 | 982 | 752 | 876 | 1172 |
| Subgraph GNN | 1724 | 2896 | – | – | – | – | – | – |
| Local 2-GNN | 1830 | 3128 | – | – | – | – | – | – |
| ChebNet | 534 | 534 | 558 | 594 | 864 | 654 | 742 | 1150 |
| **FSPECGNN(Cheb)** | 646 | 648 | 692 | 760 | 1130 | 754 | 976 | 1242 |
| ChebNetII | 646 | 648 | 660 | 716 | 954 | 750 | 914 | 1230 |
| **FSPECGNN(ChebII)** | 650 | 658 | 698 | 752 | 1062 | 756 | 998 | 1244 |
| BernNet | 648 | 648 | 658 | 698 | 1058 | 756 | 954 | 1254 |
| **FSPECGNN(Bern)** | 652 | 658 | 694 | 758 | 1102 | 774 | 1040 | 1276 |

*Table 12.* Node classification results on heterophilic datasets, where all baseline results are re-run.

| Model | Texas | Wisconsin | Chameleon | Squirrel | Roman | Minesweeper | Tolokers | Questions |
|---|---|---|---|---|---|---|---|---|
| ChebNet | $35.09_{\pm 9.71}$ | $33.00_{\pm 8.96}$ | $25.29_{\pm 7.12}$ | $19.64_{\pm 1.99}$ | $51.71_{\pm 0.29}$ | $82.71_{\pm 0.59}$ | $70.94_{\pm 1.14}$ | $60.11_{\pm 0.86}$ |
| ChebNetII | $43.29_{\pm 10.29}$ | $43.17_{\pm 10.08}$ | $28.73_{\pm 4.10}$ | $27.34_{\pm 4.15}$ | $55.35_{\pm 0.22}$ | $78.47_{\pm 0.13}$ | $70.81_{\pm 0.45}$ | $60.68_{\pm 0.37}$ |
| GPRGNN | $50.98_{\pm 3.64}$ | $47.54_{\pm 5.63}$ | $26.71_{\pm 6.98}$ | $20.05_{\pm 3.92}$ | $56.58_{\pm 0.41}$ | $\mathbf{88.38}_{\pm 0.14}$ | $68.66_{\pm 1.16}$ | $70.73_{\pm 0.35}$ |
| JacobiConv | $46.47_{\pm 3.76}$ | $41.83_{\pm 7.21}$ | $27.02_{\pm 3.86}$ | $24.19_{\pm 5.61}$ | $55.88_{\pm 0.17}$ | $86.38_{\pm 0.14}$ | $70.28_{\pm 0.19}$ | $60.96_{\pm 0.75}$ |
| BernNet | $50.58_{\pm 4.68}$ | $49.46_{\pm 6.58}$ | $28.90_{\pm 3.77}$ | $23.68_{\pm 1.98}$ | $\mathbf{58.39}_{\pm 0.26}$ | $78.18_{\pm 0.10}$ | $68.55_{\pm 0.69}$ | $70.32_{\pm 0.27}$ |
| **FSPECGNN(Cheb)** | $55.32_{\pm 4.57}$ | $49.87_{\pm 8.29}$ | $33.09_{\pm 0.92}$ | $\mathbf{39.57}_{\pm 0.67}$ | $54.35_{\pm 0.73}$ | $\underline{88.30}_{\pm 0.82}$ | $\mathbf{76.89}_{\pm 0.91}$ | $75.87_{\pm 0.37}$ |
| **FSPECGNN(ChebII)** | $\mathbf{57.05}_{\pm 3.20}$ | $\underline{50.00}_{\pm 5.42}$ | $\mathbf{39.60}_{\pm 2.77}$ | $\underline{37.70}_{\pm 0.63}$ | $\underline{56.26}_{\pm 1.18}$ | $84.25_{\pm 0.65}$ | $\underline{76.37}_{\pm 0.67}$ | $\underline{77.00}_{\pm 0.37}$ |
| **FSPECGNN(Bern)** | $\underline{56.13}_{\pm 0.46}$ | $\mathbf{54.58}_{\pm 9.47}$ | $\underline{37.91}_{\pm 3.94}$ | $37.59_{\pm 1.32}$ | $56.16_{\pm 1.19}$ | $84.17_{\pm 1.01}$ | $74.50_{\pm 0.69}$ | $\mathbf{77.11}_{\pm 0.26}$ |

**Reproduced Baselines** For fair comparison, we re-ran all baselines under the same experimental protocols as Liu & Wang (2025), including identical data splits, preprocessing procedures, random seeds, training budgets, and evaluation metrics. The reproduced baseline results and our results are summarized in Table 12.

## G.2. GNN Expressivity

For all tasks, we use the distance-encoding hyperparameter `max_dis= 5`. Since spectral models can achieve long-range propagation with fewer layers than spatial models, we use depth $L = 2$ with hidden dimension $d = 150$ for most tasks. For edge-level homomorphism-counting tasks, we use $L = 3$ and $d = 128$ to match the parameter budget. The parameter counts are reported in the table 13.

*Table 13.* Model size in different tasks.

| Task | #Hidden dim | #Parameters |
|---|---|---|
| MPNN | 128 | 314,119 |
| Subgraph GNN | 128 | 314,759 |
| Local 2-GNN | 96 | 317,388 |
| Local 2-FGNN | 96 | 317,388 |
| FSPECGNN | 150 | 319,204 |

**Time and Space Overhead** We report the average iteration time and the peak GPU memory usage for each task in Table 15. We also provide a visual mapping between task names and their corresponding substructures in Table 14.

*Table 14.* Reference of task names and their corresponding substructures.

*(a)* Homomorphism count tasks.

| Name | chordal4 | boat | chordal6 | chordal4_1 | chordal4_4 | chordal5_13 | chordal5_31 | chordal5_24 |
|---|---|---|---|---|---|---|---|---|
| Substructure |  |  |  |  |  |  |  |  |

*(b)* (Chordal) cycle count tasks.

| Name | cycle3 | cycle4 | cycle5 | cycle6 | chordal4 | chordal5 |
|---|---|---|---|---|---|---|
| Substructure |  |  |  |  |  |  |

*Table 15.* Average iteration time and peak GPU memory per model–task pair.

| Model+Task | Avg. Time (s/it) | Max Mem (MB) | Model+Task | Avg. Time (s/it) | Max Mem (MB) |
|---|---|---|---|---|---|
| boat(g)(MPNN) | 6.180 | 6730 | chordal5_13(e)(MPNN) | 6.160 | 9688 |
| boat(g)(Subgraph) | 6.230 | 11228 | chordal5_13(e)(Subgraph) | 6.288 | 9692 |
| boat(g)(Local 2-GNN) | 7.638 | 11532 | chordal5_13(e)(Local 2-GNN) | 7.492 | 11468 |
| boat(g)(Local 2-FGNN) | 4.894 | 9512 | chordal5_13(e)(Local 2-FGNN) | 4.865 | 19886 |
| boat(g)(FSPECGNN) | 1.110 | 7686 | chordal5_13(e)(FSPECGNN) | 1.173 | 9532 |
| chordal4(e)(MPNN) | 6.232 | 9688 | chordal5_24(e)(MPNN) | 6.279 | 9688 |
| chordal4(e)(Subgraph) | 6.110 | 9692 | chordal5_24(e)(Subgraph) | 6.106 | 9692 |
| chordal4(e)(Local 2-GNN) | 7.484 | 11468 | chordal5_24(e)(Local 2-GNN) | 7.559 | 11468 |
| chordal4(e)(Local 2-FGNN) | 4.888 | 19886 | chordal5_24(e)(Local 2-FGNN) | 4.874 | 19886 |
| chordal4(e)(FSPECGNN) | 1.180 | 9532 | chordal5_24(e)(FSPECGNN) | 1.192 | 9532 |
| chordal4(g)(MPNN) | 6.162 | 6730 | chordal5_31(e)(MPNN) | 6.106 | 9688 |
| chordal4(g)(Subgraph) | 5.977 | 11228 | chordal5_31(e)(Subgraph) | 6.220 | 9692 |
| chordal4(g)(Local 2-GNN) | 7.764 | 11532 | chordal5_31(e)(Local 2-GNN) | 7.658 | 11468 |
| chordal4(g)(Local 2-FGNN) | 4.812 | 9512 | chordal5_31(e)(Local 2-FGNN) | 4.856 | 19886 |
| chordal4(g)(FSPECGNN) | 1.097 | 7686 | chordal5_31(e)(FSPECGNN) | 1.172 | 9532 |
| chordal4(v)(MPNN) | 6.125 | 6730 | chordal6(g)(MPNN) | 6.090 | 6730 |
| chordal4(v)(Subgraph) | 6.176 | 11228 | chordal6(g)(Subgraph) | 6.274 | 11228 |
| chordal4(v)(Local 2-GNN) | 7.483 | 11532 | chordal6(g)(Local 2-GNN) | 7.593 | 11532 |
| chordal4(v)(Local 2-FGNN) | 4.758 | 9512 | chordal6(g)(Local 2-FGNN) | 4.791 | 9512 |
| chordal4(v)(FSPECGNN) | 1.082 | 7686 | chordal6(g)(FSPECGNN) | 1.071 | 7686 |
| chordal4_1(g)(MPNN) | 6.009 | 6730 | cycle3(e)(MPNN) | 6.179 | 9688 |
| chordal4_1(g)(Subgraph) | 6.141 | 11228 | cycle3(e)(Subgraph) | 6.121 | 9692 |
| chordal4_1(g)(Local 2-GNN) | 7.686 | 11532 | cycle3(e)(Local 2-GNN) | 7.588 | 11468 |
| chordal4_1(g)(Local 2-FGNN) | 4.894 | 9512 | cycle3(e)(Local 2-FGNN) | 4.899 | 19886 |
| chordal4_1(g)(FSPECGNN) | 1.101 | 7686 | cycle3(e)(FSPECGNN) | 1.029 | 9532 |
| chordal4_1(v)(MPNN) | 6.215 | 6730 | cycle3(g)(MPNN) | 6.114 | 6730 |
| chordal4_1(v)(Subgraph) | 6.174 | 11228 | cycle3(g)(Subgraph) | 6.245 | 11228 |
| chordal4_1(v)(Local 2-GNN) | 7.419 | 11532 | cycle3(g)(Local 2-GNN) | 7.126 | 11532 |
| chordal4_1(v)(Local 2-FGNN) | 4.727 | 9512 | cycle3(g)(Local 2-FGNN) | 5.115 | 9512 |
| chordal4_1(v)(FSPECGNN) | 1.077 | 7686 | cycle3(g)(FSPECGNN) | 1.102 | 7686 |
| chordal4_4(v)(MPNN) | 6.019 | 6730 | cycle3(v)(MPNN) | 6.344 | 6730 |
| chordal4_4(v)(Subgraph) | 6.131 | 11228 | cycle3(v)(Subgraph) | 6.199 | 11228 |
| chordal4_4(v)(Local 2-GNN) | 7.530 | 11532 | cycle3(v)(Local 2-GNN) | 7.600 | 11532 |
| chordal4_4(v)(Local 2-FGNN) | 4.808 | 9512 | cycle3(v)(Local 2-FGNN) | 4.814 | 9512 |
| chordal4_4(v)(FSPECGNN) | 1.080 | 7686 | cycle3(v)(FSPECGNN) | 1.065 | 7686 |
| chordal5(e)(MPNN) | 6.133 | 9688 | cycle4(e)(MPNN) | 6.255 | 9688 |
| chordal5(e)(Subgraph) | 6.158 | 9692 | cycle4(e)(Subgraph) | 6.200 | 9692 |
| chordal5(e)(Local 2-GNN) | 7.579 | 11468 | cycle4(e)(Local 2-GNN) | 7.689 | 11468 |
| chordal5(e)(Local 2-FGNN) | 4.839 | 19886 | cycle4(e)(Local 2-FGNN) | 4.867 | 19886 |
| chordal5(e)(FSPECGNN) | 1.184 | 9532 | cycle4(e)(FSPECGNN) | 1.157 | 9532 |
| chordal5(g)(MPNN) | 6.113 | 6730 | cycle4(g)(MPNN) | 6.083 | 6730 |
| chordal5(g)(Subgraph) | 6.186 | 11228 | cycle4(g)(Subgraph) | 6.146 | 11228 |
| chordal5(g)(Local 2-GNN) | 7.615 | 11532 | cycle4(g)(Local 2-GNN) | 7.671 | 11532 |
| chordal5(g)(Local 2-FGNN) | 4.816 | 9512 | cycle4(g)(Local 2-FGNN) | 4.859 | 9512 |
| chordal5(g)(FSPECGNN) | 1.099 | 7686 | cycle4(g)(FSPECGNN) | 1.034 | 7686 |
| chordal5(v)(MPNN) | 6.249 | 6730 | cycle4(v)(MPNN) | 6.226 | 6730 |
| chordal5(v)(Subgraph) | 6.149 | 11228 | cycle4(v)(Subgraph) | 6.050 | 11228 |
| chordal5(v)(Local 2-GNN) | 7.561 | 11532 | cycle4(v)(Local 2-GNN) | 7.680 | 11532 |
| chordal5(v)(Local 2-FGNN) | 4.864 | 9512 | cycle4(v)(Local 2-FGNN) | 4.862 | 9512 |
| chordal5(v)(FSPECGNN) | 1.105 | 7686 | cycle4(v)(FSPECGNN) | 1.011 | 7686 |
| cycle5(e)(MPNN) | 6.110 | 9688 | cycle5(e)(Subgraph) | 6.111 | 9692 |
| cycle5(e)(Local 2-GNN) | 7.460 | 11468 | cycle5(e)(Local 2-FGNN) | 4.901 | 19886 |
| cycle5(e)(FSPECGNN) | 1.173 | 9532 | cycle5(g)(MPNN) | 6.193 | 6730 |
| cycle5(g)(Subgraph) | 6.011 | 11228 | cycle5(g)(Local 2-GNN) | 7.569 | 11532 |
| cycle5(g)(Local 2-FGNN) | 4.872 | 9512 | cycle5(g)(FSPECGNN) | 1.097 | 7686 |
| cycle5(v)(MPNN) | 6.157 | 6730 | cycle5(v)(Subgraph) | 6.130 | 11228 |
| cycle5(v)(Local 2-GNN) | 7.605 | 11532 | cycle5(v)(Local 2-FGNN) | 4.858 | 9512 |
| cycle5(v)(FSPECGNN) | 1.101 | 7686 | cycle6(e)(MPNN) | 6.106 | 9688 |
| cycle6(e)(Subgraph) | 6.086 | 9692 | cycle6(e)(Local 2-GNN) | 7.565 | 11468 |
| cycle6(e)(Local 2-FGNN) | 4.825 | 19886 | cycle6(e)(FSPECGNN) | 1.198 | 9532 |
| cycle6(g)(MPNN) | 6.210 | 6730 | cycle6(g)(Subgraph) | 6.162 | 11228 |
| cycle6(g)(Local 2-GNN) | 7.622 | 11532 | cycle6(g)(Local 2-FGNN) | 4.916 | 9512 |
| cycle6(g)(FSPECGNN) | 1.085 | 7686 | cycle6(v)(MPNN) | 6.164 | 6730 |
| cycle6(v)(Subgraph) | 5.991 | 11228 | cycle6(v)(Local 2-GNN) | 7.548 | 11532 |
| cycle6(v)(Local 2-FGNN) | 4.776 | 9512 | cycle6(v)(FSPECGNN) | 1.032 | 7686 |

