# OpenReview forum: "Full-Spectrum Graph Neural Networks: Expressive and Scalable"
_ICML.cc/2026/Conference — ICML 2026 regular_

### Official Review · Reviewer_6TLg · 2026-03-09

**Soundness:** 3
**Presentation:** 3
**Significance:** 3
**Originality:** 4
**Overall Recommendation:** 5
**Confidence:** 4

**Summary:**

This paper proposes Full Spectrum Graph Neural Network (FSPECGNN), a second-order generalization of spectral GNNs that operates on node-pair signals in $V \times V$ and replaces the classical univariate spectral response $g(\lambda_i)$ with a bivariate response $g(\lambda_i,\lambda_j)$, enabling off-diagonal interactions between different Laplacian eigenspaces. The paper motivates why purely diagonal spectral operators are limited for certain class-separation operators and argues that cross-eigenspace coupling is especially relevant under heterophily. For scalability, the paper presents practical implementations that avoid explicit computations on $V \times V$. In the main instantiation, the operator is implemented in a low-rank form that can be written as $h(L)\,\varepsilon\,f(L)$, where $\varepsilon = I + \alpha\,\mathrm{GAT}$ induces feature-dependent mixing across frequencies. Experiments include heterophilic node classification on eight benchmarks under a sparse $2.5/2.5/95$ split, and two expressivity-motivated regression suites (homomorphism counting and cycle counting), along with ablations and efficiency results.

**Compliance With Llm Reviewing Policy:**

Affirmed.

**Final Justification:**

My final recommendation remains Accept. The paper is original and technically solid, with a clear and meaningful full-spectrum spectral formulation and a practically relevant low-rank implementation. My main concerns were baseline comparability, scalability evidence, and some implementation details affecting reproducibility. The rebuttal addressed these points clearly and increased my confidence in the empirical evaluation and clarity of the setup. It did not change my overall assessment, but it reinforced my prior view. Overall, I maintain my recommendation of Accept.

**Key Questions For Authors:**

1. For the heterophilic node classification baselines in Table~1, were the baseline results re-run under your exact preprocessing and the $2.5/2.5/95$ split, or taken directly from the cited prior work?
If the baselines were fully re-run and matched, I will be more confident the reported gains reflect the proposed mechanism and would slightly increase my confidence; otherwise I would reduce confidence in the magnitude of the improvements.

2. In the implementation with $\varepsilon = I + \alpha\,\mathrm{GAT}$, is $\varepsilon$ recomputed during training as part of the forward pass, and is $\alpha$ shared across layers or layer-specific?
A clear and precise answer improves reproducibility and would increase my presentation score and confidence; ambiguity here would lower my confidence in the experimental claims.

3. In the experimental setup, what values are used for the low-rank parameter $S$, and are the polynomial degrees and filter families for $f$ and $h$ chosen independently or tied?
If these choices are fixed and consistently specified, it strengthens the practicality claim; if results depend on under-specified choices, I would lower the presentation score and confidence.

**Limitations:**

yes

**Strengths And Weaknesses:**

A clear strength is the conceptual contribution: the paper gives a clean spectral formulation that explicitly targets a limitation of diagonal spectral filtering by introducing a learnable $g(\lambda_i,\lambda_j)$, and the proposed low-rank realization is simple and keeps the same asymptotic order as polynomial spectral layers while introducing nontrivial cross-frequency interactions. The experimental design is reasonably aligned with the stated goals: counting tasks are used as controlled expressivity checks rather than only SOTA chasing, and ablations indicate that the off-diagonal interaction and the in-filter contribute on representative heterophilic datasets; the efficiency table on counting tasks also suggests practical benefits compared to the included second-order spatial baselines. The main weaknesses are that the heterophilic node classification comparisons are largely scoped to spectral baselines, with many baseline numbers attributed to prior work rather than clearly demonstrated under fully matched preprocessing and splits, and that the strongest scalability evidence is reported on the counting benchmarks while the largest node-classification datasets would benefit from direct runtime and memory reporting to substantiate scalability in that regime.

---

> ### Author Rebuttal · Authors · 2026-03-30
>
> We sincerely thank the reviewer for the positive feedback and helpful comments. We reply to the comments in detail and hope to address your concerns.
>
> ### **W1+Q1: Experimental Protocol and Baseline Coverage**
>
> **A1**: Thank you for raising this important concern. We clarify our evaluation protocol and provide additional verification below.
>
> - **Original source of the baseline results.**
>   We strictly followed the protocol of the original benchmark paper, including data loading, splits, preprocessing, and the same 10 random seeds with the same injection stage. Therefore, the baseline results in Table.1 in our paper were initially taken from the original benchmark paper [1].
>
> - **Additional re-runs for verification.**
>   Following your suggestion, we additionally re-ran the baseline models under the original benchmark protocol. Specifically, we used the same split function, preprocessing pipeline, and random seeds, and performed hyperparameter search within the original search range. The resulting numbers are reported in Table.1 of the **[Supplementary Rebuttal Material: Rerun](https://anonymous.4open.science/r/Spec1-C422/rerun.jpg)**.
>
> - **Additional spatial baselines.**
> In addition, following your suggestion to broaden the comparison beyond spectral baselines, we further included two standard spatial GNNs designed for heterophilic graphs, namely H2GCN [2] and ACM-GCN [3]. These models were evaluated under the same split function, graph preprocessing pipeline, and random seeds as above. Their results are also reported in Table.1 of the **[Supplementary Rebuttal Material: Rerun](https://anonymous.4open.science/r/Spec1-C422/rerun.jpg)**.
>
> ### **W2: Runtime and Memory Reports on Large Graphs**
>
> **A2:** We agree that direct runtime and memory reporting is important for substantiating scalability. To address this concern, we report the runtime and GPU memory of all model variants and all spectral baselines across datasets; see Table.1 and Table.2 of the **[Supplementary Rebuttal Material: Runtime and GPU Memory](https://anonymous.4open.science/r/Spec1-C422/runtime_GPUmemory.jpg)**. These results show that FSpecGNN has runtime and memory usage comparable to existing spectral baselines, suggesting that our low-rank approximation is effective in making the full-spectrum model scalable in practice.
>
> Under the same experimental protocol, we also report the runtime and GPU memory of the spatial higher-order methods, Subgraph GNN and Local 2-GNN. Both run out of memory on the large-scale node-classification datasets, indicating that such methods are limited to small-scale graphs.
>
> ### **Q2: Clarification on $\epsilon$ and $\alpha$**
> **A3:**
> In our implementation, $\epsilon$ is recomputed at every forward pass during training. Regarding $\alpha$, it is layer-specific rather than shared across layers. In fact, in the models used in our experiments, there is only one `GATConv` layer, so each model has only a single $\alpha$, associated with that unique layer. For a more detailed implementation of our models, please refer to the Algorithm 1 in the **[Algorithm Box](https://anonymous.4open.science/r/Spec1-C422/Algo_box.jpg)**.
>
> ### **Q3: Heperparameter configurations**
>
> **A4:** In our experiments, the low-rank parameter is fixed uniformly as $S=1$ across all datasets. The choices of $f$ and $h$ are tied rather than independent: they use the same filter family and the same polynomial degree. For example, in FSpecGNN(Cheb), both $f$ and $h$ are instantiated as `ChebConv`, and the other variants follow the same rule. We set $\deg(f)=\deg(h)=K$, with $K=2$ for the four small graphs (Texas, Wisconsin, Chameleon, Squirrel) and $K=5$ for the four larger graphs (Roman Empire, Minesweeper, Tolokers, Questions).
>
> ### **References**
> [1] Liu, Fangbing, and Qing Wang. Asymmetric learning for spectral graph neural networks. AAAI, 2025.
>
> [2] Zhu, Jiong, et al. Beyond homophily in graph neural networks: Current limitations and effective designs. NeurIPS, 2020.
>
> [3] Luan, Sitao, et al. Revisiting heterophily for graph neural networks. NeurIPS 2022.

---

> > ### Author Rebuttal · Reviewer_6TLg · 2026-04-01
> >
> > The authors addressed the main concerns raised in my original review, including clarification of the baseline protocol, additional comparison details, runtime/memory evidence, and implementation details. The rebuttal increases my confidence in the empirical evaluation and practical setup of the method. I therefore maintain my overall recommendation of Accept.

---

> > > ### Author Response · Authors · 2026-04-06
> > >
> > > We sincerely thank the reviewer for the constructive comments and positive feedback. We will incorporate the corresponding revisions and additional details into the manuscript.

---

### Official Review · Reviewer_REN8 · 2026-03-12

**Soundness:** 3
**Presentation:** 3
**Significance:** 3
**Originality:** 3
**Overall Recommendation:** 5
**Confidence:** 4

**Summary:**

This paper proposes FSPEC GNN, a second-order generalization of spectral GNNs. Instead of operating only on node signals with univariate filters over Laplacian eigenvalues, the method lifts representations to the node-pair domain and uses bivariate filters over eigenvalue pairs. The paper argues that this full-spectrum construction strictly expands the expressive power of standard spectral GNNs, can exceed Local 2-GNN in expressivity, and is especially useful for heterophilic learning.

**Compliance With Llm Reviewing Policy:**

Affirmed.

**Key Questions For Authors:**

Many spatial methods already operate on node pairs or subgraphs (e.g., Local 2‑GNN, subgraph GNNs), but typically without an explicit spectral characterization. Can FSPECGNN be viewed as a spectral relaxation or completion of certain spatial pairwise architectures? Conversely, are there spatial pairwise models that you believe cannot be captured within your full‑spectrum framework?

Could you comment on any empirical comparisons that specifically target tasks where spatial high‑order GNNs are known to shine (e.g., edge‑level reasoning, certain combinatorial problems), and whether FSPECGNN’s spectral bias provides advantages or disadvantages there?

**Limitations:**

The authors do discuss limitations in the appendix.

**Strengths And Weaknesses:**

Soundness: The theoretical contribution is substantial: the paper carefully positions standard spectral GNNs as a diagonal special case, proves stronger expressivity results, and provides a principled spectral explanation for why heterophily can require second-order structure. The empirical section is aligned with the theory, including node classification, counting tasks, ablations, and runtime and memory analysis.

Presentation: The main idea is original and the exposition generally makes the shift from first-order to second-order spectral filtering understandable. The paper does a good job of tying theory to experiments. A small weakness is that readers unfamiliar with higher-order spectral constructions may find the implementation pathway somewhat hard to follow; an additional high-level algorithm box would improve accessibility.

Significance: The significance is high for the spectral GNN literature. The paper offers both a conceptual advance, by reframing spectral expressivity through higher-order signal domains, and a practical contribution, by showing scalable approximations that keep the method usable. The focus on heterophily and counting tasks is appropriate because those are precisely settings where stronger expressivity matters.

Originality: The move from diagonal spectral filters on node signals to full-spectrum bivariate filters on node-pair signals is a meaningful conceptual leap, and the heterophily analysis provides a compelling perspective that standard spectral filtering is structurally limited. This is not just an incremental variant of existing spectral methods.

---

> ### Author Rebuttal · Authors · 2026-03-30
>
> Thank you for your supportive and constructive comments. We are encouraged that you recognized the connection between our theoretical development and empirical evaluation, as well as the value of our theoretical analysis.
>
> ### **W1: Accessibility of the Implementation Pathway**
>
> We agree that a high-level algorithmic summary improves accessibility. We have therefore added pseudocodes for both implementations, i.e. the heterophilic node classification pipeline and the counting-task pipeline (Algorithms 1 and 2 in **[Algorithm box](https://anonymous.4open.science/r/Spec1-C422/Algo_box.jpg)**). We hope this makes the implementation pathway clearer.
>
>
> ### **Q1: Connection to Spatial Methods**
> At a high level, FSpecGNN can be viewed as a spectral parameterization of a certain class of second-order spatial propagation rules:
>
> - Both FSpecGNN and Local 2-GNN operate on the pair domain and propagate information through pairwise neighborhoods, i.e., from $(u,v)$ to $(u,v')$ with $v' \in N(v)$ (and symmetrically on the other coordinate).
>
> - The key difference is that Local 2-GNN performs this propagation through explicit local aggregation, so one layer corresponds to one-hop propagation, whereas FSpecGNN controls its receptive field through the order of the spectral filter.
>
> In this sense, FSpecGNN is better viewed as a more global and flexible spectral parameterization of related pair-domain propagation mechanisms.
>
> However, FSpecGNN can only characterize a restricted class of spatial second-order architectures. A representative counterexample is **Local 2-FGNN** [1]. Its pairwise update uses richer local second-order structure than FSpecGNN, and this finer-grained local information is not naturally describable within our second-order spectral filtering framework. Therefore, our framework should not be viewed as a universal spectral reformulation of all spatial pairwise models. This is also consistent with our counting-task results, where Local 2-FGNN performs better than FSpecGNN.
>
> ### **Q2: Advantage/disadvantage of Spectral Bias**
> We believe our counting benchmarks are well aligned with this regime. They are inherently combinatorial tasks, and they also require edge/pair-level reasoning, since second-order models form intermediate node-pair representations and obtain node/graph-level outputs by pooling over these pairwise signals.
>
> On such tasks, FSpecGNN’s spectral bias has both disadvantages and advantages.
>
> On the downside, spatial higher-order models such as Local 2-FGNN can exploit richer local higher-order structure, and are therefore often better suited to strongly local combinatorial reasoning. This is reflected in our experiments: on the counting benchmarks, FSpecGNN does not outperform Local 2-FGNN.
>
> On the upside, the spectral formulation is substantially more **efficient**. As discussed in Section 5.4, FSpecGNN can realize a larger and tunable receptive field, whereas its spatial counterparts typically require more layers to reach a similar range. For example, in our counting experiments, a two-layer FSpecGNN already reaches a similar performance level to five-layer Subgraph GNN and Local 2-GNN baselines.
>
> A second advantage is **interpretability**: the learned spectral response $g(\lambda_i,\lambda_j)$ directly shows how the model weights different frequency pairs. In our cycle-counting experiments, we observe that as the task becomes more challenging, the learned response departs from a relatively uniform pattern in which the diagonal and off-diagonal regions behave similarly, and becomes more genuinely two-dimensional, suggesting that off-diagonal frequency interactions become increasingly important.
>
> Concrete examples are shown in the **[Visualization: 3-cycle and 6-cycle](https://anonymous.4open.science/r/Spec1-C422/vis-spectral-response.pdf)**, which shows the learned spectral responses for the cycle-counting tasks in Table 2 in our paper. For relatively easier tasks such as graph/node-level 3-cycle counting, the responses remain similar to diagonal; for harder tasks such as 6-cycle counting, the responses become more twisted, indicating that diagonal behavior alone is no longer sufficient to characterize the learned filter. Here, layer 0 and layer 1 correspond to the two spectral convolution layers in FSpecGNN.
>
> Overall, while spatial higher-order models seem preferable when fine-grained local combinatorial reasoning is the priority, whereas FSpecGNN offers a different tradeoff, combining competitive performance with better efficiency and a more interpretable spectral parameterization.
>
> ### **Reference**
> [1] Zhang, Bohang, et al. Beyond Weisfeiler-Lehman: A Quantitative Framework for GNN Expressiveness. ICLR, 2024.

---

> > ### Author Rebuttal · Reviewer_REN8 · 2026-04-04
> >
> > The addition of pseudocode improves clarity and addresses the accessibility concern, although a compact high-level summary connecting spectral formulation to implementation steps would still further benefit readers less familiar with higher-order spectral constructions.
> >
> > The clarification that FSPECGNN provides a global spectral parameterization of pair-domain propagation, while not fully subsuming all spatial higher-order models (e.g., Local 2-FGNN), is clear and appropriate, and the acknowledgment of this limitation strengthens the positioning of the method.

---

> > > ### Author Response · Authors · 2026-04-06
> > >
> > > We sincerely thank the reviewer for the constructive comments and positive feedback.
> > >
> > > We added a high-level summary connecting the spectral formulation to implementation steps; please see **[MoreSummary](https://anonymous.4open.science/r/Spec2-0363/sum_formtoimp.jpg")**. We hope this added summary better aligns with your suggestion and improves accessibility for readers who are less familiar with higher-order spectral constructions.

---

### Official Review · Reviewer_DLRx · 2026-03-12

**Soundness:** 3
**Presentation:** 3
**Significance:** 1
**Originality:** 1
**Overall Recommendation:** 4
**Confidence:** 4

**Summary:**

The paper proposes a second-order generalization of spectral GNNs by looking at "joint filters" on pairs of eigenvalues. The authors state formal results regarding the expressive power of their model, and supplement it with scalability measures. The generalization ability is validated empirically. The paper also considers the case of heterophilic graphs, arguing that their model can overcome the limitations of standard spectral GNNs.

**Compliance With Llm Reviewing Policy:**

Affirmed.

**Final Justification:**

The authors resolved most of my concerns.

**Key Questions For Authors:**

-

**Limitations:**

yes

**Strengths And Weaknesses:**

I would evaluate the proposed model along the three standard axes:

1) Expressive Power: The authors claim that "FSPECGNN is expressive: it can surpass the 1-WL upper bound. In particular, under sufficient conditions, Theorem 3.8 shows that FSPECGNN can achieve distinguishing power that exceeds Local 2-GNN (Zhang et al., 2024)."

Surpassing the 1-WL is typically never a problem. More importantly, I am not sure if a comparison with a very specific model such as Local 2-GNN, that too under certain conditions, would merit as an important contribution. This is an incremental result with limited applicability.

"Complementarily, Theorem 3.4 establishes a universal approximation result for 1-dimensional node-pair signals."

The authors hide the assumption, stated more clearly in the Appendix, that the Laplacian has a simple spectrum (all eigenvalues distinct). This is a grossly simplifying assumption: Even the Graph Isomorphism problem for such graphs can be solved efficiently in polynomial time.

2) Scalability: The authors state "FSPECGNN is also scalable: Section 3.4 discusses implementations of FSPECGNN to avoid explicit computations in the node-pair domain. We further propose a low-rank approximation in Eq. (3), which approximates a full-spectrum
convolution using only a small number of polynomial spectral filters (e.g., BERNCONV (He et al., 2021)). This leads to implementations Eq. (5) whose computational complexity is comparable to that of existing spectral GNNs."

Again, this is not surprising, given the corresponding existing work for spectral filters for graphs. Moreover,  these results breakdown in the general case of large eigenspaces (say dim D): How does one even distinguish among the quadratically large space of signals (D^2) of the form $u_i u_j^T$? The provided extension is mainly incremental, and of limited mathematical interest.

3) Generalization: I do not understanding the authors's choice of benchmarking using an unpublished arxiv paper (cited as Zhao et. al., 2021). Next: Why are homomorphism counting tasks suddenly relevant here, other than the incremental reason that they are used in earlier spectral GNN papers? In case the authors wish to do this, why is there no theoretical analysis comparing the homomorphism expressive power of their GNN, vis-a-vis standard graph classes? Again, the suite of benchmark tasks is too constrained for a thorough empirical evaluation.

---

> ### Author Rebuttal · Authors · 2026-03-30
>
> We thank the reviewer for the feedback. Since the concerns span several aspects of the paper, we address them separately and clarify the scope and claims as needed.
>
> ### **W1: Concerns on Expressive Power and Universal Approximation**
> > "Surpassing the 1-WL is typically never a problem. More importantly, I am not sure if a comparison with […] Local 2-GNN […] would merit as an important contribution."
>
> We respectfully disagree. Our contribution is not simply to introduce another model that goes beyond 1-WL. Rather, to the best of our knowledge, this is the first work to identify a structurally natural **higher-order generalization of spectral GNNs** and **place it within the WL-style expressivity hierarchy**. Prior spectral GNN mainly positioned models within the 1-WL; what was missing was a principled spectral route to higher-order expressivity. This is precisely the gap our result addresses.
>
> The comparison with Local 2-GNN is structurally natural rather than ad hoc, because its node-pair propagation is exactly the spatial counterpart of the operators $L \otimes I$ and $I \otimes L$ underlying FSpecGNN.
>
> >"The authors hide the assumption [...] grossly simplifying assumption [...]"
>
> This assumption is **not hidden**: it is explicitly stated in Theorem 3.4 of the main text (“... if the normalized Laplacian $\widetilde L$ has no multiple eigenvalues ...”).
>
> We also stress that the simple-spectrum assumption is used only in specific theoretical results, analogous to prior first-order spectral universality analyses (Theorem 4.1, [1]). It is not a blanket assumption for the model or the paper as a whole: **our model and experiments do not rely on simple spectrum**, e.g. the counting experiments follow the standard benchmark setting used in prior higher-order GNN work, with no restriction to simple-spectrum graphs.
>
> ### **W2: Concerns on Scalability**
>
> >"Again, this is not surprising, given the corresponding existing work for spectral filters for graphs."
>
> We respectfully disagree. Existing spectral GNNs can use polynomial parameterizations $p(L)$ to avoid eigendecomposition. However, as discussed in Section 3.4, for the second-order polynomial filter $p(I\otimes L, L\otimes I)$, the computational cost grows quadratically with the polynomial degree. As a result, it is still not scalable when a high polynomial degree is needed, e.g., $\deg(p)=10$ as commonly used for $p(L)$.
>
> The key step is therefore to **introduce the low-rank approximation**: $p(I\otimes L, L\otimes I)\approx\sum_{r=1}^{S} f_r(L)\otimes h_r(L)$ with a small rank $S$. This reduces second-order filtering to a complexity comparable to order-one cases while retaining off-diagonal spectral interactions. For this reason, we do not think the scalability result is a routine extension of existing order-one spectral filtering techniques.
>
> >"Moreover, these results breakdown in the general case of large eigenspaces (say dim D): [...]"
>
> If the reviewer is referring to high-multiplicity eigenvalues, this is a general issue of spectral methods, rather than a limitation specific to our models. Moreover, it does not invalidate our low-rank implementation: high-multiplicity eigenvalues are common in node classification graphs [2], yet our model still performs well.
>
> ### **W3: Concerns on Experiments**
>
> The arXiv paper was published at ICLR 2022 [3]. We will revise the manuscript accordingly.
>
> >"Why are homomorphism counting [...] incremental reason [...] no theoretical analysis comparing the homomorphism expressive power [...]"
>
> Homomorphism counting tasks are standard benchmarks for evaluating GNN expressivity, since they directly test graph distinguishability. As our paper studies expressivity beyond 1-WL, they are therefore natural benchmarks for validating the corresponding theoretical predictions.
>
> About “the incremental reason that they are used in earlier spectral GNN papers”, in fact, **prior spectral GNN work did not evaluate spectral models on expressivity benchmarks**. Our work is exactly bringing spectral GNNs into an expressivity benchmark that had previously been mainly associated with higher-order spatial models.
>
> Moreover, homomorphism expressivity is only one approach to studying model expressivity through graph distinguishability; see [4,5] for other techniques. In this paper, we instead analyze graph distinguishability by directly comparing the corresponding color refinement procedures.
>
> ### **References**
>
> [1] How powerful are spectral graph neural networks. ICML, 2022.
>
> [2] Improving expressive power of spectral graph neural networks with eigenvalue correction. AAAI, 2024.
>
> [3] From Stars to Subgraphs: Uplifting Any GNN with Local Structure Awareness. ICLR, 2022.
>
> [4] Weisfeiler and leman go sparse: Towards scalable higher-order graph embeddings. NeurIPS, 2020.
>
> [5] Weisfeiler and leman go loopy: A new hierarchy for graph representational learning. NeurIPS, 2024.

---

> > ### Author Rebuttal · Reviewer_DLRx · 2026-04-04
> >
> > See comment.

---

> > > ### Author Response · Authors · 2026-04-06
> > >
> > > We thank the reviewer for the discussion and the updated assessment. We will incorporate the corresponding revision into the manuscript.

---

### Decision · Program_Chairs · 2026-04-30

**Decision:**

Accept (regular)

**Comment:**

The paper introduces a clean and principled second-order generalization of spectral GNNs via bivariate eigenvalue-pair filtering, providing the first spectral route to higher-order expressivity that maintains practical scalability. All three reviewers confirmed their concerns were resolved after a thorough rebuttal, resulting in a positive consensus (4/5/5) on a technically solid contribution to the GNN community.